# IS INVERSE REINFORCEMENT LEARNING HARDER THAN STANDARD REINFORCEMENT LEARNING?

## ABSTRACT

Inverse Reinforcement Learning (IRL)—the problem of learning reward functions from demonstrations of an *expert policy*—plays a critical role in developing intelligent systems, such as those that understand and imitate human behavior. While widely used in applications, theoretical understandings of IRL admit unique challenges and remain less developed compared with standard RL theory. For example, it remains open how to do IRL efficiently in standard *offline* settings with pre-collected data, where states are obtained from a *behavior policy* (which could be the expert policy itself), and actions are sampled from the expert policy.

This paper provides the first line of results for efficient IRL in vanilla offline and online settings using polynomial samples and runtime. We first design a new IRL algorithm for the offline setting, Reward Learning with Pessimism (RLP), and show that it achieves polynomial sample complexity in terms of the size of the MDP, a concentrability coefficient between the behavior policy and the expert policy, and the desired accuracy. Building on RLP, we further design an algorithm Reward Learning with Exploration (RLE), which operates in a natural online setting where the learner can both actively explore the environment and query the expert policy, and obtain a stronger notion of IRL guarantee from polynomial samples. We establish sample complexity lower bounds for both settings showing that RLP and RLE are nearly optimal. Finally, as an application, we show that the learned reward functions can *transfer* to another target MDP with suitable guarantees when the target MDP satisfies certain similarity assumptions with the original (source) MDP.

## 1 INTRODUCTION

Inverse Reinforcement Learning (IRL) aims to recover reward functions from demonstrations of an *expert policy* (Ng & Russell, 2000; Abbeel & Ng, 2004), in contrast to standard reinforcement learning which aims to learn optimal policies for a given reward function. IRL has applications in numerous domains such as robotics (Argall et al., 2009; Finn et al., 2016), target-driven navigation tasks (Ziebart et al., 2008; Sadigh et al., 2017; Kuderer et al., 2015; Pan et al., 2020; Barnes et al., 2023), game AI (Ibarz et al., 2018; Vinyals et al., 2019), and medical decision-making (Woodworth et al., 2018; Hantous et al., 2022). The learned reward functions in these applications are typically used for replicating the expert behaviors in similar or varying downstream environments. Broadly, the problem of learning reward functions from data is of rising importance beyond the scope of IRL—For example, it is a main step in Reinforcement Learning from Human Feedback (RLHF) (Christiano et al., 2017), a widely adopted paradigm for aligning modern large-scale systems such as Large Language Models (Ouyang et al., 2022; Bai et al., 2022; OpenAI, 2023; Touvron et al., 2023).

Despite the significant success that IRL has achieved in practical applications (Agarwal et al., 2020; Finn et al., 2016; Sadigh et al., 2017; Kuderer et al., 2015; Woodworth et al., 2018; Wu et al., 2020; Ravichandar et al., 2020; Vasquez et al., 2014), theoretical understanding is still in an early stage, especially when compared with standard RL (finding optimal policy under a given reward) where the theory is more established. The main challenge in IRL: (i) Non-uniqueness of reward for *any* IRL problem, for example, $\mathbf{0}$ is always a solution. Therefore, it is not sufficient to merely recover one reward. The recent literature considers recovering a set of feasible rewards, defined as the rewards that enable the expert policy to achieve optimality (Metelli et al., 2021; Lindner et al., 2023; Metelli et al., 2023). (ii) As far as we know, there are no results for more standard settings in RL theory, such as learning from interactive online access to the environment (known as online RL), or from offline

demonstrations (referred to as offline RL). (iii) A more nuanced issue in IRL, relating to both (i) and (ii), concerns the selection of a distance metric between the estimated reward set and the ground truth reward set. Literature has various choices, such as reward that leads to uniformly accurate $Q/V$ functions over *all* states and actions. However, these are predominantly suitable for simulator settings and are less sensible for the online or offline settings. Recently, Lindner et al. (2023) proposes a metric for measuring the accuracy of $Q$-function merely at initial states in an online setting. However, this metric is notably weak, for instance, it fails to distinguish two IRL problems with completely different transitions.

### Is IRL more difficult than standard RL?

In this paper, we initiate the study of IRL in standard episodic tabular Markov Decision Processes (MDPs) where we propose a suitable new metrics, and design learning algorithms that are efficient in the sample complexity and runtime in both online and offline settings. Our contributions can be summarized as follows.

- We build metrics for both offline and online settings, based on the recovery of reward mapping, which converts a parameter set into solutions for an IRL problem, providing an answer to the longstanding non-uniqueness issue in IRL. Our new metrics not only extend the existing uniform accurate $V$-function metric from simulator settings to trajectory settings, but they also fully capture behavior of policies. As a result, IRL in trajectory settings becomes both learnable and practical (See Section 3).
- Leveraging the estimation of an evaluation policy, we establish a performance metric for the offline setting. Informed by the pessimism principle, a common aspect in standard offline RL, we design an offline IRL algorithm, REWARD LEARNING WITH PESSIMISM(RLP). This algorithm achieves a rate of $\widetilde{\mathcal{O}}(C^\star H^4 S^2/\epsilon^2)$ (up to log factors) when $\epsilon$ is small, where $C^\star$ is a single-policy concentrability coefficient between the behavior policy and the evaluation policy (See Section 4).
- We establish a performance metric in online settings based on the estimation of all policies. Building on reward-free exploration techniques, as well as RLP, We develop an online IRL algorithm REWARD LEARNING WITH EXPLORATION, which achieves a rate of $\widetilde{\mathcal{O}}(H^4 S^2 A/\epsilon^2)$(up to log factors) for *any* online IRL problem (See Section 5).
- We establish an $\widetilde{\mathcal{O}}(C^\star H^2 S \min\{S, A\}/\epsilon^2)$ sample complexity for lower bound for *any* offline IRL problem. Turning to online settings, we prove a sample complexity lower bound of $\widetilde{\mathcal{O}}(C^\star H^2 SA \min\{S, A\}/\epsilon^2)$ for *any* online IRL problem. When $S < A$, our upper bounds for offline/online settings match the lower bounds except for the horizon factor (See Section 4-5).
- We further provide results for a *transfer learning* setting, where the learned reward mapping is transferred to and evaluated in a target MDP\R(different from the original MDP\R). We provide theoretical guarantees for both RLP and RLE, under certain similarity assumptions between the original (source) and target MDP\Rs that are more relaxed compared to the constraints in existing work within this context (See Section 6).

## 1.1 RELATED WORK

**Inverse reinforcement learning**   Inverse reinforcement learning (IRL) was first proposed by Ng & Russell (2000). Since its introduction, IRL has been enhanced by numerous influential studies that have made notable contributions to the field. These include feature matching (Abbeel & Ng, 2004), maximum margin (Ratliff et al., 2006), maximum entropy (Ziebart et al., 2008), and two model-free methods: relative entropy (Boularias et al., 2011), and generative adversarial imitation learning (Ho & Ermon, 2016). Other significant works include Bayesian IRL (Ramachandran & Amir, 2007) which subsume IRL and reduction method (Brantley et al., 2019).

IRL has been successfully applied in many domains including target-driven navigation tasks (Ziebart et al., 2008; Sadigh et al., 2017; Kuderer et al., 2015; Pan et al., 2020), robotics (Argall et al., 2009; Finn et al., 2016; Hadfield-Menell et al., 2016; Kretzschmar et al., 2016; Okal & Arras, 2016; Kumar et al., 2023; Jara-Ettinger, 2019), medical decision-making (Woodworth et al., 2018; Hantous et al., 2022; Gong et al., 2023; Yu et al., 2019; Chadi & Mousannif, 2022), and game play AI (Finn et al., 2016; Fu et al., 2017; Qureshi et al., 2018; Brown et al., 2019).

Despite their successful applications, these methods offer limited theoretical guarantees. Recently, Metelli et al. (2021) pioneered the investigation of the sample complexity of IRL within generative

models. Their work was expanded upon by Metelli et al. (2023), who introduced a framework based on Hausdorff-based metrics for measuring distances between reward sets, examined relationships between different metrics, and provided corresponding lower bounds. However, these results still rely on the premise of generative models, which might not align with real-world situations. Dexter et al. (2021) also contributed to the theoretical analysis in the generative setting, but their focus was on MDP\R with continuous states and discrete actions. Subsequently, Lindner et al. (2023) ventured into the exploration setting, thereby circumventing the need for generative models. Nonetheless, their metric, which relies on rewards leading to accurate $Q$ functions over the initial state, has a significant weakness. Our work strives to address these limitations by introducing more suitable metrics and extending IRL to both online and offline learning settings, which are more characteristic of real-world applications.

**Relationship with existing RL theory** Our work builds upon various existing techniques from the sample-efficient RL literature to design our algorithms and establish our theoretical results. For the offline setting, we utilize the pessimism principle, a commonly adopted strategy in standard offline RL, to construct our offline IRL algorithm, REWARD LEARNING WITH PESSIMISM (RLP). Moreover, our assumption (Assumption 4.1) takes inspiration from the single-policy concentrability assumption prevalent in offline RL (Kidambi et al., 2020; Jin et al., 2021; Yu et al., 2020; Kumar et al., 2020; Rashidinejad et al., 2021; Xie et al., 2021; 2022). For online exploration, we rely on the reward-free exploration techniques (Jin et al., 2020; Li et al., 2023) to find a desired behavior policy. Specifically, we adopt the exploration scheme proposed by Li et al. (2023), allowing effective exploration of the unknown environment and the identification of a desired behavior policy that fulfills a certain coverage assumption with *all* policies. Furthermore, we extend the theoretical analysis of this scheme to the online IRL setting. We note theoretical results on imitation learning (Abbeel & Ng, 2004; Ratliff et al., 2006; Ziebart et al., 2008; Levine et al., 2011; Fu et al., 2017) and reinforcement learning from human feedback (Zhu et al., 2023a;b; Wang et al., 2023; Zhan et al., 2023); our setting is related to these problems, though the techniques are different and the results do not directly imply each other. Additional related work in Appendix A.

## 2 PRELIMINARIES

**Markov Decision Processes without Reward.** We consider episodic Markov Decision Processes without Reward (MDP\R), specified by $\mathcal{M} = (\mathcal{S}, \mathcal{A}, H, \mathbb{P})$, where $\mathcal{S}$ is the state space with $|\mathcal{S}| = S$, $\mathcal{A}$ is the action space with $|\mathcal{A}| = A$, $H$ is the horizon length, $\mathbb{P} = \{\mathbb{P}_h\}_{h \in [H]}$ where $\mathbb{P}_h(\cdot|s,a) \in \Delta(\mathcal{S})$ is the transition probability at step $h$.

**Reward functions.** A reward function $r : [H] \times \mathcal{S} \times \mathcal{A} \to [-1,1]$ maps a state-action-time step triplet $(h,s,a)$ to a reward $r_h(s,a)$. Given an MDP\R $\mathcal{M}$ and a reward function $r$, we denote the MDP induced by $\mathcal{M}$ and $r$ as $\mathcal{M} \cup r$. A policy $\pi = \{\pi_h(\cdot\,|\,s)\}_{h \in [H], s \in \mathcal{S}}$, where $\pi_h : \mathcal{S} \to \Delta(\mathcal{A})$ maps a state to a acion distribution.

**Values and visitation distributions.** For any policy $\pi$ and any reward function $r$, we define the value function $V_h^\pi(\cdot\,;r) : \mathcal{S} \to \mathbb{R}$ at each time step $h \in [H]$ by the expected cumulative reward: $V_h^\pi(s;r) = \mathbb{E}_\pi \left[ \sum_{h'=h}^H r_{h'}(s_{h'}, a_{h'}) \,|\, s_h = s \right]$, where $\mathbb{E}_\pi$ denotes the expectation with respect to the random trajectory induced by $\pi$ in the MDP\R, that is, $(s_1, a_1, s_2, a_2, ..., s_H, a_H)$, where $a_h \sim \pi_h(s_h), r_h = r_h(s_h, a_h), s_{h+1} \sim \mathbb{P}_h(\cdot\,|\,s_h, a_h)$. Similarly, we denote the $Q$-function at time step $h$ as : $Q_h^\pi(s, a; r) = \mathbb{E}_\pi \left[ \sum_{h'=h}^H r_{h'}(s_{h'}, a_{h'}) \,|\, s_h = s, a_h = a \right]$. The advantage function $A_h^\pi(\cdot\,;) : \mathcal{S} \times \mathcal{A} \to \mathbb{R}$ is defined as $A_h^\pi(s, a; r) := Q_h^\pi(s, a; r) - V_h^\pi(s; r)$ and we say a policy is an optimal policy of $\mathcal{M} \cup r$ if $A_h^\pi(s, a; r) \leq 0$ holds for all $(h, s, a) \in [H] \times \mathcal{S} \times \mathcal{A}$. Additionally, we represent the set of all optimal policies for $\mathcal{M} \cup r$ as $\Pi_{\mathcal{M} \cup r}^\star$ and denote the set of all deterministic policies for $\mathcal{M} \cup r$ as $\Pi_{\mathcal{M} \cup r}^{\det}$.

We introduce $d_h^\pi$ to denote the state(-action) visitation distributions associated with policy at time step $h \in [H]$: $d_h^\pi(s) := \mathbb{P}(s_h = s|\pi)$ and $d_h^\pi(s, a) := \mathbb{P}(s_h = s, a_h = a|\pi)$. Lastly, We define the operators $\mathbb{P}_h$ and $\mathbb{V}_h$ by $[\mathbb{P}_h V_{h+1}](s, a) := \mathbb{E}[V_{h+1}(s_{h+1})|s_h = s, a_h = a]$ and $[\mathbb{V}_h V_{h+1}](s, a) := \mathrm{Var}[V_{h+1}(s_{h+1})|s_h = s, a_h = a]$ applying to any value function $V_{h+1}$ at time step $h + 1$. In this paper, we will frequently employ $\widehat{\mathbb{P}}_h$ and $\widehat{\mathbb{V}}_h$ to represent empirical counterparts of these operators constructed based on estimated models.

## 2.1 INVERSE REINFORCEMENT LEARNING

An Inverse Reinforcement Learning (IRL) problem is denoted as a pair $(\mathcal{M}, \pi^{\mathsf{E}})$, where $\mathcal{M}$ is an MDP\R and $\pi^{\mathsf{E}}$ is a policy called the *expert policy*. The goal of IRL is to interact with $(\mathcal{M}, \pi^{\mathsf{E}})$, and recover reward function $r$ that are *feasible* for $(\mathcal{M}, \pi^{\mathsf{E}})$, in the sense that $\pi^{\mathsf{E}}$ an optimal policy for MDP $\mathcal{M} \cup r$. Noting that learning *one* feasible reward function is trivial (the zero reward $r \equiv 0$ is feasible for any $\pi^{\mathsf{E}}$), we consider the stronger goal of recovering a *set* of feasible rewards. Concretely, we focus on recovering a specific and diverse set of feasible rewards widely considered in the IRL literature (Ng & Russell, 2000; Metelli et al., 2023; Lindner et al., 2023), which we restate through the concept of a *reward mapping*.

Let $\overline{\mathcal{V}} = \overline{\mathcal{V}}_1 \times \cdots \times \overline{\mathcal{V}}_H$ and $\overline{\mathcal{A}} = \overline{\mathcal{A}}_1 \times \cdots \times \overline{\mathcal{A}}_H$, where $\overline{\mathcal{V}}_h := \left\{ V_h \in \mathbb{R}^S \,|\, \|V_h\|_\infty \leq H - h + 1 \right\}$ and $\overline{\mathcal{A}}_h := \left\{ A_h \in \mathbb{R}^{S \times A}_{\geq 0} \,|\, \|A_h\|_\infty \leq H - h + 1 \right\}$. Here, sets $\overline{\mathcal{V}}$ and $\overline{\mathcal{A}}$ serve as parameter sets, representing the parameters of the $V$-functions and the advantage functions, respectively. Let $\mathcal{R}^{\mathsf{all}}$ denote the set of all possible reward functions on $[H] \times \mathcal{S} \times \mathcal{A}$.

**Definition 2.1** (Reward mapping). *The (ground truth) reward mapping $\mathscr{R} : \overline{\mathcal{V}} \times \overline{\mathcal{A}} \mapsto \mathcal{R}^{\mathsf{all}}$ for an IRL problem $(\mathcal{M}, \pi^{\mathsf{E}})$ is the mapping that maps any $(V, A) \in \overline{\mathcal{V}} \times \overline{\mathcal{A}}$ to the following reward function $r$:*

$$r_h(s,a) = [\mathscr{R}(V, A)]_h(s,a) := -A_h(s,a) \times \mathbf{1}\left\{ a \notin \mathrm{supp}\left( \pi^{\mathsf{E}}_h(\cdot \,|\, s) \right) \right\} + V_h(s) - [\mathbb{P}_h V_{h+1}](s,a), \tag{2.1}$$

*where we recall that $\mathbb{P}_h$ is the transition probability of $\mathcal{M}$ at step $h$.*

We will also use notation $\widehat{\mathscr{R}}$ to denote an estimated reward mapping, which is in general an arbitrary mapping from $\overline{\mathcal{V}} \times \overline{\mathcal{A}} \mapsto \mathcal{R}^{\mathsf{all}}$. The importance of the ground truth reward mapping is that it always produces feasible rewards, as stated in the following classical result (Ng & Russell, 2000) (see also Lindner et al. (2023)).

**Lemma 2.2** (Reward mapping produces a set of feasible rewards). *Let $\mathscr{R}$ be the (ground truth) reward mapping for $(\mathcal{M}, \pi^{\mathsf{E}})$. Then for any $(V, A) \in \overline{\mathcal{V}} \times \overline{\mathcal{A}}$, the reward function $r = \mathscr{R}(V, A)$ in Eq.(2.1) is feasible for $(\mathcal{M}, \pi^{\mathsf{E}})$. Further, if feasible reward $r$ satisfies that $\sup_{(h,s,a) \in [H] \times \mathcal{S} \times \mathcal{A}} |r_h(s,a)| \leq 1$, there exists a pair $(V, A) \in \overline{\mathcal{V}} \times \overline{\mathcal{A}}$ such that $r = \mathscr{R}(V, A)$.*

Lemma 2.2 suggests the recovery of the reward mapping $\mathscr{R}$ itself as a natural learning goal for IRL–an accurate estimator $\widehat{\mathscr{R}} \approx \mathscr{R}$ guarantees $\widehat{\mathscr{R}}(V, A) \approx \mathscr{R}(V, A)$ for any $(V, A) \in \overline{\mathcal{V}} \times \overline{\mathcal{A}}$, and thus imply accurate estimation of the entire set $\mathscr{R}(\overline{\mathcal{V}} \times \overline{\mathcal{A}})$ in precise ways which we specify in the sequel.

We will frequently consider recovering the reward mapping on a *subset* $\Theta \subset \overline{\mathcal{V}} \times \overline{\mathcal{A}}$. We use the following standard definition of covering numbers to measure the capacity of such $\Theta$'s:

**Definition 2.3** (Covering number). *For any $\Theta \subset \overline{\mathcal{V}} \times \overline{\mathcal{A}}$ let $\overline{\mathcal{V}}^\Theta_h := \{V_h : (V, A) \in \Theta\}$ denote the restriction of $\Theta$ onto $\overline{\mathcal{V}}_h$ for any $h \in [H]$. The $\epsilon$-covering number of $\Theta$ is defined as*

$$\mathcal{N}(\Theta; \epsilon) := \max_{h \in [H]} \mathcal{N}(\overline{\mathcal{V}}^\Theta_h; \epsilon),$$

*where $\mathcal{N}(\overline{\mathcal{V}}^\Theta_h; \epsilon)$ is the $\epsilon$-covering number of $\overline{\mathcal{V}}^\Theta_h$ w.r.t. $\|\cdot\|_\infty$ norm.*

We always have $\log \mathcal{N}(\Theta; \epsilon) \leq \min \{\log |\Theta|, \mathcal{O}(S \log(H/\epsilon))\}$ by combining the (trivial) bound for the finite case and the standard covering number bound for the "full set" case with $\Theta = \overline{\mathcal{V}} \times \overline{\mathcal{A}}$ (Vershynin, 2018) respectively. Further, the left-hand side can be much smaller than the right-hand side if $\Theta$ admits additional structure (for example, if $\overline{\mathcal{V}}^\Theta_h$ lies in a low-dimensional subspace of $\mathbb{R}^S$).

## 3 PERFORMANCE METRICS FOR IRL

### 3.1 METRIC FOR IRL

Our metric for IRL is to recover the ground truth reward mapping $\mathscr{R}$ in a suitable distance. Given a policy $\pi$, we first define a metric for rewards by measuring the difference between the $V$-functions of $\pi$ induced by two rewards. Then, when presented with a parameter set $\Theta \in \overline{\mathcal{V}} \times \overline{\mathcal{A}}$, we can naturally define a metric for reward mappings based on the metric for rewards we defined before.

**Definition 3.1.** *Given an MDP\R $\mathcal{M}$ and a policy $\pi$, We define the (pre)metric $d^\pi$ for any pair $(r, r') \in \mathcal{R}^{\mathsf{all}} \times \mathcal{R}^{\mathsf{all}}$ by*

$$d^\pi(r, r') := \sup_{h \in [H]} \mathbb{E}_\pi |V_h^\pi(s_h; r) - V_h^\pi(s_h; r')|. \tag{3.1}$$

*We further define $d^{\mathsf{all}}(r, r') := \sup_\pi d^\pi(r, r')$.*

**Definition 3.2.** *Given an MDP\R $\mathcal{M}$ and a policy $\pi$, we define the (pre)metric $D_\Theta^\pi$ for any pair $(\mathcal{R}, \mathcal{R}')$ by*

$$D_\Theta^\pi(\mathcal{R}, \mathcal{R}') := \sup_{(V, A) \in \Theta} d^\pi(\mathcal{R}(V, A), \mathcal{R}'(V, A)). \tag{3.2}$$

*We further define $D_\Theta^{\mathsf{all}}(\mathcal{R}, \mathcal{R}') := \sup_\pi D_\Theta^\pi(\mathcal{R}, \mathcal{R}') = \sup_{(V, A) \in \Theta} d^{\mathsf{all}}(\mathcal{R}(V, A), \mathcal{R}'(V, A))$ correspondingly.*

$d^\pi$ is expressed via $|V_h^\pi(\cdot \mid r) - V_h^\pi(\cdot \mid r')|$ over the visitation distributions at *any* horizon induced by the policy $\pi$, which guarantees that this metric comprehensively captures the behavior of $\pi$ across *all* reachable states. In Section 3.3, we will see that our paper considers the trajectory setting in which we can only interact with the environment through trajectories. As a result, we can only require accurate recovery in frequently visitable states and do not care about accurate recovery in rarely visitable states. $D_\Theta^\pi$ is defined via the worst-case $d^\pi$ over the parameter set $\Theta$. Furthermore, taking supremum over $\pi$, we obtain two stronger metircs $d^{\mathsf{all}}$ and $D_\Theta^{\mathsf{all}}$.

### 3.2 Relationship with existing metrics

Here we will mainly discuss the relationship between our metric defined in Section 3.1 and metrics in the existing works (Metelli et al., 2021; 2023; Lindner et al., 2023). The main differences lie in two aspects: (i) Metrics between rewards(differences of $d$). (ii) Reward aggregation approaches(differences of $D$).

**Metrics between rewards.** We compare our metric to the $d_{V^\star}^G$ ( Eq.(C.1)) metric proposed by Metelli et al. (2023), specifically in regards to how they measure the difference between $V$-functions associated with different rewards. We prove that our method is stronger than the one employed in Metelli et al. (2023), and on the other hand, there exist specific counterexamples where our method is able to make distinctions, while their approach fails to do so (Lemma C.1).

For metric $d^{\mathsf{all}}$, we show a significant characteristic that is not found in existing metrics: a small $d^{\mathsf{all}}$ distance between $(r, \widehat{r})$ indicates that near-optimal policies under reward $\widehat{r}$ continue to be nearly optimal in the environment induced by $r$ (Proposition C.5). This reveals a crucial application of learned rewards that exhibit small $d^{\mathsf{all}}$ distances from the ground truth rewards: these learned rewards can be utilized to execute standard RL algorithms that aim to find near-optimal policies. We defer the detailed discussion of metrics between rewards to Appendix C.1.

**Reward aggregation approaches.** Given a metric $d$ defined on $\mathcal{R}^{\mathsf{all}}$, our paper employs a reward aggregation approach that relies on utilizing a (pre)metric $D^{\mathsf{M}}$ between reward mappings, induced by $d$. Contrarily, Metelli et al. (2021); Lindner et al. (2023); Metelli et al. (2023) utilized a Hausdorff (pre)metric to derive a metric $D^{\mathsf{H}}$ between reward sets, induced by $d$. It will be demonstrated that if $D^{\mathsf{M}}$ and $D^{\mathsf{H}}$ are induced by the same reward metric, then $D^{\mathsf{M}}$ is strictly stronger than $D^{\mathsf{H}}$. Broadly speaking, our metric $D^{\mathsf{M}}$ can capture more comprehensive information for a single parameter pair $(V, A)$. Additionally, in certain scenarios, applying the Hausdorff metric can trivialize the problem. We defer the detailed discussion of the reward aggregation approaches to Appendix C.2. We also consider merics used in Lindner et al. (2023) and show that this metric can't fully capture transitions of MDP\R (Proposition C.4).

### 3.3 Learning settings

In this paper, we consider two settings: the offline setting and the online setting. The latter is also known as active exploration IRL, as proposed in Lindner et al. (2023). Next, we give detailed definitions of the two settings.

**Offline IRL.** A learner only has access to a dataset $\mathcal{D}$ consisting of $K$ trajectories without reward $\{(s_h^\tau, a_h^\tau, e_h^\tau)\}_{\tau=1, h=1}^{K, H}$. Here $s_1$ is a fixed initial state, $a_h \sim \pi_h^{\mathsf{b}}(\cdot | s_h)$ and $s_{h+1} \sim \mathbb{P}_h(\cdot | s_h, a_h)$,

where $\pi^{\mathsf{b}}$ is a behavior policy (is unknown to the learner), and $e_h$ represents an expert feedback, which is given by

$$e_h = \begin{cases} a_h^{\mathsf{E}} \sim \pi_h^{\mathsf{E}}(\cdot|s_h) & \text{in option 1,} \\ \mathbf{1}\left\{a_h \in \operatorname{supp}\left(\pi_h^{\mathsf{E}}(\cdot|s_h)\right)\right\} & \text{in option 2.} \end{cases} \tag{3.3}$$

Option 1 is a commonly employed setting in the related literature (Metelli et al., 2021; Lindner et al., 2023; Metelli et al., 2023). But we also consider Option 2, since Option 2 is simpler and more fundamental for IRL problems when compared to Option 1, as indicated by Eq.(2.1). Furthermore, if we opt for Option 1, we need to enforce the following well-posed assumption for the expert policy $\pi^{\mathsf{E}}$.

**Assumption 3.3** (Well-posedness). *We assume that there exists a $\Delta > 0$ such that*

$$\min_{(h,s,a):\pi_h^{\mathsf{E}}(a|s)\neq 0} \pi^{\mathsf{E}}(a|s) \geq \Delta. \tag{3.4}$$

Assumption 3.3 is also made in the works of Metelli et al. (2023). We remark that Assumption 3.3 is necessary: if an optimal action $a$ at a frequently visited state is never returned (which might occur when $\pi_h^{\mathsf{E}}(a|s)$ is too small), we are consequently unable to determine $\mathbf{1}\left\{a \in \operatorname{supp}\left(\pi_h^{\mathsf{E}}(\cdot|s)\right)\right\}$, which could result in a significant error as revealed in Eq.(2.1).

Constrained by the offline setting, our access is limited to the trajectories executed by the behavior policy $\pi^{\mathsf{b}}$. Here, we cannot reach states with low visitation probabilities under $\pi^{\mathsf{b}}$ and therefore expect to recover a reward mapping $\widehat{\mathscr{R}}$ with a small $D_{\Theta}^{\pi}(\widehat{\mathscr{R}}, \mathscr{R}^{\star})$ for *any* $\pi$ is unrealistic. Instead, our goal in the offline setting is to recover a reward mapping $\widehat{\mathscr{R}}$ attaining a small $D_{\Theta}^{\pi^{\mathsf{val}}}(\widehat{\mathscr{R}}, \mathscr{R}^{\star})$. for $\pi^{\mathsf{val}}$ that exhibits concentrability (Assumption 4.1) with the behavior policy $\pi^{\mathsf{b}}$, where $\mathscr{R}^{\star}$ denotes the ground truth reward mapping. In this paper, we utilize the notation $(\mathcal{M}, \pi^{\mathsf{E}}, \pi^{\mathsf{b}}, \pi^{\mathsf{val}})$ to denote an offline IRL problem.

**Online IRL** A learner can obtain a trajectory as follows. Each episode starts with a fixed initial state $s_1$.

For each $h$, the learner observes $s_h$, chooses an action $a_h$ to play, then receives a feedback $e_h$ and transits to the next states $s_{h+1} \sim \mathbb{P}_h(\cdot \mid s_h, a_h)$. In this case, when we receive the first type of feedback, the well-posed condition described in Eq.(3.4) remains necessary.

In this setting, we possess the freedom to explore the environment unrestricted by a certain behavior policy. Consequently, we can approximate the ground truth mapping on any policy within a given parameter set $\Theta \subset \overline{\mathcal{V}} \times \overline{\mathcal{A}}$. In other words, our learning goal in the online learning setting is to recover a reward mapping that exhibits a small $D_{\Theta}^{\mathsf{all}}$ distance compared to the ground truth reward mapping.

## 4 INVERSE REINFORCEMENT LEARNING IN THE OFFLINE SETTING

In this section, we initially introduce our algorithm, referred to as REWARD LEARNING WITH PESSIMISM (RLP). Subsequently, we provide the sample complexity of this algorithm and conclude by presenting a lower bound result for the IRL problem in the offline setting.

### 4.1 ALGORITHM

In this section, we propose a meta-algorithm, termed REWARD LEARNING WITH PESSIMISM (full description in Algorithm 1) to solve the offline IRL problem defined in Section 3.3. We also prove that RLP is provably sample-efficient, i.e., RLP can recover ground truth reward mapping with small learning error, within a polynomial number of samples.

At a high level, RLP simultaneously recovers the transition kernel of MDP\R and the expert policy, integrating the imitation learning algorithm (Rajaraman et al., 2020) and the pessimistic algorithm for offline RL (Li et al., 2023).

RLP utilizes empirical MDP and Pessimism frameworks, operating through two distinct phases.

- (Empirical MDP): In this phase, we construct an estimated transition kernel $\widehat{\mathbb{P}}_h$ and an estimated expert policy $\widehat{\pi}^{\mathsf{E}}$ according to Eq.(4.1) and Eq.(4.2). So that $\widehat{\mathbb{P}}_h V_{h+1} : \mathcal{S} \times \mathcal{A} \to \mathbb{R}$ approximates $\mathbb{P}_h V_{h+1} : \mathcal{S} \times \mathcal{A} \to \mathbb{R}$ and $\operatorname{supp}\left(\widehat{\pi}_h^{\mathsf{E}}(\cdot|s)\right)$ approximates $\operatorname{supp}\left(\pi_h^{\mathsf{E}}(\cdot|s)\right)$.

---

**Algorithm 1** REWARD LEARNING WITH PESSIMISM

---

1: **Input:** Dataset $\mathcal{D} = \{(s_h^\tau, a_h^\tau, e_h^\tau)\}_{\tau=1,h=1}^{K,H}$ collected by executing $\pi^{\mathsf{b}}$ in $\mathcal{M}$, parameter set $\Theta \in \overline{\mathcal{V}} \times \overline{\mathcal{A}}$, confidence level $\delta$ and error tolerance $\epsilon$.

2: **Initialization:** $\widehat{\mathscr{R}}(V, A) = 0$ for all $\theta = (V, A) \in \Theta$.

3: **for** $(h, s, a) \in [H] \times \mathcal{S} \times \mathcal{A}$ **do**

4: Compute the empirical transition kernel $\widehat{\mathbb{P}}_h$, the empirical expert policy $\widehat{\pi}^{\mathsf{E}}$ and the penalty term $b_h^\theta$ for all $\theta \in \Theta$ as follows:

$$\widehat{\mathbb{P}}_h(s' \mid s, a) = \frac{1}{N_h^b(s, a) \vee 1} \sum_{(s_h, a_h, s_{h+1}) \in \mathcal{D}} \mathbf{1}\left\{(s_h, a_h, s_{h+1}) = (s, a, s')\right\}, \tag{4.1}$$

$$\widehat{\pi}_h^{\mathsf{E}}(a \mid s) = \begin{cases} \frac{1}{N_h^b(s) \vee 1} \cdot \sum_{(s_h, a_h, e_h) \in \mathcal{D}} \mathbf{1}\left\{(s_h, e_h) = (s, a)\right\} & \text{in option 1,} \\ \frac{1}{N_{h,1}^b(s) \vee 1} \cdot \sum_{(s_h, a_h, e_h) \in \mathcal{D}} \mathbf{1}\left\{(s_h, a_h, e_h) = (s, a, 1)\right\} & \text{in option 2,} \end{cases} \tag{4.2}$$

$$b_h^\theta(s, a) =$$
$$C \cdot \min\left\{\sqrt{\frac{\log \mathcal{N}(\Theta; \epsilon/H)\iota}{N_h^b(s, a) \vee 1}\left[\widehat{\mathbb{V}}_h V_{h+1}\right](s, a)} + \frac{H \log \mathcal{N}(\Theta; \epsilon/H)\iota}{N_h^b(s, a) \vee 1} + \frac{\epsilon}{H}\left(1 + \sqrt{\frac{\log \mathcal{N}(\Theta; \epsilon/H)\iota}{N_h^b(s, a) \vee 1}}\right), H\right\}, \tag{4.3}$$

where $N_h^b(s, a) := \sum_{(s_h, a_h) \in \mathcal{D}} \mathbf{1}\left\{(s_h, a_h) = (s, a)\right\}$, $N_h^b(s) := \sum_{a \in \mathcal{A}} N_h^b(s, a)$, $N_{h,1}^b(s) := \sum_{(s_h, a_h, e_h)} \mathbf{1}\left\{(s_h, e_h) = (s, 1)\right\}$, $\iota := \log(HSA/\delta)$ and $C > 0$ is an absolute constant.

5: Compute $\widehat{\mathscr{R}}$ by

$$[\widehat{\mathscr{R}}(V, A)]_h(s, a) = -A_h(s, a) \cdot \mathbf{1}\left\{a \notin \mathrm{supp}\left(\widehat{\pi}_h^{\mathsf{E}}(\cdot | s)\right)\right\} + V_h(s) - [\widehat{\mathbb{P}}_h V_{h+1}](s, a) - b_h^\theta(s, a) \tag{4.4}$$

6: **Output**: Estimated reward mapping $\widehat{\mathscr{R}}$.

---

- (Pessimism in face of uncertainty): In this phase, a penalty function $b_h^\theta(s, a)$ is computed for any $\theta = (V, A) \in \Theta$, $(h, s, a) \in [H] \times \mathcal{S} \times \mathcal{A}$ using Eq.(4.3). Here $b_h^\theta$ serves as a measure of uncertainty arising from approximating $\widehat{\mathbb{P}}_h V_{h+1}$ with $\mathbb{P}_h V_{h+1}$. Finally, we compute estimated $\widehat{\mathscr{R}}(V, A)$ for all $\theta = (V, A) \in \Theta$ by Eq.(4.4), which incorporates the explicit form of reward mapping Eq.(2.1) with penalty terms.

## 4.2 THEORETICAL GUARANTEE

**Assumption 4.1** (Single-policy concentrability in the $\mathcal{L}^1$ form). *We assume that evaluation policy $\pi^{\mathsf{val}}$ satisfies the following single-policy concentrability with behavior policy $\pi^{\mathsf{b}}$:*

$$\sum_{h \in [H]} \sum_{(s,a) \in \mathcal{S} \times \mathcal{A}} \frac{d_h^{\pi^{\mathsf{val}}}(s, a)}{d_h^{\pi^{\mathsf{b}}}(s, a)} \leq C^\star HS., \tag{4.5}$$

*Here $C^\star$ is a constant and we follow the convention: $0/0 = 0$.*

Assumption 4.1 characterizes the gap between the visitation distributions of the evaluation policy $\pi^{\mathsf{val}}$ and the behavior policy $\pi^{\mathsf{b}}$. The assumption of single-policy concentrability is common in the literature of RL (Xie et al., 2021). Our $\mathcal{L}^1$ form single-policy concentrability assumption is milder than the $\mathcal{L}^\infty$ form typically adopted in most offline RL works (Xie et al., 2021; Rashidinejad et al., 2021). This is due to the fact that every policy pair exhibiting $C^\star$ single-policy concentrability in the $\mathcal{L}^\infty$ form also satisfies $C^\star A$ single-policy concentrability in the $\mathcal{L}^1$ form. Now, we are already to present the sample complexity of RLP (Algorithm 1).

**Theorem 4.2** (Sample complexity of RLP). *Suppose the single-policy concentrability between $\pi^{\mathsf{val}}$ and $\pi^{\mathsf{b}}$ (Assumption 4.1). In addition, we assume $\pi^{\mathsf{E}}$ is well-posed (Assumption 3.3) when we receive feedback in option 1. Then with probability at least $1 - \delta$, RLP (Algorithm 1) outputs a reward mapping $\widehat{\mathscr{R}}$ such that $\left[\widehat{\mathscr{R}}(V, A)\right]_h(s, a) \leq [\mathscr{R}^\star(V, A)]_h(s, a)$ for all $(V, A) \in \Theta$ and $(h, s, a) \in [H] \times \mathcal{S} \times \mathcal{A}$, and $D_\Theta^{\pi^{\mathsf{val}}}\left(\mathscr{R}^\star, \widehat{\mathscr{R}}\right) \leq 2\epsilon$, within*

$$\widetilde{\mathcal{O}}\left(\frac{C^\star H^4 S \log \mathcal{N}(\Theta; \epsilon/H)}{\epsilon^2} + \frac{C^\star H^2 S(\eta + H \log \mathcal{N}(\Theta; \epsilon/H))}{\epsilon}\right) \tag{4.6}$$

*samples, where $poly \log{(H, S, A, 1/\delta)}$ are omitted and $\eta$ is defined as $1/\Delta$ in option $1$ and as $1$ in option $2$ ($\Delta$ is specified in Definition 3.3).*

The proof of Theorem 4.2 is deferred to Section E. To the best of our knowledge, this result is the first theoretical guarantee for offline IRL algorithms. To highlight the main term of our results, we further character the log-covering number $\log \mathcal{N}(\Theta; \epsilon/H)$ in the following scenarios: (i) When $\Theta$ is a finite parameter set, the log-covering number $\log \mathcal{N}(\Theta; \epsilon/H)$ can be upper-bounded by its cardinality $|\Theta|$. As a result, our sample complexity becomes $\widetilde{\mathcal{O}}\big(C^{\star} H^4 S \log |\Theta|/\epsilon^2\big)$ when $\epsilon$ is small. (ii) When $\Theta = \overline{\mathcal{V}} \times \overline{\mathcal{A}}$, the log-covering number $\log \mathcal{N}(\Theta; \epsilon/H)$ can be upper-bounded by $\widetilde{\mathcal{O}}(S)$, resulting in a sample complexity of $\widetilde{\mathcal{O}}\big(C^{\star} H^4 S^2/\epsilon^2\big)$. And it's worth noting that in the case where the feedback is in option 1, $1/\Delta$ brought by the well-posedness assumption only emerges in the burn-in term.

We highlight the scenario where $\pi^{\mathsf{val}} = \pi^{\mathsf{E}}$, as this setting is prevalent in cases where only expert trajectories are available, i.e., $\pi^{\mathsf{b}} = \pi^{\mathsf{E}}$.

**Corollary 4.3** ($\pi^{\mathsf{val}} = \pi^{\mathsf{E}}$)**.** *Suppose the single-policy concentrability between $\pi^{\mathsf{val}}$ and $\pi^{\mathsf{b}}$ (Assumption 4.1) and $\sup_{(h,s,a) \in [H] \times \mathcal{S} \times \mathcal{A}} |[\mathscr{R}^{\star}(V, A)]_h(s, a)| \leq 1$ for any $(V, A) \in \Theta$, where $\mathscr{R}^{\star}$ is the ground truth reward mapping. Then with probability at least $1 - \delta$, RLP (Algorithm 1) outputs a reward mapping $\widehat{\mathscr{R}}$ such that $\left[\widehat{\mathscr{R}}(V, A)\right]_h(s, a) \leq [\mathscr{R}^{\star}(V, A)]_h(s, a)$ for all $(V, A) \in \Theta$ and $(h, s, a) \in [H] \times \mathcal{S} \times \mathcal{A}$, and $D_{\Theta}^{\pi^{\mathsf{val}}}\left(\mathscr{R}^{\star}, \widehat{\mathscr{R}}\right) \leq 2\epsilon$, within*

$$\widetilde{\mathcal{O}}\left(\frac{C^{\star} H^3 S \log \mathcal{N}(\Theta; \epsilon/H)}{\epsilon^2} + \frac{C^{\star} H^2 S(A + H \log \mathcal{N}(\Theta; \epsilon/H))}{\epsilon}\right), \tag{4.7}$$

*samples, where $poly \log{(H, S, A, 1/\delta)}$ are omitted.*

When $\pi^{\mathsf{val}} = \pi^{\mathsf{E}}$, we can gain an additional $H$ factor in the main term due to a total invariance property (See Section E). Additionally, when $\pi^{\mathsf{val}} = \pi^{\mathsf{E}}$, the necessity of the well-posedness assumption is alleviated, as demonstrated in Eq.(4.7). This stems from the concentrability between $\pi^{\mathsf{E}}$ and $\pi^{\mathsf{b}}$, facilitating the learning of supp $\left(\pi_h^{\mathsf{E}}(\cdot|s)\right)$ even without the well-posedness assumption. We remark that Theorem 4.3 also provides sample complexity of a significant scenario where the data consists of full trajectories drawn from $\pi^{\mathsf{E}}$, i.e., $\pi^{\mathsf{E}} = \pi^{\mathsf{b}}$. In this case, we have $C^{\star} = 1$, and therefore the sample complexity can be further reduced to $\widetilde{\mathcal{O}}(H^3 S \log \mathcal{N}(\Theta; \epsilon/H)/\epsilon^2)$.

**Theoretical lower bound in the offline setting.** We provide the following lower bound, for the case where $\Theta = \overline{\mathcal{V}} \times \overline{\mathcal{A}}$. This lower bound shows that our rate is sharp when $S \leq A$.

**Theorem 4.4** (Informal version of Theorem H.2)**.** *Suppose $\Theta = \overline{\mathcal{V}} \times \overline{\mathcal{A}}$, then any algorithm that returns a reward mapping up to $\epsilon$ distance with the ground truth reward mapping for all offline IRL problems with probability at least $2/3$ has to take at least $\Omega\big(C^{\star} H^2 S \min\{S, A\}/\epsilon^2\big)$ samples.*

Additionally, in Section E.4, we provide a discussion about RLP through a unifying framework to illustrate how pessimism can yield the IRL guarantee in a modular fashion.

## 5 IRL IN THE ONLINE SETTING

In this section, we extend IRL to the standard online learning setting. We present our online IRL algorithm, referred to as REWARD LEARNING WITH EXPLORATION (RLE), along with its theoretical guarantee, and subsequently establish the lower bound for online IRL problems.

### 5.1 ALGORITHM

Building on RLP, we develop REWARD LEARNING WITH EXPLORATION (RLE) to address online IRL problems as specified in Section 3.3. RLE first uses the reward-free exploration strategy (Algoeithm 2) to find a desired behavior policy $\pi^{\mathsf{b}}$ with concentrability with *all* policies, then collects $K$ episodes generated by $\pi^{\mathsf{b}}$, and finally calls to compute an estimated reward mapping. Here, we utilize $NH$ episodes for computing $\pi^{\mathsf{b}}$, where $N$ is specified in Algorithm 2. The total number of episodes of our algorithm is $K + NH$. The full description of RLE is deferred to Section F.

In the following, we present the theoretical guarantees of RLE.

**Theorem 5.1** (Sample complexity of RLE). *Suppose $\pi^{\mathsf{E}}$ is well-posed (Assumption 3.3) when we receive feedback in option* 1. *Then with probability at least* $1 - \delta$, *RLE (Algorithm 6) outputs a reward mapping* $\widehat{\mathscr{R}}$ *such that* $\left[\widehat{\mathscr{R}}(V, A)\right]_h(s, a) \leq [\mathscr{R}^\star(V, A)]_h(s, a)$ *for all* $(V, A) \in \Theta$ *and* $(h, s, a) \in [H] \times \mathcal{S} \times \mathcal{A}$, *and* $D_\Theta^{\mathsf{all}}(\mathscr{R}^\star, \widehat{\mathscr{R}}) \leq 2\epsilon$, *provided that*

$$K \geq \widetilde{\mathcal{O}}\left(\frac{H^4 SA \log \mathcal{N}(\Theta; \epsilon/H)}{\epsilon^2} + \frac{H^2 SA(\eta + \log \mathcal{N}(\Theta; \epsilon/H))}{\epsilon}\right), \quad KH \geq N \geq \widetilde{\mathcal{O}}\left(\sqrt{H^9 S^7 A^7 K}\right),$$

*where* $\eta$ *is specified in Theorem 4.2 and* $\widetilde{\mathcal{O}}$ *hides* $poly \log (H, S, A, 1/\delta)$.

**Relation with the offline setting**     When taking $N = \widetilde{\mathcal{O}}\left(\sqrt{H^9 S^7 A^7 K}\right)$ in Theorem 5.1, the total number of sample episodes $K + NH$ becomes $\widetilde{\mathcal{O}}\left(H^4 SA \log \mathcal{N}(\Theta; \epsilon/H)/\epsilon^2\right)$ (we hide the burn-in term). Comparing this sample complexity with that of our offline algorithm, which achieves $\widetilde{\mathcal{O}}\left(C^\star H^4 S \log \mathcal{N}(\Theta; \epsilon/H)/\epsilon^2\right)$ (we hide the burn-in term), we observe that the sample complexity of Algorithm 6 has one less $C^\star$ factor but one additional $A$ factor. We provide the following explanation for this. As is shown in our interaction protocol, our algorithm finds a desired behavior policy $\pi^{\mathsf{b}}$ which can be considered to have $\widetilde{\mathcal{O}}(A)$ single-policy concentrability[1](Eq.(B.3)) with all policies. Subsequently, the offline algorithm RLP is executed to obtain the estimated reward mapping according to the data collected by following the behavior policy. Therefore, in this case, we can roughly see $C^\star = A$, which yields the sample complexity of Algorithm 6 $\widetilde{\mathcal{O}}\left(H^4 SA \log \mathcal{N}(\Theta; \epsilon/H)/\epsilon^2\right)$.

**Comparison with the bound in Lindner et al. (2023)**     Lindner et al. (2023) achieves a sample complexity of $\mathcal{O}(H^4 SA/\epsilon^2)$ in the special case where $|\Theta| = poly(S, A, H)$, which matches the sample complexity of Algorithm 6. However, the metric proposed in Lindner et al. (2023) has a significant weakness, as demonstrated in Appendix C.3.

**Theoretical lower bound in the offline setting.**     We provide the following lower bound, for the case where $\Theta = \overline{\mathcal{V}} \times \overline{\mathcal{A}}$. This lower bound shows that our rate is sharp when $S \leq A$.

**Theorem 5.2** (Informal version of Theorem G.2). *Suppose* $\Theta = \overline{\mathcal{V}} \times \overline{\mathcal{A}}$, *then any algorithm that returns a reward mapping up to* $\epsilon$ *distance with the ground truth reward mapping for all online IRL problems with probability at least* $2/3$ *has to take at least* $\Omega\left(H^3 SA \min\{S, A\}/\epsilon^2\right)$ *samples.*

## 6   Transfer Learning

As a further extension, we consider the transfer learning setting, where rewards learned in a source MDP\R are transferred to a target MDP\R (possibly different from the source MDP\R). Inspired by the single-policy concentrability assumption, we define two novel concepts: *weak-transferability* (Definition I.2) and *transferability* (Definition I.3) serving as measures to ascertain the level of similarity between two MDP\Rs.

We prove that our algorithms RLP and RLE still are applicable to transfer learning. When the target MDP\R exhibits a low week-transferability (transferability), transfer learning is proven to be implemented with a polynomial sample complexity in terms of the size of the MDP\R and the week-transferability (transferability) coefficient under certain metrics (Theorem I.4 and I.5). We provide guarantees for performing RL algorithms with learned rewards in different environments (Corollary I.6, Corollary I.7). As a new application, we illustrate that our transfer learning approach can be adapted to do transfer learning between two IRL problems sample-efficiently.

## 7   Conclusion

In this paper, we propose a novel algorithm for Inverse Reinforcement Learning (IRL) in the offline setting, using polynomial samples and runtime. Building upon our offline algorithm, we design an algorithm for IRL in the online setting, employing the same complexity as prior work, but under a stronger performance metric. We believe our work opens up many important questions, such as designing better metrics for IRL problems, generalizing our IRL algorithms to the function approximation setting, and developing more computationally efficient algorithms.

---

[1]In this context, the notion of "concentrability" is somewhat distinct from the definition provided in Assumption 4.1. Nevertheless, it still yields a similar outcome, as demonstrated in the proof of Theorem 5.1.

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

## A  ADDITIONAL RELATED WORK

**Imitation learning**  A closely related field to IRL is Imitation Learning, which focuses on learning policies from demonstrations, in contrast to IRL's emphasis on learning rewards from expert demonstrations (Bain & Sammut, 1995; Abbeel & Ng, 2004; Ratliff et al., 2006; Ziebart et al., 2008; Pan et al., 2017; Finn et al., 2016). Imitation learning has been extensively studied in the active setting (Ross et al., 2011; Ross & Bagnell, 2014; Sun et al., 2017), and theoretical analyses for Imitation Learning have been provided by (Rajaraman et al., 2020; Xu et al., 2020a; Chang et al., 2021). More recently, the concept of Representation Learning for Imitation Learning has gained considerable attention (Arora et al., 2020; Nachum & Yang, 2021). While Imitation can be implemented by IRL (Abbeel & Ng, 2004; Ratliff et al., 2006; Ziebart et al., 2008), it is important to note that IRL has wider capabilities than Imitation Learning since the rewards learned through IRL can be transferred across different environments (Levine et al., 2011; Fu et al., 2017).

**Reinforcement learning from human feedback**  Reinforcement Learning from Human Feedback (RLHF) bears a close relation to IRL, particularly because the process of learning rewards is a crucial aspect of both approaches (Zhu et al., 2023a;b; Wang et al., 2023; Zhan et al., 2023). RLHF has been successfully applied in various domains, including robotics (Jain et al., 2013; Sadigh et al., 2017; Ding et al., 2023) and game playing (Ibarz et al., 2018). Recently, RLHF has attracted considerable attention due to its remarkable capability to integrate human knowledge with large language models (Ouyang et al., 2022; OpenAI, 2023). Furthermore, the theoretical foundations of RLHF have been extensively developed in both tabular and function approximation settings (Zhan et al., 2023; Xu et al., 2020b; Pacchiano et al., 2021; Novoseller et al., 2020; Zhu et al., 2023a; Wang et al., 2023).

## B  USEFUL ALGORITHMIC SUBROUTINES FROM PRIOR WORKS

In this section, we give the algorithm procedures of finding behavior policy $\pi^{\mathsf{b}}$ in Algorithm 6. The algorithm procedures are directly quoted from Li et al. (2023), with slight modification.

### B.1  ALGORITHM: FINDING BEHAVIOR POLICY $\pi^{\mathsf{b}}$

Algorithm 2, a component of Li et al. (2023, Algorithm 1), aims to identify a suitable behavior policy. This is achieved by estimating the occupancy distribution $d^{\pi}$, which is induced by any deterministic policy $\pi$, through a meticulously designed exploration strategy. At each stage $h$, Algorithm2 invokes Algorithm procedure 3 to compute an appropriate exploration policy, denoted as $\pi^{\mathsf{explore},h}$, and subsequently collects $N$ sample trajectories by executing $\pi^{\mathsf{explore},h}$. These steps facilitate the estimation of the occupancy distribution $d_{h+1}^{\pi}$ for the next stage, $h + 1$. Finally, the behavior policy $\pi^{\mathsf{b}} \sim \mu_{\mathsf{b}}$ is computed by invoking Algorithm 4.

We highlight that the behavior policy $\pi$ output by Algorithm 2 has following property Li et al. (2023)

$$\sum_{h \in [H]} \sum_{(s,a) \in \mathcal{S} \times \mathcal{A}} \frac{\widehat{d}_h^{\pi}(s,a)}{\mathbb{E}_{\pi' \sim \mu_{\mathsf{b}}}\left[\widehat{d}_h^{\pi'}(s,a)\right]} \lesssim HSA, \tag{B.3}$$

for any deterministic policy $\pi \in \Pi^{\mathsf{det}}$.

### B.2  SUBROUTINE: COMPUTING EXPLORATION POLICY $\pi^{\mathsf{explore},h}$

We proceed to describe Algorithm 3, originally proposed in Li et al. (2023, Algorithm 3), which is designed to compute the desired exploration policy $\pi^{\mathsf{explore},h}$. At a high level, this algorithm calculates the exploration policy by approximately solving the subsequent optimization sub-problem, utilizing the Frank-Wolfe algorithm:

$$\widehat{\mu}^h \approx \arg \max_{\mu \in \Delta(\Pi)} \sum_{(s,a) \in \mathcal{S} \times \mathcal{A}} \log \left[ \frac{1}{KH} + \mathbb{E}_{\pi \sim \mu} \left[ \widehat{d}_h^{\pi}(s,a) \right] \right], \tag{B.4}$$

---

**Algorithm 2** Subroutine for computing behavior policy (Li et al., 2023)

---

1: **Input:** state space $\mathcal{S}$, action space $\mathcal{A}$, horizon length $H$, initial state distribution $\rho$, target success probability $1 - \delta$, threshold $\xi = c_\xi H^3 S^3 A^3 \log(HSA/\delta)$.

2: Draw $N$ i.i.d. initial states $s_1^{n,0} \overset{\text{i.i.d.}}{\sim} \rho\ (1 \le n \le N)$, and define the following functions

$$\widehat{d}_1^\pi(s) = \frac{1}{N} \sum_{n=1}^N \mathbb{1}\{s_1^{n,0} = s\}, \qquad \widehat{d}_1^\pi(s,a) = \widehat{d}_1^\pi(s) \cdot \pi_1(a|s) \tag{B.1}$$

for any deterministic policy $\pi : [H] \times \mathcal{S} \to \Delta(\mathcal{A})$ and any $(s,a) \in \mathcal{S} \times \mathcal{A}$.

3: **for** $h = 1, ..., H - 1$ **do**

4:     Call Algorithm 3 to compute an exploration policy $\pi^{\text{explore},h}$.

5:     Draw $N$ independent trajectories $\{s_1^{n,h}, a_1^{n,h}, \ldots, s_{h+1}^{n,h}\}_{1 \le n \le N}$ using policy $\pi^{\text{explore},h}$ and compute

$$\widehat{\mathbb{P}}_h(s'|s,a) = \frac{\mathbf{1}\{N_h(s,a) > \xi\}}{\max\{N_h(s,a), 1\}} \sum_{n=1}^N \mathbf{1}\left\{s_h^{n,h} = s, a_h^{n,h} = a, s_{h+1}^{n,h} = s'\right\}, \qquad \forall(s,a,s') \in \mathcal{S} \times \mathcal{A} \times \mathcal{S},$$

where $N_h(s,a) = \sum_{n=1}^N \mathbf{1}\left\{s_h^{n,h} = s, a_h^{n,h} = a\right\}$.

6:     For any deterministic policy $\pi : \mathcal{S} \times [H] \to \Delta(\mathcal{A})$ and any $(s,a) \in \mathcal{S} \times \mathcal{A}$, define
$$\widehat{d}_{h+1}^\pi(s) = \langle \widehat{\mathbb{P}}_h(s|\cdot,\cdot), \widehat{d}_h^\pi(\cdot,\cdot)\rangle, \qquad \widehat{d}_{h+1}^\pi(s,a) = \widehat{d}_{h+1}^\pi(s) \cdot \pi_{h+1}(a|s). \tag{B.2}$$

   .

7: Call Algorithm 4 to compute a behavior policy $\pi^{\mathsf{b}}$.

8: **Output:** the behavior policy $\pi^{\mathsf{b}}$.

---

**Algorithm 3** Subroutine for solving Eq.(B.4) (Li et al., 2023).

---

1: **Initialize:** $\mu^{(0)} = \delta_{\pi_{\text{init}}}$ for an arbitrary policy $\pi_{\text{init}} \in \Pi$, $T_{\max} = \lfloor 50SA \log(KH) \rfloor$.

2: **for** $t = 0, 1..., T_{\max}$ **do**

3:     Compute the optimal deterministic policy $\pi^{(t),\mathsf{b}}$ of the MDP $\mathcal{M}_{\mathsf{b}}^h = (\mathcal{S} \cup \{s_{\text{aug}}\}, \mathcal{A}, H, \widehat{\mathbb{P}}^{\text{aug},h}, r_{\mathsf{b}}^h)$, where $r_{\mathsf{b}}^h$ is defined in Eq.(B.5), and $\widehat{\mathbb{P}}^{\text{aug},h}$ is defined in Eq.(B.6); let $\pi^{(t)}$ be the corresponding optimal deterministic policy of $\pi^{(t),\mathsf{b}}$ in the original state space.

4:     Compute
$$\alpha_t = \frac{\frac{1}{SA} g(\pi^{(t)}, \widehat{d}, \mu^{(t)}) - 1}{g(\pi^{(t)}, \widehat{d}, \mu^{(t)}) - 1}, \quad \text{where} \quad g(\pi, \widehat{d}, \mu) = \sum_{(s,a) \in \mathcal{S} \times \mathcal{A}} \frac{\frac{1}{KH} + \widehat{d}_h^\pi(s,a)}{\frac{1}{KH} + \mathbb{E}_{\pi \sim \mu}[\widehat{d}_h^\pi(s,a)]}.$$
   Here, $\widehat{d}_h^\pi(s,a)$ is computed via Eq.(B.1) for $h = 1$, and Eq.(B.2) for $h \ge 2$.

5:     If $g(\pi^{(t)}, \widehat{d}, \mu^{(t)}) \le 2SA$ then exit for-loop.

6:     Update
$$\mu^{(t+1)} = (1 - \alpha_t)\, \mu^{(t)} + \alpha_t \mathbb{1}_{\pi^{(t)}}.$$

7: **Output**: the exploration policy $\pi^{\text{explore},h} = \mathbb{E}_{\pi \sim \mu^{(t)}}[\pi]$ and the weight $\widehat{\mu}^h = \mu^{(t)}$.

---

Here $\mathcal{M}_{\mathsf{b}}^h = (\mathcal{S} \cup \{s^{\text{aug}}\}, \mathcal{A}, H, \widehat{\mathbb{P}}^{\text{aug},h}, r_{\mathsf{b}}^h)$, where $s_{\text{aug}}$ is an augmented state as before, and the reward function is chosen to be

$$r_{\mathsf{b},j}^h(s,a) = \begin{cases} \frac{1}{\frac{1}{KH} + \mathbb{E}_{\pi \sim \mu^{(t)}}\left[\widehat{d}_h^\pi(s,a)\right]} \in [0, KH], & \text{if } (s,a,j) \in \mathcal{S} \times \mathcal{A} \times \{h\}; \\ 0, & \text{if } s = s_{\text{aug}} \text{ or } j \ne h. \end{cases} \tag{B.5}$$

In addition, the augmented probability transition kernel $\widehat{\mathbb{P}}^{\text{aug},h}$ is constructed based on $\widehat{\mathbb{P}}$ as follows:

$$\widehat{\mathbb{P}}_j^{\text{aug},h}(s'|s,a) = \begin{cases} \widehat{\mathbb{P}}_j(s'|s,a), & \text{if } s' \in \mathcal{S} \\ 1 - \sum_{s' \in \mathcal{S}} \widehat{\mathbb{P}}_j(s'|s,a), & \text{if } s' = s_{\text{aug}} \end{cases} \quad \text{for all } (s,a,j) \in \mathcal{S} \times \mathcal{A} \times [h]; \tag{B.6a}$$

$$\widehat{\mathbb{P}}_j^{\text{aug},h}(s'|s,a) = \mathbb{1}(s' = s_{\text{aug}}) \qquad\qquad\qquad\quad \text{if } s = s_{\text{aug}} \text{ or } j > h. \tag{B.6b}$$

### B.3 SUBROUTINE: COMPUTING FINAL BEHAVIOR POLICY $\pi^{\mathsf{b}}$

We proceed to describe Algorithm 4, originally proposed in (Li et al., 2023, Algorithm 2), which is designed to compute the final behavior policy $\pi^{\mathsf{b}}$ $\pi^{\mathsf{explore},h}$, based on the estimated occupancy distributions specified in Algorithm 2. Algorithm 4 follows a similar fashion of Algorithm 3. Algorithm 4 computes the behavior policy by approximately solving the subsequent optimization sub-problem, utilizing the Frank-Wolfe algorithm:

$$\widehat{\mu}^{\mathsf{b}} \approx \arg \max_{\mu \in \Delta(\Pi)} \left\{ \sum_{h=1}^{H} \sum_{(s,a) \in \mathcal{S} \times \mathcal{A}} \log \left[ \frac{1}{KH} + \mathbb{E}_{\pi \sim \mu} \left[ \widehat{d}_h^\pi(s,a) \right] \right] \right\}. \tag{B.7}$$

---

**Algorithm 4** Subroutine for solving Eq.(B.7) (Li et al., 2023).

---

1: **Initialize:** $\mu_{\mathsf{b}}^{(0)} = \delta_{\pi_{\mathsf{init}}}$ for an arbitrary policy $\pi_{\mathsf{init}} \in \Pi$, $T_{\max} = \lfloor 50SAH \log(KH) \rfloor$.
2: **for** $t = 0, 1..., T_{\max}$ **do**
3:     Compute the optimal deterministic policy $\pi^{(t),\mathsf{b}}$ of the MDP $\mathcal{M}_{\mathsf{b}} = (\mathcal{S} \cup \{s_{\mathsf{aug}}\}, \mathcal{A}, H, \widehat{\mathbb{P}}^{\mathsf{aug}}, r_{\mathsf{b}})$, where $r_{\mathsf{b}}$ is defined in Eq.(B.8), and $\widehat{\mathbb{P}}^{\mathsf{aug}}$ is defined in Eq.(B.9); let $\pi^{(t)}$ be the corresponding optimal deterministic policy of $\pi^{(t),\mathsf{b}}$ in the original state space.
4:     Compute

$$\alpha_t = \frac{\frac{1}{SAH} g(\pi^{(t)}, \widehat{d}, \mu_{\mathsf{b}}^{(t)}) - 1}{g(\pi^{(t)}, \widehat{d}, \mu_{\mathsf{b}}^{(t)}) - 1}, \quad \text{where} \quad g(\pi, \widehat{d}, \mu) = \sum_{h=1}^{H} \sum_{(s,a) \in \mathcal{S} \times \mathcal{A}} \frac{\frac{1}{KH} + \widehat{d}_h^\pi(s,a)}{\frac{1}{KH} + \mathbb{E}_{\pi \sim \mu} \left[ \widehat{d}_h^\pi(s,a) \right]}.$$

    Here, $\widehat{d}_h^\pi(s,a)$ is computed via Eq.(B.1) for $h = 1$, and Eq.(B.2) for $h \geq 2$.
5:     If $g(\pi^{(t)}, \widehat{d}, \mu_{\mathsf{b}}^{(t)}) \leq 2HSA$ then exit for-loop. Update

$$\mu_{\mathsf{b}}^{(t+1)} = (1 - \alpha_t) \mu_{\mathsf{b}}^{(t)} + \alpha_t \mathbf{1}_{\pi^{(t)}}.$$

6: **Output:** the behavior policy $\pi^{\mathsf{b}} = \mathbb{E}_{\pi \sim \mu_{\mathsf{b}}^{(t)}}[\pi]$ and the associated weight $\widehat{\mu}_b = \mu_{\mathsf{b}}^{(t)}$.

---

Here, $\mathcal{M}_{\mathsf{b}} = (\mathcal{S} \cup \{s_{\mathsf{aug}}\}, \mathcal{A}, H, \widehat{\mathbb{P}}^{\mathsf{aug}}, r_{\mathsf{b}})$, where $s_{\mathsf{aug}}$ is an augmented state and the reward function is chosen to be

$$r_{\mathsf{b},h}(s,a) = \begin{cases} \frac{1}{\frac{1}{KH} + \mathbb{E}_{\pi \sim \mu_{\mathsf{b}}^{(t)}} \left[ \widehat{d}_h^\pi(s,a) \right]} \in [0, KH], & \text{if } (s,a,h) \in \mathcal{S} \times \mathcal{A} \times [H]; \\ 0, & \text{if } (s,a,h) \in \{s_{\mathsf{aug}}\} \times \mathcal{A} \times [H]. \end{cases} \tag{B.8}$$

In addition, the augmented probability transition kernel $\widehat{\mathbb{P}}^{\mathsf{aug}}$ is constructed based on $\widehat{\mathbb{P}}$ as follows:

$$\widehat{\mathbb{P}}_h^{\mathsf{aug}}(s' \mid s,a) = \begin{cases} \widehat{\mathbb{P}}_h(s' \mid s,a), & \text{if } s' \in \mathcal{S} \\ 1 - \sum_{s' \in \mathcal{S}} \widehat{\mathbb{P}}_h(s' \mid s,a), & \text{if } s' = s_{\mathsf{aug}} \end{cases} \quad \text{for all } (s,a,h) \in \mathcal{S} \times \mathcal{A} \times [H]; \tag{B.9a}$$

$$\widehat{\mathbb{P}}_h^{\mathsf{aug}}(s' \mid s_{\mathsf{aug}}, a) = \mathbb{1}(s' = s_{\mathsf{aug}}) \qquad \text{for all } (a,h) \in \mathcal{A} \times [H]. \tag{B.9b}$$

It's evident that the augmented state behaves as an absorbing state, associated with zero immediate rewards.

## C RELATIONSHIP WITH EXISTING METRICS

### C.1 METRICS BETWEEN REWARDS

Here we will mainly discuss the relationship between our metric defined in Section 3.1 and metrics in the existing works (Metelli et al., 2021; 2023; Lindner et al., 2023). Metelli et al. (2023) consider the

following metric[2]

$$d_{V^\star}^G(r, \widehat{r}) = \max_{\widehat{\pi} \in \Pi^\star_{\mathcal{M} \cup \widehat{r}}} \max_{(h,s) \in [H] \times \mathcal{S}} \left| V_h^\star(s; r) - V_h^{\widehat{\pi}}(s; r) \right|. \tag{C.1}$$

We remark that $d_{V^\star}^G(r, \widehat{r})$ is utilized in the simulator setting (Metelli et al., 2023) and is not directly comparable with our metrics since $d_{V^\star}^G$ takes maximum over *all* $(h, s)$. However, we can compare the following aspect: Metelli et al. (2023) considers $V^\star - V^\pi$, whereas we consider $V^\pi$ on $r$ vs $\widehat{r}$. It can be shown that our method that considers $V^\pi$ on $r$ vs $\widehat{r}$ subsumes theirs when we consider optimal policies.

**Lemma C.1.** *For any $\mathcal{M}$, $(r, \widehat{r})$ and $(h, s) \in [H] \times \mathcal{S}$, we have*

$$\sup_{\pi \in \Pi^\star_{\mathcal{M} \cup r} \cup \Pi^\star_{\mathcal{M} \cup \widehat{r}}} |V_h^\star(s; r) - V_h^\pi(s; r)| \leq 2 \sup_{\pi \in \Pi^\star_{\mathcal{M} \cup r} \cup \Pi^\star_{\mathcal{M} \cup \widehat{r}}} |V_h^\pi(s; r) - V_h^\pi(s; \widehat{r})|.$$

*Conversely, there exist a $\mathcal{M}$ with $S = A = 2$, $H = 2$, $(h, s, a) \in [H] \times \mathcal{S} \times \mathcal{A}$ and a pair $(r, \widehat{r})$ such that $\sup_{\pi \in \Pi^\star_{\mathcal{M} \cup r} \cup \Pi^\star_{\mathcal{M} \cup \widehat{r}}} |V_h^\star(s; r) - V_h^\pi(s; r)| = 0$, but $\sup_{\pi \in \Pi^\star_{\mathcal{M} \cup r} \cup \Pi^\star_{\mathcal{M} \cup \widehat{r}}} |V_h^\pi(s; r) - V_h^\pi(s; \widehat{r})| \geq 1$.*

## C.2 REWARD AGGREGATION METHODS

Since we have defined the metric between rewards and characterized the relationship with metrics in Metelli et al. (2023). This naturally leads to the following question: given a metric for rewards, how to measure IRL algorithms? Our method is to use the reward mapping metric induced by the metric for rewards. But the metric for IRL algorithms used in Metelli et al. (2023) is a bit different from ours. Before presenting the performance metric for IRL algorithms used in Metelli et al. (2023), we first define the feasible reward set. Given a reward mapping $\mathscr{R} : \overline{\mathcal{V}} \times \overline{\mathcal{A}} \mapsto \mathcal{R}^{\mathsf{all}}$, we say a reward set $\mathcal{R} \subset \mathcal{R}^{\mathsf{all}}$ is a feasible reward set induced by $\mathscr{R}$, if $\mathcal{R} = \mathrm{image}(\mathscr{R})$. They used the Hausdorff (pre)metric $D^{\mathsf{H}}(\mathcal{R}, \widehat{\mathcal{R}})$ between the exact feasible reward set $\mathcal{R}$ and the estimated feasible set $\widehat{\mathcal{R}} \subset \mathcal{R}^{\mathsf{all}}$ which is defined by

$$D^{\mathsf{H}}(\mathcal{R}, \widehat{\mathcal{R}}) := \max \Big\{ \sup_{r \in \mathcal{R}} \inf_{\widehat{r} \in \widehat{\mathcal{R}}} d(r, \widehat{r}), \sup_{\widehat{r} \in \widehat{\mathcal{R}}} \inf_{r \in \mathcal{R}} d(r, \widehat{r}) \Big\},$$

where $d$ is a given metric between rewards. Their learning goal is to find an estimated feasible set $\widehat{\mathcal{R}}$, attaining a small $D^{\mathsf{H}}(\mathcal{R}, \widehat{\mathcal{R}})$. Different from our mapping-based metric, the Hausdorff metric measures only the gap between the exact feasible set $\mathcal{R}$ and the estimated feasible set $\widehat{\mathcal{R}}$, but can't measure the gap between rewards for every parameter $(V, A)$. In fact, given a metric between rewards, our mapping-based metric is stronger than the Hausdorff metric.

**Lemma C.2** ($D^{\mathsf{M}}$ is stronger than $D^{\mathsf{H}}$). *Given an IRL problem $(\mathcal{M}, \pi^E)$ and a (pre)metric $d : \mathcal{R}^{\mathsf{all}} \times \mathcal{R}^{\mathsf{all}} \mapsto \mathbb{R}_{\geq 0}$ between rewards. We define the corresponding Hausdorff metric $D^{\mathsf{H}}$ for any pair $(\mathcal{R}, \mathcal{R}')$ by*

$$D^{\mathsf{H}}(\mathcal{R}, \mathcal{R}') := \max \left\{ \sup_{r \in \mathcal{R}} \inf_{r' \in \mathcal{R}'} d(r, r'), \sup_{r' \in \mathcal{R}'} \inf_{r \in \mathcal{R}} d(r, r') \right\},$$

*and the mapping-based metric is defined for any pair $(\mathscr{R}, \mathscr{R}')$ by*

$$D^{\mathsf{M}}(\mathscr{R}, \mathscr{R}') := \sup_{V \in \overline{\mathcal{V}}, A \in \overline{\mathcal{A}}} d(\mathscr{R}(V, A), \mathscr{R}'(V, A)),$$

*Let $\mathcal{R}$ and $\mathcal{R}'$ are feasible reward set induced by $\mathscr{R}$ and $\mathscr{R}'$, then we have*

$$D^{\mathsf{H}}(\mathcal{R}, \mathcal{R}') \leq D^{\mathsf{M}}(\mathscr{R}, \mathscr{R}').$$

We then present the following lemma which demonstrates that for some $d$, $D^{\mathsf{M}}$ is *strictly* stronger than $D^{\mathsf{H}}$.

---

[2]Technically, the metric defined in Metelli et al. (2023) considers the value function difference overall $(h, s) \in [H] \times \mathcal{S}$, which is more suitable for the generative model setting, and clearly cannot be minimized in the online setting in general, due to the potential existence of unreachable states; We have adapted their metric to look at the initial value function only, which is more suitable for the online setting.

**Lemma C.3.** *There exists a metric $d$ which is a metric between rewards such that for any IRL problem $(\mathcal{M}, \pi^{\mathsf{E}})$, there exists another IRL problem $(\widehat{\mathcal{M}}, \widehat{\pi}^{\mathsf{E}})$ such that $D^{\mathsf{H}}(\mathcal{R}^{\star}, \widehat{\mathcal{R}}) = 0$, but $D^{\mathsf{M}}(\mathscr{R}^{\star}, \widehat{\mathscr{R}}) \geq 1/2$, where $D^{\mathsf{H}}$ and $D^{\mathsf{M}}$ are the Hausdorff metric and mapping-based metric induced by $d$, respectively; $\mathcal{R}^{\star}$ and $\widehat{\mathcal{R}}$ are the feasible sets of $(\mathcal{M}, \pi^{\mathsf{E}})$ and $(\widehat{\mathcal{M}}, \widehat{\pi}^{\mathsf{E}})$, respectively; $\mathscr{R}^{\star}$ and $\widehat{\mathscr{R}}$ are the reward mappings induced by $(\mathcal{M}, \pi^{\mathsf{E}})$ and $(\widehat{\mathcal{M}}, \widehat{\pi}^{\mathsf{E}})$, respectively.*

## C.3 Disscussion of existing metric for online IRL

We also consider the performance metric proposed in Lindner et al. (2023). They consider the metric between IRL problems $\tau = (\mathcal{M}, \pi^{\mathsf{E}})$ and $\widehat{\tau} = (\widehat{\mathcal{M}}, \widehat{\pi}^{\mathsf{E}})$, where $(\widehat{\mathcal{M}}, \widehat{\pi}^{\mathsf{E}})$ is the recovered IRL problem. We then present the performance metric of Lindner et al. (2023):

$$D^{L}(\tau, \widehat{\tau}) := \max\left\{ \sup_{r \in \mathcal{R}_{\tau}} \inf_{\widehat{r} \in \mathcal{R}_{\widehat{\tau}}} \sup_{\widehat{\pi}^{\star} \in \Pi^{\star}_{\widehat{\mathcal{M}} \cup \widehat{r}}} \max_{a} \left| Q_1^{\pi^{\star}}(s_1, a; \mathcal{M} \cup r) - Q_1^{\widehat{\pi}^{\star}}(s_1, a; \mathcal{M} \cup r) \right|, \quad \text{(C.2)} \right.$$

$$\left. \sup_{\widehat{r} \in \mathcal{R}_{\widehat{\tau}}} \inf_{r \in \mathcal{R}_{\tau}} \sup_{\pi^{\star} \in \Pi^{\star}_{\mathcal{M} \cup r}} \max_{a} \left| Q_1^{\pi^{\star}}(s_1, a; \mathcal{M} \cup r) - Q_1^{\widehat{\pi}^{\star}}(s_1, a; \mathcal{M} \cup r) \right| \right\}, \quad \text{(C.3)}$$

where the $\mathcal{R}_{\tau}, \mathcal{R}_{\widehat{\tau}}$ the set of all feasible rewards set for IRL problems $\tau, \widehat{\tau}$, respectively, $\pi^{\star} \in \Pi^{\star}_{\mathcal{M} \cup r}, \widehat{\pi}^{\star} \in \Pi^{\star}_{\widehat{\mathcal{M}} \cup \widehat{r}}$, and $Q_1^{\pi}(\cdot | \mathcal{M} \cup r)$ represent the $Q$-function induced by $\mathcal{M} \cup r$. The following proposition demonstrates that if $\pi^{\mathsf{E}} = \widehat{\pi}^{\mathsf{E}}$, then $D^{L}(\tau, \widehat{\tau}) = 0$, indicating that $D$ solely measures the closeness between $\pi^{\mathsf{E}}$ and $\widehat{\pi}^{\mathsf{E}}$ without capturing the closeness between $\mathcal{M}$ and $\widehat{\mathcal{M}}$. However, in contrast to imitation learning, where the recovery of the expert policy suffices, IRL necessitates the retrieval of transition dynamics. This becomes especially crucial when leveraging learned rewards for transfer learning. Merely recovering the expert policy is insufficient for performing RL algorithms with learned rewards in a different environment.

**Proposition C.4.** *For any $\tau = (\mathcal{M}, \pi^{\mathsf{E}})$ and $\widehat{\tau} = (\widehat{\mathcal{M}}, \widehat{\pi}^{\mathsf{E}})$, if $\pi^{\mathsf{E}} = \widehat{\pi}^{\mathsf{E}}$, then $D^{L}(\tau, \widehat{\tau}) = 0$.*

## C.4 Guarantee for performing RL algorithm with learned rewards

Significantly, the metric $d^{\mathsf{all}}$ possesses a substantial advantage: a policy that is near-optimal under a certain reward remains near-optimal under a reward that is close to the original one as per $d^{\mathsf{all}}$. This is a crucial feature that was lacking in the previously established metrics. Specifically, the metrics presented in the works of Metelli et al. (2023); Lindner et al. (2023) only took into consideration the optimal policy, without providing for the evaluation of near-optimal policies.

**Proposition C.5** ($d^{\mathsf{all}}$ suffices for recovering $\pi^{\mathsf{E}}$). *Given an MDP\R $\mathcal{M}$, let $(r, \widehat{r})$ be a pair of rewards such that $d^{\mathsf{all}}(r, \widehat{r}) \leq \epsilon$. Suppose that $\widehat{\pi}$ is a $\epsilon'$-optimal policy in the MDP $\mathcal{M} \cup \widehat{r}$, i.e, $V_1^{\star}(s_1; \widehat{r}) - V_1^{\widehat{\pi}}(s_1; \widehat{r}) \leq \epsilon'$, then*

$$V_1^{\star}(s_1; r) - V_1^{\widehat{\pi}}(s_1; r) \leq 3\epsilon + \epsilon'.$$

Proposition C.5 provides a guarantee for performing standard RL algorithms aiming to find near-optimal policies with learned rewards (having a small $d^{\mathsf{all}}$ error).

**Proposition C.6.** *Given an MDP\R $\mathcal{M}$, let $(r, \widehat{r})$ be a pair of reward such that $d^{\pi}(r, \widehat{r}) \leq \epsilon$, where $\pi$ is an $\bar{\epsilon}$-optimal policy in the MDP $\mathcal{M} \cup r$. Suppose that $\widehat{r}_h(s, a) \leq r_h(s, a)$ for any $(h, s, a) \in [H] \times \mathcal{S} \times \mathcal{A}$. Let $\widehat{\pi}$ be an $\epsilon'$-optimal policy in the MDP $\mathcal{M} \cup \widehat{r}$, i.e, $V_1^{\star}(s_1; \widehat{r}) - V_1^{\widehat{\pi}}(s_1; \widehat{r}) \leq \epsilon'$, then we have*

$$V_1^{\star}(s_1; r) - V_1^{\widehat{\pi}}(s_1; r) \leq \epsilon + \epsilon' + 2\bar{\epsilon}.$$

Proposition C.6 illustrates that when the monotonicity condition: $\widehat{r}_h(s, a) \leq r_h(s, a)$ is satisfied, the performance of performing RL algorithms with learned rewards (having small $d^{\pi}$ error) can be guaranteed. This proposition is applicable to our offline IRL algorithm, and in the algorithm design, we employ pessimism techniques to ensure that the recovered rewards satisfy the monotonicity condition. Later, we will observe that Proposition C.6 is applicable to our offline and online IRL algorithms. It is important to note that Proposition C.6 also provides a guarantee for performing standard RL algorithms with learned rewards (with a small $d^{\pi}$ error) in *different* environments (see Corollary I.6, Corollary I.7).

## C.5 Proofs for Section C

*proof of Lemma C.1.* When $\pi \in \Pi^\star_{\mathcal{M} \cup r}$, we have

$$|V_h^\star(s; r) - V_h^\pi(s; r)| = |V_h^\star(s; r) - V_h^\star(s; r)| = 0 \leq \sup_{\pi \in \Pi^\star_{\mathcal{M} \cup r} \cup \Pi^\star_{\mathcal{M} \cup \widehat{r}}} |V_h^\pi(s; r) - V_h^\pi(s; \widehat{r})|.$$

$$(C.4)$$

Let $\pi^\star \in \Pi^\star_{\mathcal{M} \cup r}$, and then we directly have

$$V^{\pi^\star}(s; r) = V^\star(s; r).$$

When $\pi \in \Pi^\star_{\mathcal{M} \cup \widehat{r}}$, we have

$$\begin{aligned}
0 \leq V_h^\star(s; r) - V_h^\pi(s; r) &\leq V_h^{\pi^\star}(s; r) - V_h^\pi(s; r) + V_h^\pi(s; \widehat{r}) - V_h^{\pi^\star}(s; \widehat{r}) \\
&= V_h^{\pi^\star}(s; r) - V_h^{\pi^\star}(s; \widehat{r}) + V_h^\pi(s; \widehat{r}) - V_h^\pi(s; r) \\
&\leq \left| V_h^{\pi^\star}(s; r) - V_h^{\pi^\star}(s; \widehat{r}) \right| + |V_h^\pi(s; \widehat{r}) - V_h^\pi(s; r)| \\
&\leq 2 \sup_{\pi \in \Pi^\star_{\mathcal{M} \cup r} \cup \Pi^\star_{\mathcal{M} \cup \widehat{r}}} |V_h^\pi(s; r) - V_h^\pi(s; \widehat{r})|.
\end{aligned}$$

$$(C.5)$$

where the first line comes from $\pi \in \Pi^\star_{\mathcal{M} \cup \widehat{r}}$: $V_h^\pi(s; \widehat{r}) - V_h^{\pi^\star}(s; \widehat{r}) \geq 0$ and the second line uses triangle inequality. Combining Eq.(C.4) and Eq.(C.5), we complete the proof. For the second part of Lemma C.1, we consider an MDP\R $\mathcal{M}$ with $H = 1, S = 2$ and $A = 2$, which is specified by $\mathcal{S} = \{s_a, s_b\}, \mathcal{A} = a_g, a_b$. The transition dynamics are given by

$$\mathbb{P}_1(s_a | s_a, a_g) = 1, \qquad \mathbb{P}_1(s_a | s_a, a_b) = \mathbb{P}_1(s_b | s_a, a_b) = 1/2, \text{ and } \mathbb{P}(s_1 = s_g) = 1,$$

which implies that the initial state is $s_g$. We then construct rewards $r$ and $\widehat{r}$ as follows:

$$\begin{aligned}
r_1(s_a, a_g) &= 1, & r_1(s_a, a_b) &= 1, \\
\widehat{r}_1(s_a, a_g) &= 1, & \widehat{r}_1(s_a, a_b) &= 0.
\end{aligned}$$

By the construction of $r$, for any policy $\pi$, we have $\pi \in \Pi^\star_{\mathcal{M} \cup r}$, which means $\Pi^\star_{\mathcal{M} \cup \widehat{r}} \Pi^\star_{\mathcal{M} \cup r}$. Hence, we obtain that

$$\sup_{\pi \in \Pi^\star_{\mathcal{M} \cup r} \cup \Pi^\star_{\mathcal{M} \cup \widehat{r}}} |V_1^\star(s_a; r) - V_1^\pi(s_a; r)| = \sup_{\pi \in \Pi^\star_{\mathcal{M} \cup r} \cup \Pi^\star_{\mathcal{M} \cup \widehat{r}}} |V_1^\star(s_a; r) - V_1^\star(s_a; r)| = 0. \quad (C.6)$$

On the other hand, we consider the policy $\pi^{\text{bad}}$ which is given by $\pi_1^{\text{bad}}(a_b | s_a) = 1$. Then, we have

$$\sup_{\pi \in \Pi^\star_{\mathcal{M} \cup r} \cup \Pi^\star_{\mathcal{M} \cup \widehat{r}}} |V_h^\pi(s; r) - V_h^\pi(s; \widehat{r})| \geq |V_h^{\pi^{\text{bad}}}(s; r) - V_h^{\pi^{\text{bad}}}(s; \widehat{r})| = |1 - 0| = 1. \quad (C.7)$$

$$\square$$

*Proof of Lemma C.2.* Since $\mathcal{R}$ and $\mathcal{R}'$ are induced by $\mathscr{R}$ and $\mathscr{R}'$, then for any $r \in \mathcal{R}$ and $r' \in \mathcal{R}$, there exist $V, V' \in \overline{\mathcal{V}}, A, A \in \overline{\mathcal{A}}$ such that

$$r = \mathscr{R}(V, A), \qquad r' = \mathscr{R}'(V', A').$$

Then, we have

$$\begin{aligned}
\sup_{r \in \mathcal{R}} \inf_{r' \in \mathcal{R}'} d(r, r') &= \sup_{V \in \overline{\mathcal{V}}, A \in \overline{\mathcal{A}}} \inf_{V' \in \overline{\mathcal{V}}, A' \in \overline{\mathcal{A}}} d(\mathscr{R}(V, A), \mathscr{R}'(V', A')) \\
&\leq \sup_{V \in \overline{\mathcal{V}}, A \in \overline{\mathcal{A}}} d(\mathscr{R}(V, A), \mathscr{R}'(V, A)) = D^{\mathsf{M}}(\mathscr{R}, \mathscr{R}').
\end{aligned}$$

Similarly, we obtain

$$\sup_{r' \in \mathcal{R}'} \inf_{r \in \mathcal{R}} d(r, r') \leq D^{\mathsf{M}}(\mathscr{R}, \mathscr{R}').$$

Hence, we conclude that

$$D^{\mathsf{H}}(\mathcal{R}, \mathcal{R}') \leq D^{\mathsf{M}}(\mathscr{R}, \mathscr{R}').$$

$$\square$$

*Proof of Lemma C.3.* Fix a $(\bar{s}, \bar{a}) \in \mathcal{S} \times \mathcal{A}$, we define metric $d$ by

$$d(r, r') := |r_1(\bar{s}, \bar{a}) - r'_1(\bar{s}, \bar{a})|. \tag{C.8}$$

Give an IRL problem $(\mathcal{M}, \pi^{\mathsf{E}})$, let $\mathbb{P}$ be the transition dynamics of $(\mathcal{M}, \pi^{\mathsf{E}})$. Let $s^{\star} := \arg\min_{s \in \mathcal{S}} \mathbb{P}_1(s|\bar{s}, \bar{a})$. By the Pigeonhole Principle, we have $\mathbb{P}_1(s^{\star}|\bar{s}, \bar{a}) \leq 1/S \leq 1/2$. We construct transition $\mathbb{P}'$ by

$$\mathbb{P}'_1(s^{\star}|\bar{s}, \bar{a}) = 1. \tag{C.9}$$

Let $\widehat{\mathcal{M}} = (\mathcal{S}, \mathcal{A}, H, \mathbb{P}')$, $\widehat{\pi}^{\mathsf{E}} = \pi^{\mathsf{E}}$, and $\widehat{\mathscr{R}}$ be the reward mapping induced by $(\widehat{\mathcal{M}}, \widehat{\pi}^{\mathsf{E}})$. For any $(V, A) \in \bar{\mathcal{V}} \times \bar{\mathcal{A}}$, we define $(V', A') \in \bar{\mathcal{V}} \times \bar{\mathcal{A}}$ by

$$\begin{cases} V_2(s^{\star}) = [\mathbb{P}_1 V_2](\bar{s}, \bar{a}) \\ V_h(s) = V_h(s) \qquad (h, s, a) \neq (1, \bar{s}, \bar{a}), \end{cases} \qquad A' = A. \tag{C.10}$$

Then we have

$$d\Big(\mathscr{R}(V, A), \widehat{\mathscr{R}}(V', A')\Big) = |[\mathscr{R}(V, A)]_1(\bar{s}, \bar{a}) - [\mathscr{R}(V, A)]_1(\bar{s}, \bar{a})| \tag{C.11}$$

$$= \Big| - A_1(\bar{s}, \bar{a}) \cdot \mathbf{1}\big\{\bar{a} \in \text{supp}\big(\pi_1^{\mathsf{E}}(\cdot|\bar{s})\big)\big\} + V_1(s) - [\mathbb{P}_1 V_2](\bar{s}, \bar{a})$$

$$- \big\{-A'_1(\bar{s}, \bar{a}) \cdot \mathbf{1}\big\{\bar{a} \in \text{supp}\big(\widehat{\pi}_1^{\mathsf{E}}(\cdot|\bar{s})\big)\big\} + V'_1(s) - [\mathbb{P}'_1 V'_2](\bar{s}, \bar{a})\big\}\Big|$$

$$= |[\mathbb{P}'_1 V'_2](\bar{s}, \bar{a}) - [\mathbb{P}_1 V_2](\bar{s}, \bar{a})|$$

$$= |V_2(s^{\star}) - [\mathbb{P}_1 V_2](\bar{s}, \bar{a})| = 0 \tag{C.12}$$

On the other hand, for any $(V', A') \in \bar{\mathcal{V}} \times \bar{\mathcal{A}}$, we define $(V, A) \in \bar{\mathcal{V}} \times \bar{\mathcal{A}}$ by

$$\begin{cases} V_2(s) = V'_2(s^{\star}), & s \in \mathcal{S}, \\ V_h(s) = V'_h(s) & h \neq 2, \end{cases} \tag{C.13}$$

which implies that

$$[\mathbb{P}_1 V_2](\bar{s}, \bar{a}) = V'_2(s^{\star}) = [\mathbb{P}'_1 V'_2](\bar{s}, \bar{a}). \tag{C.14}$$

Hence, we have

$$d\Big(\mathscr{R}(V, A), \widehat{\mathscr{R}}(V', A')\Big) = |[\mathscr{R}(V, A)]_1(\bar{s}, \bar{a}) - [\mathscr{R}(V, A)]_1(\bar{s}, \bar{a})| \tag{C.15}$$

$$= \Big| - A_1(\bar{s}, \bar{a}) \cdot \mathbf{1}\big\{\bar{a} \in \text{supp}\big(\pi_1^{\mathsf{E}}(\cdot|\bar{s})\big)\big\} + V_1(s) - [\mathbb{P}_1 V_2](\bar{s}, \bar{a})$$

$$- \big\{-A'_1(\bar{s}, \bar{a}) \cdot \mathbf{1}\big\{\bar{a} \in \text{supp}\big(\widehat{\pi}_1^{\mathsf{E}}(\cdot|\bar{s})\big)\big\} + V'_1(s) - [\mathbb{P}'_1 V'_2](\bar{s}, \bar{a})\big\}\Big|$$

$$= |[\mathbb{P}'_1 V'_2](\bar{s}, \bar{a}) - [\mathbb{P}_1 V_2](\bar{s}, \bar{a})| = 0 \tag{C.16}$$

Combining Eq.(C.11) and Eq.(C.15), we have $D^{\mathsf{H}}(\mathcal{R}, \widehat{\mathcal{R}}) = 0$.

On the other hand, we define $(\widetilde{V}, \widetilde{A}) \in \bar{\mathcal{V}} \times \bar{\mathcal{A}}$ as follows:

$$\begin{cases} \widetilde{V}_2(s^{\star}) = H - 1, \\ \widetilde{V}_h(s) = 0, \qquad (h, s) \neq (2, s^{\star}), \end{cases} \qquad \widetilde{A} \equiv \mathbf{0}. \tag{C.17}$$

Then we have

$$D^{\mathsf{M}}(\mathscr{R}, \widehat{\mathscr{R}}) \geq d\Big(\mathscr{R}(\widetilde{V}, \widetilde{A}), \widehat{\mathscr{R}}(\widetilde{V}, \widetilde{A})\Big) = \Big|\big[\mathbb{P}'_1 \widetilde{V}_2\big](\bar{s}, \bar{a}) - \big[\mathbb{P}_1 \widetilde{V}_2\big](\bar{s}, \bar{a})\Big|$$

$$= |(H - 1)(\mathbb{P}_1(s^{\star}|\bar{s}, \bar{a}) - 1)| \geq \frac{H - 1}{2} \geq \frac{1}{2}. \tag{C.18}$$

$\square$

*Proof of Proposition C.4.* Let $\mathscr{R}^\tau$ and $\mathscr{R}^{\widehat{\tau}}$ be the reward mappings induced by $\tau$ and $\widehat{\tau}$, respectively. For any $\theta = (V, A)$, we define $r^\theta = \mathscr{R}^\tau(V, A)$ and $\widehat{r}^\theta = \mathscr{R}^{\widehat{\tau}}(V, A)$. By construction of reward mapping and definition of optimal policy, we have $\pi \in \Pi^\star_{\mathcal{M} \cup r}$ is equivalent to

$$A_h(s, a) \cdot \mathbf{1}\left\{\pi^{\mathsf{E}}_h(a|s) = 0\right\} = 0, \qquad \forall (h, s, a) \text{ s.t. } \pi_h(a|s) \neq 0. \tag{C.19}$$

Similarly, $\pi \in \Pi^\star_{\widehat{\mathcal{M}} \cup \widehat{r}^\theta}$ is equivalent to

$$A_h(s, a) \cdot \mathbf{1}\left\{\widehat{\pi}^{\mathsf{E}}_h(a|s) = 0\right\} = A_h(s, a) \cdot \mathbf{1}\left\{\pi^{\mathsf{E}}_h(a|s) = 0\right\} = 0, \qquad \forall (h, s, a) \text{ s.t. } \pi_h(a|s) \neq 0. \tag{C.20}$$

Hence, we can conclude that $\Pi^\star_{\mathcal{M} \cup r^\theta} = \Pi^\star_{\widehat{\mathcal{M}} \cup \widehat{r}^\theta}$. Notice that $\mathcal{R}_\tau = \left\{r^\theta \mid \theta = (V, A)\right\}$ and $\mathcal{R}_\tau = \left\{\widehat{r}^\theta \mid \theta = (V, A)\right\}$, we further obtain that $D^L(\tau, \widehat{\tau}) = 0$. Then we have

$$\sup_{r \in \mathcal{R}_\tau} \inf_{\widehat{r} \in \mathcal{R}_{\widehat{\tau}}} \sup_{\widehat{\pi}^\star \in \Pi^\star_{\widehat{\mathcal{M}} \cup \widehat{r}}} \max_a \left|Q^{\pi^\star}_1(s_1, a; \mathcal{M} \cup r) - Q^{\widehat{\pi}^\star}_1(s_1, a; \mathcal{M} \cup r)\right|$$

$$= \sup_{\theta \in \Theta} \inf_{\theta' \in \Theta} \sup_{\widehat{\pi}^\star \in \Pi^\star_{\widehat{\mathcal{M}} \cup \widehat{r}^{\theta'}}} \max_a \left|Q^{\pi^\star}_1(s_1, a; \mathcal{M} \cup r^\theta) - Q^{\widehat{\pi}^\star}_1(s_1, a; \mathcal{M} \cup r^\theta)\right|$$

$$= \inf_{\theta \in \Theta} \sup_{\widehat{\pi}^\star \in \Pi^\star_{\widehat{\mathcal{M}} \cup \widehat{r}^\theta}} \max_a \left|Q^{\pi^\star}_1(s_1, a; \mathcal{M} \cup r^\theta) - Q^{\widehat{\pi}^\star}_1(s_1, a; \mathcal{M} \cup r^\theta)\right| = 0, \tag{C.21}$$

where the last line is due to $\Pi^\star_{\mathcal{M} \cup r^\theta} = \Pi^\star_{\widehat{\mathcal{M}} \cup \widehat{r}^\theta}$. Follow the same proof of Eq.(C.21), we have

$$\sup_{\widehat{r} \in \mathcal{R}_{\widehat{\tau}}} \inf_{r \in \mathcal{R}_\tau} \sup_{\pi^\star \in \Pi^\star_{\mathcal{M} \cup r}} \max_a \left|Q^{\pi^\star}_1(s_1, a; \mathcal{M} \cup r) - Q^{\widehat{\pi}^\star}_1(s_1, a; \mathcal{M} \cup r)\right| = 0. \tag{C.22}$$

Combining Eq.(C.21) and Eq.(C.22), we conclude that $D^L(\tau, \widehat{\tau}) = 0$. $\qquad\square$

*proof of Proposition C.5.* Let $\pi$ be an optimal policy of $\mathcal{M} \cup \widehat{r}$, then we have

$$V^\star(s_1; r) - V^{\widehat{\pi}}(s_1; r) = V^\star(s_1; r) - V^\pi(s_1; r) + V^\pi(s_1; r) - V^\pi(s_1; \widehat{r}) + V^\pi(s_1; \widehat{r}) - V^{\widehat{\pi}}(s_1; r)$$

$$\leq |V^\star(s_1; r) - V^\pi(s_1; r)| + |V^\pi(s_1; r) - V^\pi(s_1; \widehat{r})| + \left|V^\pi(s_1; \widehat{r}) - V^{\widehat{\pi}}(s_1; r)\right|$$

$$\leq 2d^{\mathsf{all}}(r, \widehat{r}) + d^{\mathsf{all}}(r, \widehat{r}) + \epsilon' = 3\epsilon + \epsilon',$$

where the last line is valid since

$$|V^\star(s_1; r) - V^\pi(s_1; r)| \leq 2d^{\mathsf{all}}(r, \widehat{r}) \qquad\qquad \text{(by Lemma C.5.)}$$

$$|V^\pi(s_1; r) - V^\pi(s_1; \widehat{r})| \leq d^{\mathsf{all}}(r, \widehat{r}) \qquad\qquad \text{(by definition of } d^{\mathsf{all}}.)$$

$$\left|V^\pi(s_1; \widehat{r}) - V^{\widehat{\pi}}(s_1; r)\right| \leq \epsilon'. \qquad\qquad \text{(by definition of } \widehat{\pi}.)$$

$$\square$$

*Proof of Proposition C.6.* Since $\widehat{\pi}$ is an $\epsilon$-optiaml policy in $\mathcal{M} \cup \widehat{r}$, we have

$$\epsilon' + V^{\widehat{\pi}}(s_1; \widehat{r}) \geq V^\pi(s_1; \widehat{r}). \tag{C.23}$$

In the same way, $\pi$ is an $\bar{\epsilon}$-optiaml policy in $\mathcal{M} \cup r$, and therefore, we obtain that

$$\bar{\epsilon} + V^\pi(s_1; r) \geq V^{\widehat{\pi}}(s_1; r). \tag{C.24}$$

And by $\widehat{r}_h(s, a) \leq r_h(s, a)$ for all $(h, s, a) \in [H] \times \mathcal{S} \times \mathcal{A}$, we have

$$V^\pi(s_1; r) \geq V^\pi(s_1; \widehat{r}), \qquad V^{\widehat{\pi}}(s_1; r) \geq V^{\widehat{\pi}}(s_1; \widehat{r}). \tag{C.25}$$

Combining Eq.(C.23), Eq.(C.24) and Eq.(C.25), we conclude that

$$\epsilon' + \bar{\epsilon} + V^\pi(s_1; r) \geq \epsilon' + V^{\widehat{\pi}}(s_1; r) \geq \epsilon' + V^{\widehat{\pi}}(s_1; \widehat{r}) \geq V^\pi(s_1; \widehat{r}). \tag{C.26}$$

Hence, we have

$$|V^\pi(s_1; r) - V^{\widehat{\pi}}(s_1; r)| \leq |\epsilon' + \bar{\epsilon} + V^\pi(s_1; r) - \epsilon' + V^{\widehat{\pi}}(s_1; r)| + \bar{\epsilon}$$

$$\leq |\epsilon' + \bar{\epsilon} + V^\pi(s_1; r) - V^\pi(s_1; \widehat{r})| + \bar{\epsilon}$$

$$\leq 2\bar{\epsilon} + \epsilon' + |V^\pi(s_1; r) - V^\pi(s_1; \widehat{r})| \leq \epsilon + \epsilon' + 2\bar{\epsilon}, \tag{C.27}$$

where the first and last line is by triangle inequality and the second is by Eq.(C.26). $\qquad\square$

## D    TECHNICAL TOOLS

**Lemma D.1** (Xie et al. (2021)). *Suppose $N \sim \text{Bin}(n, p)$ where $n \geq 1$ and $p \in [0, 1]$. Then with probability at least $1 - \delta$, we have*

$$\frac{p}{N \vee 1} \leq \frac{8 \log(1/\delta)}{n}.$$

**Theorem D.2** (Metelli et al. (2023)). *Let $\mathbb{P}$ and $\mathbb{Q}$ be probability measures on the same measurable space $(\Omega, \mathcal{F})$, and let $A \in \mathcal{F}$ be an arbitrary event. Then,*

$$\mathbb{P}(A) + \mathbb{Q}(A^c) \geq \frac{1}{2} \exp\left(-D_{KL}(\mathbb{P}, \mathbb{Q})\right),$$

*where $A^c = \Omega \setminus \mathcal{A}$ is the complement of A.*

**Theorem D.3** (Metelli et al. (2023)). *Let $\mathbb{P}_0, \mathbb{P}_1, \ldots, \mathbb{P}_M$ be probability measures on the same measurable space $(\Omega, \mathcal{F})$, and let $A_1, \ldots, A_M \in \mathcal{F}$ be a partition of $\Omega$. Then,*

$$\frac{1}{M} \sum_{i=1}^{M} \mathbb{P}_i(A_i^c) \geq 1 - \frac{\frac{1}{M} \sum_{i=1}^{M} D_{KL}(\mathbb{P}_i, \mathbb{P}_0) - \log 2}{\log M},$$

*where $A^c = \Omega \setminus A$ is the complement of A.*

## E    PROOFS FOR SECTION 4

### E.1    SOME LEMMAS

**Lemma E.1** (Concentration event ). *Under the setting of Theorem 4.2, there exists an absolute constant $C_1$, $C_2$ such that the concentration event $\mathcal{E}$ holds with probability at least $1 - \delta$, where*

$$\mathcal{E} := \Bigg\{ (i): \left| \left[ (\mathbb{P}_h - \widehat{\mathbb{P}}_h) V_{h+1} \right](s, a) \right| \leq b_h^\theta(s, a) \quad \forall \theta = (V, A) \in \Theta, \ (h, s, a) \in [H] \times \mathcal{S} \times \mathcal{A}, \tag{E.1}$$

$$(ii): \frac{1}{N_h(s, a) \vee 1} \leq \frac{C_1 \iota}{K d_h^{\pi^b}(s, a)} \quad \forall (h, s, a) \in [H] \times \mathcal{S} \times \mathcal{A}, \tag{E.2}$$

$$(iii): N_h^e(s, a) \geq 1 \quad \forall (s, a) \in \mathcal{S} \times \mathcal{A} \ s.t. \ d_h^{\pi^b}(s, a) \geq \frac{C_2 \eta \iota}{K}, a \in \text{supp}\left(\pi_h^{\mathsf{E}}(\cdot|s)\right) \Bigg\}, \tag{E.3}$$

*where $b_h(s, a)$ is defined in Eq.(4.3), $C^\star$ is specified in Definition 4.1, and $N_h^e(s), \eta$ are given by*

$$N_h^e(s, a) := \begin{cases} \sum_{(s_h, a_h, e_h) \in \mathcal{D}} \mathbf{1}\{(s_h, e_h) = (s, a)\} & \text{in option 1,} \\ N_h^b(s, a) & \text{in option 2,} \end{cases} \qquad \eta := \begin{cases} \frac{1}{\Delta} & \text{in option 1,} \\ 1 & \text{in option 2.} \end{cases}$$

*Proof.* When $N_h(s, a) = 0$, then $\widehat{\mathbb{P}}_h(\cdot|s, a) = 0$, as a result, claim (i) holds trivially. We then consider the case where $N_h(s, a) \geq 1$.

For any $h \in [H]$, we define $\mathcal{N}_{\epsilon,h}$ as an $\epsilon/H$-net with respect to $\|\cdot\|_\infty$ norm for $\overline{\mathcal{V}}_h^\Theta$. By definition of $\mathcal{N}(\Theta; \epsilon/H)$, we have

$$\log |\mathcal{N}_{\epsilon,h}| \leq \log \mathcal{N}(\Theta; \epsilon/H).$$

For fixed $\widetilde{V}_{h+1} \in \mathcal{N}_{\epsilon,h+1}$, $(h, s, a) \in [H] \times \mathcal{S} \times \mathcal{A}$, by the empirical Bernstein inequality (Maurer & Pontil, 2009, Theorem 4), there exists some absolute constant $c > 0$ such that

$$\left| \left[ (\mathbb{P}_h - \widehat{\mathbb{P}}_h) \widetilde{V}_{h+1} \right](s, a) \right| \leq \sqrt{\frac{c}{N_h^b(s, a) \vee 1} \left[ \widehat{\mathbb{V}}_h \widetilde{V}_{h+1} \right](s, a) \log \frac{3HSA \cdot |\mathcal{N}_{\epsilon,h+1}|}{\delta}}$$

$$+ \frac{cH}{N_h^b(s,a) \vee 1} \log \frac{3HSA \cdot |\mathcal{N}_{\epsilon,h+1}|}{\delta}$$

$$\lesssim \sqrt{\frac{c \log \mathcal{N}(\Theta; \epsilon/H)\iota}{N_h^b(s,a) \vee 1} \left[\widehat{\mathbb{V}}_h \widetilde{V}_{h+1}\right](s,a)} + \frac{cH \log \mathcal{N}(\Theta; \epsilon/H)\iota}{N_h^b(s,a) \vee 1}$$

with probability at least $1 - \delta/(3HSA|\mathcal{N}_{\epsilon,h}|)$. Here $\lesssim$ hides absolute constants. Taking the union bound over all $\widetilde{V}_{h+1} \in \mathcal{N}_{\epsilon,h+1}$ and $(h,s,a) \in [H] \times \mathcal{S} \times \mathcal{A}$, we know that with probability at least $1 - \delta/3$,

$$\left|\left[(\mathbb{P}_h - \widehat{\mathbb{P}}_h)\widetilde{V}_{h+1}\right](s,a)\right| \lesssim \sqrt{\frac{c \log \mathcal{N}(\Theta; \epsilon/H)\iota}{N_h^b(s,a) \vee 1} \left[\widehat{\mathbb{V}}_h \widetilde{V}_{h+1}\right](s,a)} + \frac{cH \log \mathcal{N}(\Theta; \epsilon/H)\iota}{N_h^b(s,a) \vee 1}$$

hold simultaneously for all $\widetilde{V} \in \mathcal{N}_{\epsilon,h}$ and $(h,s,a) \in [H] \times \mathcal{S} \times \mathcal{A}$.

For any $(V, A) \in \Theta$ and $h \in [H]$, there exists a $\widetilde{V}_h$ such that $\|V_h - \widetilde{V}_h\|_\infty \leq \epsilon/H$. Denote $(\widetilde{V}_1, \ldots, \widetilde{V}_H)$ as $\widetilde{V}$. By applying the triangle inequality, we deduce that

$$\left|\left[(\mathbb{P}_h - \widehat{\mathbb{P}}_h)V_{h+1}\right](s,a)\right| \leq \left|\left[(\widehat{\mathbb{P}}_h - \mathbb{P}_h)\widetilde{V}_{h+1}\right](s,a)\right| + 2\|\widetilde{V} - V\|_\infty$$

$$\lesssim \sqrt{\frac{c \log \mathcal{N}(\Theta; \epsilon/H)\iota}{N_h^b(s,a) \vee 1} \left[\widehat{\mathbb{V}}_h \widetilde{V}_{h+1}\right](s,a)} + \frac{cH \log \mathcal{N}(\Theta; \epsilon/H)\iota}{N_h^b(s,a) \vee 1} + \frac{\epsilon}{H}$$

$$\leq \sqrt{\frac{c \log \mathcal{N}(\Theta; \epsilon/H)\iota}{N_h^b(s,a) \vee 1} \left[\widehat{\mathbb{V}}_h V_{h+1}\right](s,a)} + \sqrt{\frac{c \log \mathcal{N}(\Theta; \epsilon/H)\iota}{N_h^b(s,a) \vee 1} \left[\widehat{\mathbb{V}}_h \left(\widetilde{V}_{h+1} - V_{h+1}\right)\right](s,a)}$$

$$+ \frac{cH \log \mathcal{N}(\Theta; \epsilon/H)\iota}{N_h^b(s,a) \vee 1} + \frac{\epsilon}{H}$$

$$\leq \sqrt{\frac{c \log \mathcal{N}(\Theta; \epsilon/H)\iota}{N_h^b(s,a) \vee 1} \left[\widehat{\mathbb{V}}_h V_{h+1}\right](s,a)} + \sqrt{\frac{c \log \mathcal{N}(\Theta; \epsilon/H)\iota \epsilon^2}{H^2 \cdot N_h^b(s,a) \vee 1}} + \frac{cH \log \mathcal{N}(\Theta; \epsilon/H)\iota}{N_h^b(s,a) \vee 1} + \frac{\epsilon}{H}$$

$$= \sqrt{\frac{c \log \mathcal{N}(\Theta; \epsilon/H)\iota}{N_h^b(s,a) \vee 1} \left[\widehat{\mathbb{V}}_h V_{h+1}\right](s,a)} + \frac{cH \log \mathcal{N}(\Theta; \epsilon/H)\iota}{N_h^b(s,a) \vee 1} + \frac{\epsilon}{H}\left(1 + \sqrt{\frac{c \log \mathcal{N}(\Theta; \epsilon/H)\iota}{N_h^b(s,a) \vee 1}}\right)$$

$$\tag{E.4}$$

holds with probability at least $1 - \delta/3$ for all $\theta = (V, A) \in \Theta$ and $(h,s,a) \in [H] \times \mathcal{S} \times \mathcal{A}$. Here the third line is by $\left[\widehat{\mathbb{V}}_h \widetilde{V}_{h+1}\right](s,a) \leq \left[\widehat{\mathbb{V}}_h V_{h+1}\right](s,a) + \left[\widehat{\mathbb{V}}_h \left(\widetilde{V}_{h+1} - V_{h+1}\right)\right](s,a)$. On the other hand, by $|V_h|_\infty \leq H - h + 1$, we obtain that

$$\left|\left[\left(\mathbb{P}_h - \widehat{\mathbb{P}}_h\right)V_{h+1}\right](s,a)\right| \leq 2(H - h + 1) \leq 2H, \tag{E.5}$$

for all $V \in \Theta$ and $(h,s,a) \in [H] \times \mathcal{S} \times \mathcal{A}$. Recall that, $b_h^\theta(s,a)$ is given by

$$b_h^\theta(s,a) = C \cdot \min\left\{\sqrt{\frac{\log \mathcal{N}(\Theta; \epsilon/H)\iota}{N_h^b(s,a) \vee 1} \left[\widehat{\mathbb{V}}_h V_{h+1}\right](s,a)} + \frac{H \log \mathcal{N}(\Theta; \epsilon/H)\iota}{N_h^b(s,a) \vee 1}\right.$$

$$\left. + \frac{\epsilon}{H}\left(1 + \sqrt{\frac{\log \mathcal{N}(\Theta; \epsilon/H)\iota}{N_h^b(s,a) \vee 1}}\right), H\right\}, \tag{E.6}$$

for some absolute constant $C$. Combining Eq.(E.4) and Eq.(E.5), it turns out that Claims (ii) holds.

For claim (ii), notice that $N_h(s,a) \sim \text{Bin}(K, d_h^{\pi^b}(s,a))$. Applying Lemma D.1 yields that

$$\frac{1}{N_h(s,a) \vee 1} \leq \frac{8}{K \cdot d_h^{\pi^b}(s_h, a_h^E)} \cdot \log\left(\frac{3HSA}{\delta}\right) \leq \frac{C_1 \iota}{K d_h^{\pi^b}(s,a)}$$

for some absolute constant $C_1$, with probability at least $1 - \delta/(3HSA)$. Taking the union bound yields claim (ii) over all $(h, s, a)$ with probability at least $1 - \delta/3$.

For claim (iii), in option 2, for any $(h, s, a) \in [H] \times \mathcal{S} \times \mathcal{A}$ such that $a \in \text{supp}\left(\pi_h^{\mathsf{E}}(\cdot|s)\right)$ and $d_h^{\pi^{\mathsf{b}}}(s, a) \geq \frac{C_2 \eta \iota}{K}$, we have $N_h^e(s, a) \sim \text{Bin}\left(K, d_h^{\pi^{\mathsf{b}}}(s, a) \cdot \pi_h^{\mathsf{E}}(a|s)\right)$. By direct computing, we obtain that

$$
\begin{aligned}
\mathbb{P}[N_h^e(s, a) = 0] &= (1 - d_h^{\pi^{\mathsf{b}}}(s, a) \cdot \pi_h^{\mathsf{E}}(a|s))^K \leq \left(1 - \Delta \cdot d_h^{\pi^{\mathsf{b}}}(s, a)\right)^K \\
&= \left[1 - \left(\frac{\delta}{3HSA}\right)^{1/K} + \left(\frac{\delta}{3HSA}\right)^{1/K} - \Delta \cdot d_h^{\pi^{\mathsf{b}}}(s, a)\right]^K \\
&\leq \left[\left(\frac{\delta}{3HSA}\right)^{1/K} + \underbrace{1 - \left(\frac{\delta}{3HSA}\right)^{1/K} - \Delta \cdot d_h^{\pi^{\mathsf{b}}}(s, a)}_{\leq 0}\right]^K \\
&\leq \left(\frac{\delta}{3HSA}\right)^{1/K \cdot K} = \frac{\delta}{3HSA},
\end{aligned}
$$

where the second line follows from the well-posedness condition: $\pi^{\mathsf{E}}(a|s) \geq \Delta$ and the last inequality is valid since

$$
1 - \left(\frac{\delta}{3HSA}\right)^{1/K} = 1 - \exp(-\frac{1}{K} \log \frac{\delta}{3HSA}) \leq -\frac{\widetilde{C}_2}{K} \log \frac{\delta}{3HSA} \leq \frac{C_2 \iota}{K} \leq \Delta \cdot d_h^{\pi^{\mathsf{b}}}(s, a),
$$

where $\widetilde{C}_2$ and $C_2$ are absolute constants and the last inequality comes from $d_h^{\pi^{\mathsf{b}}}(s, a) \geq \frac{C_2 \eta \iota}{K} = \frac{C_2 \iota}{\Delta \cdot K}$. Hence, it holds that

$$
N_h^e(s, a) \geq 1,
$$

with probability at least $1 - \delta/(3HSA)$. Taking the union bound over all $(h, s, a) \in [H] \times \mathcal{S} \times \mathcal{A}$ yields that

$$
N_h^e(s, a) \geq 1
$$

holds with probability at least $1 - \delta/3$ for all $(s, a) \in \mathcal{S} \times \mathcal{A}$ s.t. $d_h^{\pi^{\mathsf{b}}}(s, a) \geq \frac{C_2 \eta \iota}{K}, a \in \text{supp}\left(\pi_h^{\mathsf{E}}(\cdot|s)\right)$, which implies that claim (iii) holds.

In option 1, notice that $\mathbb{P}[N_h^e(s, a) = 0] = \left(1 - d_h^{\pi^{\mathsf{b}}(s,a)}(s, a)\right)^K$, with a similar argument, we can prove the claim (iii) in option 1.

Further, we can conclude that the concentration event $\mathcal{E}$ holds with probability at least $1 - \delta$. $\qquad \square$

**Lemma E.2** (Performance decomposition for RLE). *For any $\theta = (V, A) \in \Theta$, let $r^\theta = \mathscr{R}^\star(V, A)$ and $\widehat{r}^\theta = \widehat{\mathscr{R}}(V, A)$, where $\mathscr{R}^\star$ is the ground truth reward mapping and $\widehat{\mathscr{R}}$ is the estimated reward mapping output by* RLP. *On the event defined in Lemma E.1, for all $\theta \in \Theta$, $h \in [H]$, we have*

$$
d^{\pi^{\mathsf{val}}}\left(r^\theta, \widehat{r}^\theta\right) \lesssim \frac{C^\star H^2 S \eta \iota}{K} + \sum_{h \in [H]} \sum_{(s,a) \in \mathcal{S} \times \mathcal{A}} d_h^{\pi^{\mathsf{val}}}(s, a) b_h^\theta(s, a),
$$

*where $C^\star$ is defined in Assumption 4.1 and $\eta$ is specified in Lemma E.1*

*Proof.* Consider a tuple $(h, s, a) \in [H] \times \mathcal{S} \times \mathcal{A}$. When $a \notin \text{supp}\left(\pi_h^{\mathsf{E}}(\cdot|s)\right)$, by definition of $N_h^e(s, a)$, we have $N_h^e(s, a) = 0$ By construction of $\widehat{\pi}_h^{\mathsf{E}}(a|s)$ of Algorithm 1, we deduce that $\widehat{\pi}_h^{\mathsf{E}}(a|s) = 0$, and therefore

$$
\left|\mathbf{1}\left\{a \notin \text{supp}\,\widehat{\pi}_h^{\mathsf{E}}(\cdot|s)\right\} - \mathbf{1}\left\{a \notin \text{supp}\left(\pi_h^{\mathsf{E}}(\cdot|s)\right)\right\}\right| = |0 - 0| = 0.
$$

When $a \in \text{supp}\left(\pi_h^{\mathsf{E}}(\cdot|s)\right)$ and $d_h^{\pi^{\mathsf{b}}}(s, a) < \frac{C_2 \eta \iota}{K}$, then

$$
\left|\mathbf{1}\left\{a \notin \text{supp}\left(\widehat{\pi}_h^{\mathsf{E}}(\cdot|s)\right)\right\} - \mathbf{1}\left\{a \notin \text{supp}\left(\pi_h^{\mathsf{E}}(\cdot|s)\right)\right\}\right| \leq 2.
$$

If $a \in \text{supp}\left(\pi_h^{\mathsf{E}}(\cdot|s)\right)$ and $d_h^{\pi^{\mathsf{b}}}(s,a) \geq \frac{C_2 \eta \iota}{K}$, then by concentration event $\mathcal{E}$ (iii), $N_h^e(s,a) \geq 1$ which implies that $\widehat{\pi}_h^{\mathsf{E}}(a|s) > 0$. Hence, we obtain that

$$\left|\mathbf{1}\left\{a \notin \text{supp}\left(\widehat{\pi}_h^{\mathsf{E}}(\cdot|s)\right)\right\} - \mathbf{1}\left\{a \notin \text{supp}\left(\pi_h^{\mathsf{E}}(\cdot|s)\right)\right\}\right| = |1 - 1| = 0.$$

Thus We can conclude that

$$\left|\mathbf{1}\left\{a \notin \text{supp}\left(\widehat{\pi}_h^{\mathsf{E}}(\cdot|s)\right)\right\} - \mathbf{1}\left\{a \notin \text{supp}\left(\pi_h^{\mathsf{E}}(\cdot|s)\right)\right\}\right| \leq 2 \cdot \mathbf{1}\left\{d_h^{\pi^{\mathsf{b}}}(s,a) < \frac{C_2 \eta \iota}{K}, a \in \text{supp}\left(\pi_h^{\mathsf{E}}(\cdot|s)\right)\right\}. \tag{E.7}$$

We then bound the $\left|\left[r_h^\theta - \widehat{r}_h^\theta\right](s,a)\right|$ for all $(h,s,a) \in [H] \times \mathcal{S} \times \mathcal{A}$:

$$\left|\left[r_h^\theta - \widehat{r}_h^\theta\right](s,a)\right| = \left| - A_h(s,a) \cdot \mathbf{1}\left\{a \notin \text{supp}\left(\pi_h^{\mathsf{E}}(\cdot|s)\right)\right\} + V_h(s) - [\mathbb{P}_h V_{h+1}](s,a)\right.$$

$$\left. + A_h(s,a) \cdot \mathbf{1}\left\{a \notin \text{supp}\left(\widehat{\pi}_h^{\mathsf{E}}(\cdot|s)\right)\right\} - V_h(s) + \left[\widehat{\mathbb{P}}_h V_{h+1}\right](s,a) + b_h^\theta(s,a)\right|$$

$$\leq A_h(s,a) \cdot \left|\mathbf{1}\left\{a \notin \text{supp}\left(\widehat{\pi}_h^{\mathsf{E}}(\cdot|s)\right)\right\} - \mathbf{1}\left\{a \notin \text{supp}\left(\pi_h^{\mathsf{E}}(\cdot|s)\right)\right\}\right| + \left|\left[(\mathbb{P}_h - \widehat{\mathbb{P}}_h)V_{h+1}\right](s,a)\right| + b_h^\theta(s,a)$$

$$\leq 2H \cdot \mathbf{1}\left\{d_h^{\pi^{\mathsf{b}}}(s,a) < \frac{C_2 \eta \iota}{K}, a \in \text{supp}\left(\pi_h^{\mathsf{E}}(\cdot|s)\right)\right\} + \left|\left[(\mathbb{P}_h - \widehat{\mathbb{P}}_h)V_{h+1}\right](s,a)\right| + b_h^\theta(s,a)$$

$$\leq 2H \cdot \mathbf{1}\left\{d_h^{\pi^{\mathsf{b}}}(s,a) < \frac{C_2 \eta \iota}{K}, a \in \text{supp}\left(\pi_h^{\mathsf{E}}(\cdot|s)\right)\right\} + 2b_h^\theta(s,a), \tag{E.8}$$

where the second line follows from the triangle inequality, the third line comes from Eq.(E.7), the second last line follows from $\|A_h\|_\infty \leq H$, the last line comes from the concentration event $\mathcal{E}$ (i). Finally, we give the bound of $\mathbb{E}_{\pi^{\mathsf{val}}}|V_h^{\pi^{\mathsf{val}}}(s; r^\theta) - V_h^{\pi^{\mathsf{val}}}(s; \widehat{r}^\theta)|$. By definition of the $V$ function we have

$$\mathbb{E}_{\pi^{\mathsf{val}}}\left|V_h^{\pi^{\mathsf{val}}}(s; r^\theta) - V_h^{\pi^{\mathsf{val}}}(s; \widehat{r}^\theta)\right| = \sum_{s \in \mathcal{S}} d_h^{\pi^{\mathsf{val}}}(s) \cdot \left|V_h^{\pi^{\mathsf{val}}}(s; r^\theta) - V_h^{\pi^{\mathsf{val}}}(s; \widehat{r}^\theta)\right|$$

$$= \sum_{s' \in \mathcal{S}} d_h^{\pi^{\mathsf{val}}}(s') \cdot \left|\sum_{h' \geq h}\sum_{(s,a) \in \mathcal{S} \times \mathcal{A}} d_{h'}^{\pi^{\mathsf{val}}}(s_{h'} = s, a_{h'} = a|s_h = s') \cdot \left[r_{h'}^\theta - \widehat{r}_{h'}^\theta\right](s,a)\right|$$

$$\leq \sum_{h' \geq h}\sum_{(s,a) \in \mathcal{S} \times \mathcal{A}} \left\{\sum_{s \in \mathcal{S}} d_h^{\pi^{\mathsf{val}}}(s) \cdot d_{h'}^{\pi^{\mathsf{val}}}(s,a|s_h = s)\right\} \cdot \left|\left[r_{h'}^\theta - \widehat{r}_{h'}^\theta\right](s,a)\right|$$

$$\overset{(i)}{\leq} \sum_{h' \geq h}\sum_{(s,a) \in \mathcal{S} \times \mathcal{A}} d_{h'}^{\pi^{\mathsf{val}}}(s,a) \cdot \left|\left[r_{h'}^\theta - \widehat{r}_{h'}^\theta\right](s,a)\right|$$

$$\overset{(ii)}{\leq} \sum_{h' \geq h}\sum_{(s,a) \in \mathcal{S} \times \mathcal{A}} d_{h'}^{\pi^{\mathsf{val}}}(s,a) \cdot \left[2H \cdot \mathbf{1}\left\{d_h^{\pi^{\mathsf{b}}}(s,a) < \frac{C_2 \eta \iota}{K}, a \in \text{supp}\left(\pi_h^{\mathsf{E}}(\cdot|s)\right)\right\} + 2b_h^\theta(s,a)\right]$$

$$\leq \sum_{h \in [H]}\sum_{(s,a) \in \mathcal{S} \times \mathcal{A}} \frac{2H d_h^{\pi^{\mathsf{val}}}(s,a)}{d_h^{\pi^{\mathsf{b}}}(s,a)} d_h^{\pi^{\mathsf{b}}}(s,a) \cdot \mathbf{1}\left\{d_h^{\pi^{\mathsf{b}}}(s,a) < \frac{C_2 \eta \iota}{K}, a \in \text{supp}\left(\pi_h^{\mathsf{E}}(\cdot|s)\right)\right\}$$

$$+ \sum_{h \geq 1}\sum_{(s,a) \in \mathcal{S} \times \mathcal{A}} 2d_h^{\pi^{\mathsf{val}}}(s,a) b_h^\theta(s,a)$$

$$\leq 2H \cdot \frac{C_2 \eta \iota}{K} \cdot \sum_{h \in [H]}\sum_{(s,a) \in \mathcal{S} \times \mathcal{A}} \frac{d_h^{\pi^{\mathsf{val}}}(s,a)}{d_h^{\pi^{\mathsf{b}}}(s,a)} + \sum_{h \geq 1}\sum_{(s,a) \in \mathcal{S} \times \mathcal{A}} 2d_h^{\pi^{\mathsf{val}}}(s,a) b_h^\theta(s,a)$$

$$\overset{(iii)}{\lesssim} \frac{C^\star H^2 S \eta \iota}{K} + \sum_{h \in [H]}\sum_{(s,a) \in \mathcal{S} \times \mathcal{A}} d_h^{\pi^{\mathsf{val}}}(s,a) b_h^\theta(s,a),$$

where $d_{h'}^{\pi^{\mathsf{val}}}(s_{h'} = s, a_{h'} = a|s_h = s) = \mathbb{P}_h(s_{h'} = s, a_{h'} = a|s_h = s)$, (i) is due to $d_{h'}^{\pi^{\mathsf{val}}}(s,a) = \sum_{s \in \mathcal{S}} d_h^{\pi^{\mathsf{val}}}(s) \cdot d_{h'}^{\pi^{\mathsf{val}}}(s_{h'} = s, a_{h'} = a|s_h = s)$, (ii) follows from Eq.(E.8) and (iii) comes from definition of $C^\star$-concentrability. This completes the proof. $\qquad\square$

### E.2 Proofs of Theorem 4.2

*Proof.* By Lemma E.2, we have

$$D_\Theta^{\pi^{\text{val}}}(\mathscr{R}^\star, \widehat{\mathscr{R}}) \lesssim \frac{C^\star H^2 S \eta \iota}{K} + \underbrace{\sup_{\theta \in \Theta} \sum_{h \in [H]} \sum_{(s,a) \in \mathcal{S} \times \mathcal{A}} d_h^{\pi^{\text{val}}}(s,a) b_h^\theta(s,a)}_{\text{(I)}}. \tag{E.9}$$

Plugging Eq.(E.6) into Eq.(E.9), we obtain that

$$\text{(I)} = \sum_{h \in [H]} \sum_{(s,a) \in \mathcal{S} \times \mathcal{A}} d_h^{\pi^{\text{val}}}(s,a) b_h(s,a)$$

$$\lesssim \sum_{h \in [H]} \sum_{(s,a) \in \mathcal{S} \times \mathcal{A}} d_h^{\pi^{\text{val}}}(s,a) \cdot \left\{ \sqrt{\frac{\log \mathcal{N}(\Theta; \epsilon/H) \iota}{N_h^b(s,a) \vee 1} \left[\widehat{\mathbb{V}}_h V_{h+1}\right](s,a)} \right. \tag{E.10}$$

$$\left. + \frac{H \log \mathcal{N}(\Theta; \epsilon/H) \iota}{N_h^b(s,a) \vee 1} + \frac{\epsilon}{H} \left(1 + \sqrt{\frac{\log \mathcal{N}(\Theta; \epsilon/H) \iota}{N_h^b(s,a) \vee 1}}\right) \right\}$$

$$\leq \underbrace{\sum_{h \in [H]} \sum_{(s,a) \in \mathcal{S} \times \mathcal{A}} d_h^{\pi^{\text{val}}}(s,a) \cdot \sqrt{\frac{\log \mathcal{N}(\Theta; \epsilon/H) \iota}{N_h^b(s,a) \vee 1} [\mathbb{V}_h V_{h+1}](s,a)}}_{\text{(I.a)}}$$

$$+ \underbrace{\sum_{h \in [H]} \sum_{(s,a) \in \mathcal{S} \times \mathcal{A}} d_h^{\pi^{\text{val}}}(s,a) \cdot \sqrt{\frac{\log \mathcal{N}(\Theta; \epsilon/H) \iota}{N_h^b(s,a) \vee 1} \left[\left(\widehat{\mathbb{V}}_h - \mathbb{V}_h\right) V_{h+1}\right](s,a)}}_{\text{(I.b)}}$$

$$+ \underbrace{\sum_{h \in [H]} \sum_{(s,a) \in \mathcal{S} \times \mathcal{A}} d_h^{\pi^{\text{val}}}(s,a) \cdot \frac{H \log \mathcal{N}(\Theta; \epsilon/H) \iota}{N_h^b(s,a) \vee 1}}_{\text{(I.c)}}$$

$$+ \underbrace{\epsilon \cdot \sum_{h \in [H]} \sum_{(s,a) \in \mathcal{S} \times \mathcal{A}} d_h^{\pi^{\text{val}}}(s,a) \cdot \left(\frac{1}{H} + \sqrt{\frac{\log \mathcal{N}(\Theta; \epsilon/H) \iota}{H^2 \cdot N_h^b(s,a) \vee 1}}\right)}_{\text{(I.d)}} \tag{E.11}$$

where the last inequality comes from the triangle inequality. We study the four terms separately. For the term (I.a), on the concentration event $\mathcal{E}$, we have

$$\text{(I.a)} = \sum_{h \in [H]} \sum_{(s,a) \in \mathcal{S} \times \mathcal{A}} d_h^{\pi^{\text{val}}}(s,a) \cdot \sqrt{\frac{\log \mathcal{N}(\Theta; \epsilon/H) \iota}{N_h^b(s,a) \vee 1} [\mathbb{V}_h V_{h+1}](s,a)}$$

$$\lesssim \sum_{h \in [H]} \sum_{(s,a) \in \mathcal{S} \times \mathcal{A}} d_h^{\pi^{\text{val}}}(s,a) \cdot \sqrt{\frac{\log \mathcal{N}(\Theta; \epsilon/H) \iota}{K d^{\pi^b}(s,a)} [\mathbb{V}_h V_{h+1}](s,a)}$$

$$\leq \sqrt{\frac{H^2 \log \mathcal{N}(\Theta; \epsilon/H) \iota}{K}} \cdot \sum_{h \in [H]} \sum_{(s,a) \in \mathcal{S} \times \mathcal{A}} \sqrt{d_h^{\pi^{\text{val}}}(s,a)} \cdot \sqrt{\frac{d_h^{\pi^{\text{val}}}(s,a)}{d_h^{\pi^b}(s,a)}}$$

$$\leq \sqrt{\frac{H^2 \log \mathcal{N}(\Theta; \epsilon/H) \iota}{K}} \cdot \sqrt{\sum_{h \in [H]} \sum_{(s,a) \in \mathcal{S} \times \mathcal{A}} \frac{d_h^{\pi^{\text{val}}}(s,a)}{d_h^{\pi^b}(s,a)}} \cdot \sqrt{\sum_{h \in [H]} \sum_{(s,a) \in \mathcal{S} \times \mathcal{A}} d_h^{\pi^{\text{val}}}(s,a)}$$

$$\text{(by Cauchy-Schwarz inequality)}$$

$$\leq \sqrt{\frac{C^\star H^4 S \log \mathcal{N}(\Theta; \epsilon/H) \iota}{K}}, \tag{E.12}$$

where the second line comes from concentration event $\mathcal{E}$(ii), the third line is valid since $\|V_{h+1}\|_\infty \leq H$ and the last is by thw definition of $C^\star$-concentrability.

Next, we study the term (I.b). For any $(h, s, a)$, we have

$$
\left| \left[ \left( \widehat{\mathbb{V}}_h - \mathbb{V}_h \right) V_{h+1} \right](s, a) \right|
$$
$$
= \left[ (\widehat{\mathbb{P}}_h V_{h+1})^2 - (\widehat{\mathbb{P}}_h V_{h+1})^2 - \left( \mathbb{P}_h (V_{h+1})^2 - (\mathbb{P}_h V_{h+1})^2 \right) \right](s, a)
$$
$$
\leq \left| \left[ (\widehat{\mathbb{P}}_h - \mathbb{P}_h)(V_{h+1})^2 \right](s, a) \right| + \left| \left[ (\widehat{\mathbb{P}}_h + \mathbb{P}_h) V_{h+1} \cdot (\widehat{\mathbb{P}}_h - \mathbb{P}_h) V_{h+1} \right](s, a) \right|
$$
$$
\leq \left| \left[ (\widehat{\mathbb{P}}_h - \mathbb{P}_h)(V_{h+1}) \right](s, a) \right| + 2H \left| \left[ (\widehat{\mathbb{P}}_h - \mathbb{P}_h) V_{h+1} \right](s, a) \right|
$$
$$
\lesssim c \sqrt{\frac{H^4 \iota}{N_h^b(s, a) \vee 1}}, \tag{E.13}
$$

where the second last inequality is by $\|V_{h+1}\|_\infty \leq H$ and the last inequality follows from the Azuma-Hoeffding inequality. By applying Eq.(E.13), we can obtain the bound for the term (I.b):

$$
\text{(I.b)} = \sum_{h \in [H]} \sum_{(s,a) \in \mathcal{S} \times \mathcal{A}} d_h^{\pi^{\text{val}}}(s, a) \cdot \sqrt{\frac{\log \mathcal{N}(\Theta; \epsilon/H) \iota}{N_h^b(s, a) \vee 1}} \left[ \left( \widehat{\mathbb{V}}_h - \mathbb{V}_h \right) V_{h+1} \right](s, a)
$$
$$
\leq \sum_{h \in [H]} \sum_{(s,a) \in \mathcal{S} \times \mathcal{A}} d_h^{\pi^{\text{val}}}(s, a) \cdot \sqrt{\frac{\log \mathcal{N}(\Theta; \epsilon/H) \iota}{N_h^b(s, a) \vee 1}} \cdot \sqrt{\frac{H^4 \iota}{\widehat{N}_h^b(s, a) \vee 1}}
$$
$$
= (\log \mathcal{N}(\Theta; \epsilon/H))^{1/2} \cdot \sum_{h \in [H]} \sum_{(s,a) \in \mathcal{S} \times \mathcal{A}} d_h^{\pi^{\text{val}}}(s, a) \cdot \frac{H \iota^{3/4}}{\{N_h^b(s, a) \vee 1\}^{3/4}}
$$
$$
\leq \underbrace{(\log \mathcal{N}(\Theta; \epsilon/H))^{1/2} \cdot \sum_{h \in [H]} \sum_{(s,a) \in \mathcal{S} \times \mathcal{A}} d_h^{\pi^{\text{val}}}(s, a) \cdot \sqrt{\frac{1}{N_h^b(s, a) \vee 1}}}_{\text{(I.b.1)}}
$$
$$
+ \underbrace{(\log \mathcal{N}(\Theta; \epsilon/H))^{1/2} \cdot \sum_{h \in [H]} \sum_{(s,a) \in \mathcal{S} \times \mathcal{A}} d_h^{\pi^{\text{val}}}(s, a) \cdot \frac{H^2 \iota^{3/2}}{N_h^b(s, a) \vee 1}}_{\text{(I.b.2)}}, \tag{E.14}
$$

where the last line is from AM-GM inequality. For the term (I.b.1), on the concentration event $\mathcal{E}$, we have

$$
\text{(I.b.1)} = (\log \mathcal{N}(\Theta; \epsilon/H))^{1/2} \cdot \sum_{h \in [H]} \sum_{(s,a) \in \mathcal{S} \times \mathcal{A}} d_h^{\pi^{\text{val}}}(s, a) \cdot \sqrt{\frac{1}{N_h^b(s, a) \vee 1}}
$$
$$
\leq (\log \mathcal{N}(\Theta; \epsilon/H))^{1/2} \cdot \sum_{h \in [H]} \sum_{(s,a) \in \mathcal{S} \times \mathcal{A}} \sqrt{d_h^{\pi^{\text{val}}}(s, a)} \cdot \sqrt{\frac{d_h^{\pi^{\text{val}}}(s, a)}{K d_h^{\pi^b}(s, a)}}
$$
$$
\leq (\log \mathcal{N}(\Theta; \epsilon/H))^{1/2} \cdot \sqrt{\frac{1}{K}} \cdot \sqrt{\sum_{h \in [H]} \sum_{(s,a) \in \mathcal{S} \times \mathcal{A}} \frac{d_h^{\pi^{\text{val}}}(s, a)}{d_h^{\pi^b}(s, a)}} \cdot \sqrt{\sum_{h \in [H]} \sum_{(s,a) \in \mathcal{S} \times \mathcal{A}} d_h^{\pi^{\text{val}}}(s, a)}
$$
$$
\leq \sqrt{\frac{C^\star H S \log \mathcal{N}(\Theta; \epsilon/H)}{K}} \cdot \sqrt{\sum_{h \in [H]} \sum_{(s,a) \in \mathcal{S} \times \mathcal{A}} d_h^{\pi^{\text{val}}}(s, a)}
$$
$$
= \sqrt{\frac{C^\star H^2 S \log \mathcal{N}(\Theta; \epsilon/H)}{K}}, \tag{E.15}
$$

where the second line is directly from concentration event $\mathcal{E}(ii)$, the third line follows from Cauchy-Schwarz inequality and the second last line comes from the definition of $C^\star$-concentrability. For the

term (I.b.2), on the concentration event $\mathcal{E}$, we obtain

$$
\begin{aligned}
(\text{I.b.2}) &= (\log \mathcal{N}(\Theta; \epsilon/H))^{1/2} \cdot \sum_{h \in [H]} \sum_{(s,a) \in \mathcal{S} \times \mathcal{A}} d_h^{\pi^{\text{val}}}(s,a) \cdot \frac{H^2 \iota^{3/2}}{N_h^b(s,a) \vee 1} \\
&\le (\log \mathcal{N}(\Theta; \epsilon/H))^{1/2} \cdot \sum_{h \in [H]} \sum_{(s,a) \in \mathcal{S} \times \mathcal{A}} d_h^{\pi^{\text{val}}}(s,a) \cdot \frac{H^2 \iota^{5/2}}{K d_h^{\pi^b}(s,a)} \\
&\le (\log \mathcal{N}(\Theta; \epsilon/H))^{1/2} \cdot \frac{H^2 \iota^{5/2}}{K} \sum_{h \in [H]} \sum_{(s,a) \in \mathcal{S} \times \mathcal{A}} \frac{d_h^{\pi^{\text{val}}}(s,a)}{d_h^{\pi^b}(s,a)} \\
&= \frac{C^\star H^3 S \log \mathcal{N}(\Theta; \epsilon/H) \iota^{5/2}}{K},
\end{aligned}
\tag{E.16}
$$

where the second line comes from concentration event $\mathcal{E}(ii)$, the third line follows from the definition of $C^\star$-concentrability.

Combining Eq.(E.15) and Eq.(E.16), the term (I.b) can be bounded as follows:

$$
(\text{I.b}) \lesssim \sqrt{\frac{C^\star H^2 S \log \mathcal{N}(\Theta; \epsilon/H)}{K}} + \frac{C^\star H^3 S \log \mathcal{N}(\Theta; \epsilon/H) \iota^{5/2}}{K}.
\tag{E.17}
$$

For the term (I.c), observe that

$$
(\text{I.c}) = (\text{I.b.2})/(H \iota^{3/2}).
$$

Hence, by Eq.(E.16), we deduce that

$$
(\text{I.c}) \le \frac{C^\star H^2 S \log \mathcal{N}(\Theta; \epsilon/H) \iota}{K}
\tag{E.18}
$$

For the term (I.d),

$$
\begin{aligned}
(\text{I.d}) &= \epsilon \cdot \sum_{h \in [H]} \sum_{(s,a) \in \mathcal{S} \times \mathcal{A}} d_h^{\pi^{\text{val}}}(s,a) \cdot \left( \frac{1}{H} + \sqrt{\frac{\log \mathcal{N}(\Theta; \epsilon/H) \iota}{H^2 \cdot N_h^b(s,a) \vee 1}} \right) \\
&= \epsilon + \epsilon \cdot \sum_{h \in [H]} \sum_{(s,a) \in \mathcal{S} \times \mathcal{A}} d_h^{\pi^{\text{val}}}(s,a) \cdot \sqrt{\frac{\log \mathcal{N}(\Theta; \epsilon/H) \iota}{H^2 \cdot N_h^b(s,a) \vee 1}} \\
&= \epsilon + \epsilon \cdot \sum_{h \in [H]} \sum_{(s,a) \in \mathcal{S} \times \mathcal{A}} d_h^{\pi^{\text{val}}}(s,a) \cdot \sqrt{\frac{\log \mathcal{N}(\Theta; \epsilon/H) \iota}{H^2 K d_h^{\pi^b}(s,a)}} \\
&= \epsilon + \epsilon \sqrt{\frac{\log \mathcal{N}(\Theta; \epsilon/H) \iota}{H^2 K}} \cdot \sum_{h \in [H]} \sum_{(s,a) \in \mathcal{S} \times \mathcal{A}} \sqrt{d_h^{\pi^{\text{val}}}(s,a)} \sqrt{\frac{d_h^{\pi^{\text{val}}}(s,a)}{d_h^{\pi^b}(s,a)}} \\
&\le \epsilon + \epsilon \sqrt{\frac{\log \mathcal{N}(\Theta; \epsilon/H) \iota}{H^2 K}} \sqrt{\sum_{h \in [H]} \sum_{(s,a) \in \mathcal{S} \times \mathcal{A}} d_h^{\pi^{\text{val}}}(s,a)} \cdot \sqrt{\sum_{h \in [H]} \sum_{(s,a) \in \mathcal{S} \times \mathcal{A}} \frac{d_h^{\pi^{\text{val}}}(s,a)}{d_h^{\pi^b}(s,a)}} \\
&\le \epsilon \cdot \left( 1 + \sqrt{\frac{C^\star S \log \mathcal{N}(\Theta; \epsilon/H) \iota}{K}} \right),
\end{aligned}
\tag{E.19}
$$

where the second last line is by Cauchy-Schwarz inequality and the last line is by definition of $C^\star$-concentrablity.

Combining Eq.(E.12), Eq.(E.17), Eq.(E.18) and Eq.(E.19), we deduce that

$$
\begin{aligned}
(\text{I}) &\lesssim (\text{I.a}) + (\text{I.b}) + (\text{I.c}) + (\text{I.d}) \\
&\lesssim \sqrt{\frac{C^\star H^4 S \log \mathcal{N}(\Theta; \epsilon/H) \iota}{K}} + \sqrt{\frac{C^\star H^2 S \log \mathcal{N}(\Theta; \epsilon/H)}{K}} + \frac{C^\star H^3 S \log \mathcal{N}(\Theta; \epsilon/H) \iota^{5/2}}{K} \\
&\quad + \frac{C^\star H^2 S \log \mathcal{N}(\Theta; \epsilon/H) \iota}{K} + \epsilon \cdot \left( 1 + \sqrt{\frac{C^\star S \log \mathcal{N}(\Theta; \epsilon/H) \iota}{K}} \right)
\end{aligned}
$$

$$\lesssim \sqrt{\frac{C^\star H^4 S \log \mathcal{N}(\Theta; \epsilon/H)\iota}{K}} + \frac{C^\star H^3 S \log \mathcal{N}(\Theta; \epsilon/H)\iota^{5/2}}{K} + \epsilon.$$

Finally, plugging into Eq.(E.9), the final bound is given by

$$D_\Theta^{\pi^{\mathsf{val}}}(\mathcal{R}^\star, \widehat{\mathcal{R}}) \lesssim \frac{C^\star H^2 S \eta \iota}{K} + \sqrt{\frac{C^\star H^4 S \log \mathcal{N}(\Theta; \epsilon/H)\iota}{K}} + \frac{C^\star H^3 S \log \mathcal{N}(\Theta; \epsilon/H)\iota^{5/2}}{K} + \epsilon$$

The right-hand-side is upper bounded by $2\epsilon$ as long as

$$K \geq \widetilde{\mathcal{O}}\left( \frac{C^\star H^4 S \log \mathcal{N}(\Theta; \epsilon/H)}{\epsilon^2} + \frac{C^\star H^2 S(\eta + H \log \mathcal{N}(\Theta; \epsilon/H))}{\epsilon} \right).$$

Here $poly \log (H, S, A, 1/\delta)$ are omitted.

$\square$

### E.3 PROOF OF COROLLARY 4.3

*Proof of Corollary 4.3.* In this section, we will consider the case that $\pi^{\mathsf{val}} = \pi^{\mathsf{E}}$.

**Lemma E.3** (Concentration event ). *Under the setting of Theorem 4.2, there exists an absolute constant $C_1$, $C_2$ such that the concentration event $\mathcal{E}$ holds with probability at least $1 - \delta$, where*

$$\mathcal{E} := \left\{ (i) \colon \left| \left[ (\mathbb{P}_h - \widehat{\mathbb{P}}_h) V_{h+1} \right](s, a) \right| \leq b_h^\theta(s, a) \quad \forall \theta = (V, A) \in \Theta, \ (h, s, a) \in [H] \times \mathcal{S} \times \mathcal{A}, \right.$$

(E.20)

$$(ii) \colon \frac{1}{N_h(s, a) \vee 1} \leq \frac{C_1 \iota}{K d_h^{\pi^{\mathsf{b}}}(s, a)}, \quad \forall (h, s, a) \in [H] \times \mathcal{S} \times \mathcal{A},$$

(E.21)

$$\left. (iii) \colon N_h^e(s, a) \geq 1 \quad \forall (h, s, a) \in \mathcal{S} \times \mathcal{A} \ s.t. \ \bar{d}_h(s, a) \geq \frac{C_2 \iota}{K}, a \in \text{supp}\left(\pi_h^{\mathsf{E}}(\cdot|s)\right) \right\},$$

(E.22)

*where $b_h(s, a)$ is defined in Eq.(4.3), $C^\star$ is specified in Definition 4.1, and $N_h^e(s)$ is given by*

$$N_h^e(s, a) := \begin{cases} \sum_{(s_h, a_h, e_h) \in \mathcal{D}} \mathbf{1}\left\{ (s_h, e_h) = (s, a) \right\} & \text{in option 1,} \\ N_h^b(s, a) & \text{in option 2,} \end{cases}$$

$$\bar{d}_h(s, a) := \begin{cases} d_h^{\pi^{\mathsf{b}}}(s) \cdot \pi^{\mathsf{E}}(a|s) & \text{in option 1,} \\ d_h^{\pi^{\mathsf{b}}}(s, a) & \text{in option 2.} \end{cases}$$

(E.23)

*Proof.* Repeating the arguments in the proof of Lemma E.1, we prove that claim (i), (ii) holds with probability at least $1 - \frac{2\delta}{3}$.

For claim (iii), in option 2, for any $(h, s, a) \in [H] \times \mathcal{S} \times \mathcal{A}$ such that $a \in \text{supp}\left(\pi_h^{\mathsf{E}}\cdot|s\right)$ and $\bar{d}_h(s, a) \geq \frac{C_2 \iota}{K}$, $N_h^e(s, a) \sim \text{Bin}\left(K, \bar{d}_h(s, a)\right)$. By direct computing, we obtain that

$$\mathbb{P}[N_h^e(s, a) = 0] = \left(1 - \bar{d}_h(s, a)\right)^K = \left[ 1 - \left(\frac{\delta}{3HSA}\right)^{1/K} + \left(\frac{\delta}{3HSA}\right)^{1/K} - \bar{d}_h(s, a) \right]^K$$

$$\leq \left[ \left(\frac{\delta}{3HSA}\right)^{1/K} + \underbrace{1 - \left(\frac{\delta}{3HSA}\right)^{1/K} - \bar{d}_h(s, a)}_{\leq 0} \right]^K$$

$$\leq \left(\frac{\delta}{3HSA}\right)^{1/K \cdot K} = \frac{\delta}{3HSA},$$

where the last inequality is valid since

$$1 - \left(\frac{\delta}{3HSA}\right)^{1/K} = 1 - \exp(-\frac{1}{K}\log\frac{\delta}{3HSA}) \leq -\frac{\widetilde{C}_2}{K}\log\frac{\delta}{3HSA} \leq \frac{C_2\iota}{K} \leq \bar{d}_h(s,a),$$

where $\widetilde{C}_2$ and $C_2$ are absolute constants. Hence, it holds that

$$N_h^e(s,a) \geq 1,$$

with probability at least $1 - \delta/(3HSA)$. Taking the union bound over all $(h,s,a) \in [H] \times \mathcal{S} \times \mathcal{A}$ yields that

$$N_h^e(s,a) \geq 1$$

holds with probability at least $1 - \delta/3$ for all $(h,s,a) \in \mathcal{S} \times \mathcal{A}$ s.t. $\bar{d}_h(s,a) \geq \frac{C_2\iota}{K}$, $a \in \text{supp}\left(\pi_h^{\mathsf{E}}(\cdot|s)\right)$, which implies that claim (iii) holds. In option 2, it is crucial to observe that: that $\mathbb{P}[N_h^e(s,a)] = (1 - d_h^{\pi^{\mathsf{b}}}(s,a))^K = (1 - \bar{d}_h(s,a))^K$. By repeating similar arguments utilized in the proof of the case of option 2, we can prove the claim (iii) holds. Further, we can conclude that the concentration event $\mathcal{E}$ holds with probability at least $1 - \delta$. $\qquad\square$

Recall that $r^\theta = \mathscr{R}^\star(V,A)$ and $\widehat{r}^\theta = \widehat{\mathscr{R}}(V,A)$ for any $\theta = (V,A) \in \Theta$. When $\pi^{\mathsf{val}} = \pi^{\mathsf{E}}$, repeating the arguments in Lemma E.2, we have following decomposition:

$$
\begin{aligned}
d^{\pi^{\mathsf{val}}}\left(r^\theta,\widehat{r}^\theta\right) &\leq 2H \cdot \sum_{h\in[H]}\sum_{(s,a)\in\mathcal{S}\times\mathcal{A}} d_h^{\pi^{\mathsf{E}}}(s,a)\cdot\mathbf{1}\left\{\bar{d}_h(s,a) < \frac{C_2\iota}{K}, a\in\text{supp}\left(\pi_h^{\mathsf{E}}(\cdot|s)\right)\right\} \\
&\quad + \sum_{h\in[H]}\sum_{(s,a)\in\mathcal{S}\times\mathcal{A}} 2d_h^{\pi^{\mathsf{val}}}(s,a)b_h^\theta(s,a) \\
&\leq 2H \cdot \sum_{h\in[H]}\sum_{(s,a)\in\mathcal{S}\times\mathcal{A}} \frac{d_h^{\pi^{\mathsf{E}}}(s,a)}{\bar{d}_h(s,a)}\cdot\bar{d}_h(s,a)\cdot\mathbf{1}\left\{\bar{d}_h(s,a) < \frac{C_2\iota}{K}, a\in\text{supp}\left(\pi_h^{\mathsf{E}}(\cdot|s)\right)\right\} \\
&\quad + \sum_{h\in[H]}\sum_{(s,a)\in\mathcal{S}\times\mathcal{A}} 2d_h^{\pi^{\mathsf{val}}}(s,a)b_h^\theta(s,a) \\
&\lesssim \frac{H\iota}{K}\cdot\sum_{h\in[H]}\sum_{(s,a)\in\mathcal{S}\times\mathcal{A}}\frac{d_h^{\pi^{\mathsf{E}}}(s,a)}{\bar{d}_h(s,a)} + \sum_{h\in[H]}\sum_{(s,a)\in\mathcal{S}\times\mathcal{A}} d_h^{\pi^{\mathsf{val}}}(s,a)b_h^\theta(s,a) \\
&\leq \frac{C^\star H^2 SA\iota}{K} + \sum_{h\in[H]}\sum_{(s,a)\in\mathcal{S}\times\mathcal{A}} d_h^{\pi^{\mathsf{val}}}(s,a)b_h^\theta(s,a).
\end{aligned}
$$

where the second last line is valid since

$$
\begin{aligned}
\sum_{h\in[H]}\sum_{(s,a)\in\mathcal{S}\times\mathcal{A}}\frac{d_h^{\pi^{\mathsf{E}}}(s,a)}{\bar{d}_h(s,a)} &= \sum_{h\in[H]}\sum_{(s,a)\in\mathcal{S}\times\mathcal{A}}\frac{d_h^{\pi^{\mathsf{E}}}(s)\cdot\pi_h^{\mathsf{E}}(s,a)}{d_h^{\pi^{\mathsf{b}}}(s)\cdot\pi_h^{\mathsf{E}}(s,a)} \\
&= A\cdot\sum_{h\in[H]}\sum_{s\in\mathcal{S}}\frac{d_h^{\pi^{\mathsf{E}}}(s)}{d_h^{\pi^{\mathsf{b}}}(s)} \\
&= A\cdot\sum_{h\in[H]}\sum_{s\in\mathcal{S}}\frac{\sum_{a\in\mathcal{A}}d_h^{\pi^{\mathsf{E}}}(s,a)}{\sum_{a\in\mathcal{A}}d_h^{\pi^{\mathsf{b}}}(s,a)} \\
&\leq A\cdot\sum_{h\in[H]}\sum_{(s,a)\in\mathcal{S}\times\mathcal{A}}\max_{a\in\mathcal{A}}\frac{d_h^{\pi^{\mathsf{E}}}(s,a)}{d_h^{\pi^{\mathsf{b}}}(s,a)} \\
&\leq A\cdot\sum_{h\in[H]}\sum_{(s,a)\in\mathcal{S}\times\mathcal{A}}\frac{d_h^{\pi^{\mathsf{E}}}(s,a)}{d_h^{\pi^{\mathsf{b}}}(s,a)} \\
&\leq C^\star HSA.
\end{aligned}
$$

Similar as Eq.(E.9), we can decompose $D_{\Theta}^{\pi^{\mathsf{E}^{\mathrm{val}}}}(\mathscr{R}^{\star}, \widehat{\mathscr{R}})$ as follows:

$$D_{\Theta}^{\pi^{\mathsf{E}}}(\mathscr{R}^{\star}, \widehat{\mathscr{R}}) \lesssim \frac{C^{\star} H^2 S \eta \iota}{K} + \underbrace{\sup_{\theta \in \Theta} \sum_{(s,a) \in \mathcal{S} \times \mathcal{A}} d_h^{\pi^{\mathrm{val}}}(s,a) b_h^{\theta}(s,a)}_{(\mathrm{I})} \tag{E.24}$$

We can decompose terms (I) into four terms (I.a), (I.b), (I.c), and (I.d) as in Eq.(E.11). Since we don't use claim (iii) in the proof of bounding (I.b), (I.c), and (I.d), Eq.(E.17), Eq.(E.18) and Eq.(E.19) still holds on the concentration event $\mathcal{E}$ defined in Lemma E.1. In the following, we give an improved bound of the term (I.a):

$$(\mathrm{I.a}) = \sum_{h \in [H]} \sum_{(s,a) \in \mathcal{S} \times \mathcal{A}} d_h^{\pi^{\mathsf{E}}}(s,a) \cdot \sqrt{\frac{\log \mathcal{N}(\Theta; \epsilon/H) \iota}{N_h^b(s,a) \vee 1}} [\mathbb{V}_h V_{h+1}](s,a)$$

$$\leq \sum_{h \in [H]} \sum_{(s,a) \in \mathcal{S} \times \mathcal{A}} d_h^{\pi^{\mathsf{E}}}(s,a) \cdot \sqrt{\frac{\log \mathcal{N}(\Theta; \epsilon/H) \iota}{K d^{\pi^b}(s,a)}} [\mathbb{V}_h V_{h+1}](s,a)$$

$$= \sqrt{\frac{\log \mathcal{N}(\Theta; \epsilon/H) \iota}{K}} \cdot \sum_{h \in [H]} \sum_{(s,a) \in \mathcal{S} \times \mathcal{A}} \sqrt{d_h^{\pi^{\mathsf{E}}}(s,a) \cdot [\mathbb{V}_h V_{h+1}](s,a)} \cdot \sqrt{\frac{d_h^{\pi^{\mathsf{E}}}(s,a)}{d_h^{\pi^b}(s,a)}}$$

$$\leq \sqrt{\frac{\log \mathcal{N}(\Theta; \epsilon/H) \iota}{K}} \cdot \sqrt{\sum_{h \in [H]} \sum_{(s,a) \in \mathcal{S} \times \mathcal{A}} d_h^{\pi^{\mathsf{E}}}(s,a) \cdot [\mathbb{V}_h V_{h+1}](s,a)} \cdot \sqrt{\sum_{h \in [H]} \sum_{(s,a) \in \mathcal{S} \times \mathcal{A}} \frac{d_h^{\pi^{\mathsf{E}}}(s,a)}{d_h^{\pi^b}(s,a)}}$$

$$\sqrt{\frac{C^{\star} H S \log \mathcal{N}(\Theta; \epsilon/H) \iota}{K}} \cdot \sqrt{\sum_{h \in [H]} \sum_{(s,a) \in \mathcal{S} \times \mathcal{A}} d_h^{\pi^{\mathsf{E}}}(s,a) \cdot [\mathbb{V}_h V_{h+1}](s,a)} \tag{E.25}$$

We then give a sharp bound of $\sum_{h \in [H]} \sum_{(s,a) \in \mathcal{S} \times \mathcal{A}} d_h^{\pi^{\mathsf{E}}}(s,a) \cdot [\mathbb{V}_h V_{h+1}](s,a)$.

$$\sum_{h \in [H]} \sum_{(s,a) \in \mathcal{S} \times \mathcal{A}} d_h^{\pi^{\mathsf{E}}}(s,a) \cdot [\mathbb{V}_h V_{h+1}](s,a)$$

$$= \sum_{h=1}^{H} \mathbb{E}_{\pi^{\mathsf{E}}} \left[ \mathrm{Var}_{\pi^{\mathsf{E}}} \left[ V_{h+1}(s_{h+1}) | s_h, a_h \right] \right]$$

$$\overset{(\mathrm{i})}{=} \sum_{h=1}^{H} \mathbb{E}_{\pi^{\mathsf{E}}} \left[ \mathbb{E} \left[ \left( V_{h+1}(s_{h+1}) + A_h(s_h, a_h) \cdot \mathbf{1} \left\{ a_h \notin \mathrm{supp} \left( \pi^{\mathsf{E}}(\cdot | s_h) \right) \right\} + r_h^{\theta}(s_h, a_h) - V_h(s_h) \right)^2 \Big| s_h, a_h \right] \right]$$

$$= \sum_{h=1}^{H} \mathbb{E}_{\pi^{\mathsf{E}}} \left[ \left( V_{h+1}(s_{h+1}) + A_h(s_h, a_h) \cdot \mathbf{1} \left\{ a_h \notin \mathrm{supp} \left( \pi^{\mathsf{E}}(\cdot | s_h) \right) \right\} + r_h^{\theta}(s_h, a_h) - V_h(s_h) \right)^2 \right]$$

$$\overset{(\mathrm{ii})}{=} \sum_{h=1}^{H} \mathbb{E}_{\pi^{\mathsf{E}}} \left[ \left( V_{h+1}(s_{h+1}) + r_h^{\theta}(s_h, a_h) - V_h(s_h) \right)^2 \right]$$

$$= \mathbb{E}_{\pi^{\mathsf{E}}} \left[ \left( \sum_{h=1}^{H} \left( V_{h+1}(s_{h+1}) + r_h^{\theta}(s_h, a_h) - V_h(s_h) \right) \right)^2 \right]$$

$$+ 2 \sum_{1 \leq h < h' \leq H} \mathbb{E}_{\pi^E} \left[ \left( V_{h+1}(s_{h+1}) + r_h^{\theta}(s_h, a_h) - V_h(s_h) \right) \cdot \left( V_{h'+1}(s_{h'+1}) + r^{\theta}(s_h', a_h') - V_h(s_h') \right) \right]$$

$$\overset{(\mathrm{iii})}{=} \mathbb{E}_{\pi^{\mathsf{E}}} \left[ \left( \sum_{h=1}^{H} \left( V_{h+1}(s_{h+1}) + r_h^{\theta}(s_h, a_h) - V_h(s_h) \right) \right)^2 \right]$$

$$= \mathbb{E}_{\pi^{\mathsf{E}}} \left[ \left( \sum_{h=1}^{H} r_h^{\theta}(s_h, a_h) + \sum_{h=1}^{H} \left( V_{h+1}(s_{h+1}) - V_h(s_h) \right) \right)^2 \right]$$

$$= \mathbb{E}_{\pi^{\mathsf{E}}} \left[ \left( \sum_{h=1}^{H} r_h^{\theta}(s_h, a_h) - V_1(s_1) \right)^2 \right]$$

$$\stackrel{\text{(iv)}}{=} \mathrm{Var}_{\pi^{\mathsf{E}}} \left( \sum_{h=1}^{H} r_h^{\theta}(s_h, a_h) \right) \le H^2., \tag{E.26}$$

where (i) is by definition of reward mapping $\mathbb{P}_h V_{h+1}(s, a) = -A_h(s, a) \cdot \mathbf{1}\left\{ a \in \mathrm{supp}\left( \pi_h^{\mathsf{E}}(\cdot|s) \right) \right\} - r_h^{\theta}(s, a) + V_h(s)$, (ii) comes from

$$\mathbf{1}\left\{ a_h \in \mathrm{supp}\left( \pi_h^{\mathsf{E}}(\cdot|s_h) \right) \right\} = 0$$

for any $(s_h, a_h) \in \mathrm{supp}\left( d_h^{\pi^{\mathsf{E}}}(\cdot) \right)$, (iii) is valid since

$$\left( V_{h+1}(s_{h+1}) + r_h^{\theta}(s_h, a_h) - V_h(s_h) \right) \mathbb{E}_{d^{\pi^{\mathsf{E}}}}[V_{h'+1}(s_{h'+1}) - V_{h'}(s_{h'}) + r_{h'}^{\theta}(s_{h'}, a_{h'})|\mathcal{F}_{h+1}] = 0,$$

and (iv) is by $\Theta \in \overline{\Theta}$. Plugging Eq.(E.26) into Eq.(E.25), we deduce that

$$\text{(I.a)} \le \sqrt{\frac{C^{\star} H^3 S \log \mathcal{N}(\Theta; \epsilon/H)\iota}{K}}. \tag{E.27}$$

Combining Eq.(E.27), Eq.(E.17), Eq.(E.18) and Eq.(E.19), we have

$$\begin{aligned}
\text{(I)} &\lesssim \text{(I.a)} + \text{(I.b)} + \text{(I.c)} + \text{(I.d)} \\
&\lesssim \sqrt{\frac{C^{\star} H^3 S \log \mathcal{N}(\Theta; \epsilon/H)\iota}{K}} + \sqrt{\frac{C^{\star} H^2 S \log \mathcal{N}(\Theta; \epsilon/H)}{K}} + \frac{C^{\star} H^3 S \log \mathcal{N}(\Theta; \epsilon/H)\iota^{5/2}}{K} \\
&\quad + \frac{C^{\star} H^2 S \log \mathcal{N}(\Theta; \epsilon/H)\iota}{K} + \epsilon \cdot (1 + \epsilon \sqrt{\frac{C^{\star} S \log \mathcal{N}(\Theta; \epsilon/H)\iota}{K}}) \\
&\lesssim \sqrt{\frac{C^{\star} H^3 S \log \mathcal{N}(\Theta; \epsilon/H)\iota}{K}} + \frac{C^{\star} H^3 S \log \mathcal{N}(\Theta; \epsilon/H)\iota^{5/2}}{K} + \epsilon.
\end{aligned}$$

Pligging into Eq.(E.24), the final bound is given by

$$D_{\Theta}^{\pi^{\text{val}}}(\mathscr{R}^{\star}, \widehat{\mathscr{R}}) \lesssim \frac{C^{\star} H^2 S A \iota}{K} + \sqrt{\frac{C^{\star} H^3 S \log \mathcal{N}(\Theta; \epsilon/H)\iota}{K}} + \frac{C^{\star} H^3 S \log \mathcal{N}(\Theta; \epsilon/H)\iota^{5/2}}{K} + \epsilon$$

The right-hand-side is upper bounded by $2\epsilon$ as long as

$$K \ge \widetilde{\mathcal{O}} \left( \frac{C^{\star} H^3 S \log \mathcal{N}(\Theta; \epsilon/H)}{\epsilon^2} + \frac{C^{\star} H^2 S (A + H \log \mathcal{N}(\Theta; \epsilon/H))}{\epsilon} \right).$$

Here $poly \log (H, S, A, 1/\delta)$ are omitted.

$\square$

## E.4 FRAMEWORK FOR OFFLINE INVERSE REINFORCEMENT LEARNING

**Pessimism**  As shown in Eq.(E.28), that estimator reward mapping involves a penalty term $b_h^{\theta}(s, a)$. The reason for introducing the penalty term $b_h^{\theta}(s, a)$ is to ensure that our reward satisfies the monotonicity condition: $\left[ \widehat{\mathscr{R}}(V, A) \right]_h (s, a) \le \left[ \widehat{\mathscr{R}}(V, A) \right]_h (s, a)$, which is crucial for the guarantee of the performance of RL algorithms with learned rewards, as demonstrated in Proposition C.6 and Corollary I.6.

**Condition E.4** . With probability at least $1 - \delta$, we have $\sup_{(V, A) \in \Theta} \left| \left[ (\mathbb{P}_h - \widehat{\mathbb{P}}_h) V_{h+1} \right] (s, a) \right| \le b_h^{\theta}(s, a)$ and $\mathrm{supp}\left( \widehat{\pi}_h^{\mathsf{E}}(\cdot|s) \right) \subset \mathrm{supp}\left( \pi_h^{\mathsf{E}}(\cdot|s) \right)$ for all $(h, s) \in [H] \times \mathcal{S}$ and all $(V, A) \in \Theta$.

---

**Algorithm 5** FRAMEWORK FOR OFFLINE INVERSE REINFORCEMENT LEARNING

1: **Input:** Dataset $\mathcal{D}$ collected by executing $\pi^{\mathsf{b}}$ in $\mathcal{M}$.
2: Recover the transition dynamics $\widehat{\mathbb{P}} : [H] \times \mathcal{S} \times \mathcal{A} \to \Delta\mathcal{S}$ and expert policy $\widehat{\pi}^{\mathsf{E}} = \left\{ \widehat{\pi}_h^{\mathsf{E}} : \mathcal{S} \times \Delta(\mathcal{S}) \right\}$ and design the bonus $b : [H] \times \mathcal{S} \times \mathcal{A} \times \Theta \to \mathbb{R}_{\geq 0}$ .
3: Compute $\widehat{\mathscr{R}}$ by

$$[\widehat{\mathscr{R}}(V, A)]_h(s, a) = -A_h(s, a) \cdot \mathbf{1}\left\{ a \notin \mathrm{supp}\left( \widehat{\pi}_h^{\mathsf{E}}(\cdot|s) \right) \right\} + V_h(s) - [\widehat{\mathbb{P}}_h V_{h+1}](s, a) - b_h^\theta(s, a) \tag{E.28}$$

4: **Output**: Estimated reward mapping $\widehat{\mathscr{R}}$.

---

**Theorem E.5** (Learning bound for Algorithm 5). *Suppose that Condition E.4 holds. With probability at least $1 - \delta$, we have $\left[ \widehat{\mathscr{R}}(V, A) \right]_h(s, a) \leq [\mathscr{R}^\star(V, A)]_h(s, a)$ for all $(h, s, a) \in [H] \times \mathcal{S} \times \mathcal{A}$, and*

$$D_\Theta^{\pi^{\mathsf{val}}}\left( \mathscr{R}^\star, \widehat{\mathscr{R}} \right) \leq \sup_{\theta \in \Theta} \Bigg\{ H \cdot \sum_{h \in [H]} \mathbb{E}_{(s,a) \sim d_h^{\pi^{\mathsf{val}}}} \left[ \mathbf{1}\left\{ a \in \mathrm{supp}\left( \pi_h^{\mathsf{E}}(\cdot|s) \right), a \notin \mathrm{supp}\left( \widehat{\pi}_h^{\mathsf{E}}(\cdot|s) \right) \right\} \right]$$

$$+ 2 \sum_{h \in [H]} \mathbb{E}_{(s,a) \sim d_h^{\pi^{\mathsf{val}}}} \left[ b_h^\theta(s, a) \right] \Bigg\}. \tag{E.29}$$

*Proof.* When $\sup_{(V, A) \in \Theta} \left| \left[ (\mathbb{P}_h - \widehat{\mathbb{P}}_h) V_{h+1} \right](s, a) \right| \leq b_h^\theta(s, a)$ and $\mathrm{supp}\, \widehat{\pi}_h^{\mathsf{E}}(\cdot|s) \subset \mathrm{supp}\, \pi_h^{\mathsf{E}}(\cdot|s)$ holds for all $(h, s) \in [H] \times \mathcal{S}$ and all $(V, A) \in \Theta$ hold, we have

$$\left[ \widehat{\mathscr{R}}(V, h) \right]_h(s, a) - [\mathscr{R}^\star(V, h)]_h(s, a)$$

$$= -A_h(s, a) \cdot \left[ \mathbf{1}\left\{ a \notin \mathrm{supp}\left( \widehat{\pi}_h^{\mathsf{E}}(\cdot|s) \right) \right\} - \mathbf{1}\left\{ a \notin \mathrm{supp}\left( \pi_h^{\mathsf{E}}(\cdot|s) \right) \right\} \right] - \left[ \left( \widehat{\mathbb{P}}_h - \mathbb{P}_h \right) V_{h+1} \right](s, a) - b_h^\theta(s, a)$$

$$= \underbrace{-A_h(s, a) \cdot \mathbf{1}\left\{ a \in \mathrm{supp}\left( \pi_h^{\mathsf{E}}(\cdot|s) \right), a \notin \mathrm{supp}\left( \widehat{\pi}_h^{\mathsf{E}}(\cdot|s) \right) \right\}}_{\leq 0} \underbrace{- \left[ \left( \widehat{\mathbb{P}}_h - \mathbb{P}_h \right) V_{h+1} \right](s, a) - b_h^\theta(s, a)}_{\leq 0} \leq 0, \tag{E.30}$$

where the second line is by $\mathrm{supp}\left( \widehat{\pi}_h^{\mathsf{E}}(\cdot|s) \right) \subset \mathrm{supp}\left( \pi_h^{\mathsf{E}}(\cdot|s) \right)$ and $\sup_{(V,A) \in \Theta} \left| \left[ (\mathbb{P}_h - \widehat{\mathbb{P}}_h) V_{h+1} \right](s, a) \right| \leq b_h^\theta(s, a)$. Further, by triangle inequality, we obtain that

$$\left| \left[ \widehat{\mathscr{R}}(V, h) \right]_h(s, a) - [\mathscr{R}^\star(V, h)]_h(s, a) \right|$$

$$\leq A_h(s, a) \cdot \mathbf{1}\left\{ a \in \mathrm{supp}\left( \pi_h^{\mathsf{E}}(\cdot|s) \right), a \notin \mathrm{supp}\left( \widehat{\pi}_h^{\mathsf{E}}(\cdot|s) \right) \right\} + \left| \left[ \left( \widehat{\mathbb{P}}_h - \mathbb{P}_h \right) V_{h+1} \right](s, a) \right| + b_h^\theta(s, a)$$

$$\leq H \cdot \mathbf{1}\left\{ a \in \mathrm{supp}\left( \pi_h^{\mathsf{E}}(\cdot|s) \right), a \notin \mathrm{supp}\left( \widehat{\pi}_h^{\mathsf{E}}(\cdot|s) \right) \right\} + 2 b_h^\theta(s, a), \tag{E.31}$$

where the last line is due to $A_h(s, a) \leq H$ and $\left| \left[ \left( \widehat{\mathbb{P}}_h - \mathbb{P}_h \right) V_{h+1} \right](s, a) \right| \leq b_h^\theta(s, a)$. Similar to the proof of Lemma E.2, we have

$$d^{\pi^{\mathsf{val}}}\left( \widehat{\mathscr{R}}(V, A), \mathscr{R}^\star(V, A) \right) \leq \sum_{h \in [H]} \mathbb{E}_{(s,a) \sim d_h^{\pi^{\mathsf{val}}}} \left[ \left| \left[ \widehat{\mathscr{R}}(V, h) \right]_h(s, a) - [\mathscr{R}^\star(V, h)]_h(s, a) \right| \right]. \tag{E.32}$$

Combining Eq.(E.31) and Eq.(E.32), we obtain that

$$d^{\pi^{\mathsf{val}}}\left( \widehat{\mathscr{R}}(V, A), \mathscr{R}^\star(V, A) \right) \leq H \cdot \sum_{h \in [H]} \mathbb{E}_{(s,a) \sim d_h^{\pi^{\mathsf{val}}}} \left[ \mathbf{1}\left\{ a \in \mathrm{supp}\left( \pi_h^{\mathsf{E}}(\cdot|s) \right), a \notin \mathrm{supp}\left( \widehat{\pi}_h^{\mathsf{E}}(\cdot|s) \right) \right\} \right]$$

$$+ 2 \sum_{h \in [H]} \mathbb{E}_{(s,a) \sim d_h^{\pi^{\mathsf{val}}}} \left[ b_h^\theta(s, a) \right]. \tag{E.33}$$

By the definition of $D_\Theta^{\pi^{\mathsf{val}}}$: $D_\Theta^{\pi^{\mathsf{val}}}\left( \mathscr{R}^\star, \widehat{\mathscr{R}} \right) = \sup_{\theta \in \Theta} d^{\pi^{\mathsf{val}}}\left( \widehat{\mathscr{R}}(V, A), \mathscr{R}^\star(V, A) \right)$, we complete the proof. $\qquad\square$

By Theorem E.5, all we need to do is design $b_h^\theta$ and learn $\widehat{\mathbb{P}}, \widehat{\pi}^{\mathsf{E}}$ from the data to satisfy Condition E.4, thereby obtaining an IRL algorithm. The crux of the problem lies in the design of $b_h^\theta, \widehat{\mathbb{P}}$ and $\widehat{\pi}^{\mathsf{E}}$. In RLP, we employ the pessimism technique from offline RL, and the construction of $b_h^\theta$ and $\widehat{\pi}^{\mathsf{E}}$ using pessimism in RLP satisfies Condition E.4, as illustrated in the proof of Theorem 4.2.

## F  PROOFS FOR SECTION 5

### F.1  FULL DESCRIPTION OF REWARD LEARNING WITH EXPLORATION

We propose a meta-algorithm, named REWARD LEARNING WITH EXPLORATION(RLE). The pseudocode of RLE is presented in Algorithm 6, where the algorithm contains the following three main component

- (Exploring the unknown environment:) This segment involves computing a desired behavior policy $\pi^{\mathsf{b}} = \mathbb{E}_{\pi \sim \mu^{\mathsf{b}}}[\pi]$, which takes the form of a finite mixture of deterministic policies. To achieve this, we need to collect $NH$ episodes of samples. We then execute this policy to gather a total of $K$ episodes worth of samples. Our exploration approach is based on leveraging the exploration scheme outlined in (Li et al., 2023, Algorithm 1). A comprehensive description of this exploration method is postponed and will be provided in Section B. (cf. line 2-3).

- (Subsampling:) For the sake of theoretical simplicity, we apply subsampling. For each $(h, s, a) \in [H] \times \mathcal{S} \times \mathcal{A}$, we populate the new dataset with $\min\left\{\widehat{N}_h^{\mathsf{b}}(s, a), N_h(s, a)\right\}$ sample transitions. Here, $\widehat{N}_h^{\mathsf{b}}(s, a)$, as defined in Eq.(F.1), acts as a lower bound on the total number of visits to $(h, s, a)$ among these $K$ sample episodes, with high probability. (cf. line 4).

- (Computing estimated reward mapping:) With the previously collected dataset at hand, we then utilize the offline IRL algorithm RLP to compute the desired reward mapping.

---

**Algorithm 6** REWARD LEARNING WITH EXPLORATION

1: **Input:** threshold $\xi = c_\xi H^3 S^3 A^3 \iota$, confidence level $\delta$.
2: Call Algorithm 2 to compute the explore policy $\pi^{\mathsf{b}}$.
3: Collect a dataset $\mathcal{D} = \{(s_h^\tau, a_h^\tau, e_h^\tau)\}_{\tau=1, h=1}^{K, H}$ by executing $\pi^{\mathsf{b}}$ in $\mathcal{M}$.
4: Subsampling: subsample $\mathcal{D}$ to obtain $\mathcal{D}^{\mathsf{trim}}$, such that for each $(h, s, a) \in [H] \times \mathcal{S} \times \mathcal{A}$, $\mathcal{D}^{\mathsf{trim}}$ contains $\min\left\{\widehat{N}_h^{\mathsf{b}}(s, a), N_h(s, a)\right\}$ sample transitions randomly drawn from $\mathcal{D}$, where $\widehat{N}_h^{\mathsf{b}}(s, a)$ and $N_h(s, a)$ are defined by

$$N_h(s, a) := \sum_{\tau=1}^{K} \mathbf{1}\left\{(s_h^\tau, a_h^\tau = (s, a)\right\} \quad \widehat{N}_h^{\mathsf{b}}(s, a) := \left[\frac{K}{4}, \mathbb{E}_{\pi \sim \mu^{\mathsf{b}}}[\widehat{d}_h^\pi(s, a)] - \frac{K\xi}{8N} - 3\log\frac{HSA}{\delta}\right]_+,$$

(F.1)

where $\widehat{d}_h^\pi(s, a)$ is specified in Algorithm 2.
5: Call Algorithm 1 to compute the recovered reward mapping $\widehat{\mathscr{R}}$.
6: **Output:** estimated reward mapping $\widehat{\mathscr{R}}$.

---

We remark that our algorithm RLE follows a similar approach to that of Li et al. (2023, Algorithm 1). We begin by computing a desired behavior policy, then proceed to collect data, and finally compute results through the invocation of an offline algorithm. In contrast to the offline setting, we have the flexibility to select the desired behavior. In the following, we will observe that the behavior policy $\pi^{\mathsf{b}}$ exhibits concentrability with any deterministic policy, as shown in Eq.(B.3). This property enables us to achieve our learning goal within the online setting.

## F.2 PROOF OF THEOREM 5.1

**Lemma F.1** (Li et al. (2023)). *Recall that $\xi = c_\xi H^3 S^3 A^3 \log \frac{HSA}{\delta}$ for some large enough constant $c_\xi > 0$ (see line 1 in Algorithm 2). Then, with probability at least $1 - \delta$, the estimated occupancy distributions specified in Eq.(B.1) and (B.2) of Algorithm 2 satisfy*

$$\frac{1}{2}\widehat{d}_h^\pi(s,a) - \frac{\xi}{4N} \leq d_h^\pi(s,a) \leq 2\widehat{d}_h^\pi(s,a) + 2e_h^\pi(s,a) + \frac{\xi}{4N} \tag{F.2}$$

*simultaneously for all $(h,s,a) \in [H] \times \mathcal{S} \times \mathcal{A}$ and all deterministic Markov policy $\pi \in \Pi^{\mathrm{det}}$, provided that*

$$KH \geq N \geq C_N \sqrt{H^9 S^7 A^7 K} \log \frac{HSA}{\delta} \qquad and \qquad K \geq C_K HSA \tag{F.3}$$

*for some large enough constants $C_N, C_K > 0$, where, $\{e_h^\pi(s,a) \in \mathbb{R}_+\}$ satisfies that*

$$\sum_{(s,a)\in\mathcal{S}\times\mathcal{A}} e_h^\pi(s,a) \leq \frac{2SA}{K} + \frac{13SAH\xi}{N} \lesssim \sqrt{\frac{SA}{HK}} \qquad \forall h \in [H], \pi \in \Pi^{\mathrm{det}} \tag{F.4}$$

Notice that Eq.(F.2) only holds for $\pi \in \Pi^{\mathrm{det}}$, however, we will show similar results also hold for any stochastic policy.

For any stochastic policy $\pi = \mathbb{E}_{\pi'\sim\mu}[\pi']$, $d^\pi$ can be expressed as

$$d_h^\pi(s,a) = \mathbb{E}_{\pi'\sim\mu}\left[\widehat{d}_h^{\pi'}(s,a)\right], \qquad \forall(h,s,a) \in [H] \times \mathcal{S} \times \mathcal{A},$$

where $\mu \in \Delta(\Pi^{\mathrm{det}})$. Hence, we can define $\widehat{d}^\pi$ as

$$\widehat{d}_h^\pi(s,a) = \mathbb{E}_{\pi'\sim\mu}\left[\widehat{d}_h^{\pi'}(s,a)\right], \qquad \forall(h,s,a) \in [H] \times \mathcal{S} \times \mathcal{A}.$$

our definition of We remark that although $\mu$ is not unique, our definition of $\widehat{d}^\pi$ necessitates the selection of a specific $\mu \in \Delta(\Pi^{\mathrm{det}})$.

By Eq.(F.2), we have

$$\frac{1}{2}\widehat{d}_h^\pi(s,a) - \frac{\xi}{4N} \leq d_h^\pi(s,a) = \mathbb{E}_{\pi'\sim\mu}\left[d_h^{\pi'}(s,a)\right] \leq 2\widehat{d}_h^\pi(s,a) + 2\mathbb{E}_{\pi'\sim\mu}\left[e_h^{\pi'}(s,a)\right] + \frac{\xi}{4N}$$

$$\frac{1}{2}\widehat{d}_h^\pi(s,a) - \frac{\xi}{4N} \leq d_h^\pi(s,a) = \mathbb{E}_{\pi'\sim\mu}\left[d_h^{\pi'}(s,a)\right] \leq 2\widehat{d}_h^\pi(s,a) + 2\mathbb{E}_{\pi'\sim\mu}\left[e_h^{\pi'}(s,a)\right] + \frac{\xi}{4N}.$$

$$\left(e_h^\pi(s,a) := \mathbb{E}_{\pi'\sim\mu}\left[e_h^{\pi'}(s,a)\right]\right)$$

We also have

$$\sum_{(s,a)\in\mathcal{S}\times\mathcal{A}} e_h^\pi(s,a) = \sum_{(s,a)\in\mathcal{S}\times\mathcal{A}} \mathbb{E}_{\pi'\sim\mu}\left[e_h^{\pi'}(s,a)\right] \leq \frac{2SA}{K} + \frac{13SAH\xi}{N} \lesssim \sqrt{\frac{SA}{HK}},$$

provided Eq.(F.3).

**Lemma F.2** (Concentration event). *Suppose Eq.(F.3). Under the setting of Theorem 5.1, there exists an absolute constants $C_1, C_2 \geq 2$ such that the concentration event $\mathcal{E}$ holds with probability at least $1 - \delta$, where*

$$\mathcal{E} := \Bigg\{ (i)\text{: } \left|\left[(\mathbb{P}_h - \widehat{\mathbb{P}}_h)V_{h+1}\right](s,a)\right| \leq b_h^\theta(s,a) \ \forall \theta = (V,A) \in \Theta, (h,s,a) \in [H] \times \mathcal{S} \times \mathcal{A},$$

$$(ii)\text{: } \frac{1}{2}\widehat{d}_h^\pi(s,a) - \frac{\xi}{4N} \leq d_h^\pi(s,a) \leq 2\widehat{d}_h^\pi(s,a) + 2e_h^\pi(s,a) + \frac{\xi}{4N} \ \forall(h,s,a) \in [H] \times \mathcal{S} \times \mathcal{A}, \pi \in \Pi,$$

$$(iii)\text{: } \widehat{N}_h^b(s,a) \leq N_h^b(s,a) \ \forall(h,s,a) \in [H] \times \mathcal{S} \times \mathcal{A},$$

$$(iv)\text{: } \widehat{N}_h^e(s,a) \geq 1 \ \forall(s,a) \in \mathcal{S} \times \mathcal{A} \text{ s.t. } \widehat{N}_h^b(s,a) \geq \max\{C_2\eta\iota, 1\} \Bigg\}$$

where $b_h^\theta(s,a)$ is defined in Eq.(4.3), $N_h^b(s,a)$ $\widehat{N}_h^b(s,a)$ is defined in Eq.(F.1), $\eta$ are specified in Lemma E.1 and $\widehat{N}_h^e(s,a)$ is given by

$$\widehat{N}_h^e(s,a) := \begin{cases} \sum_{(s_h,a_h,e_h)\in\mathcal{D}^{\mathsf{trim}}} \mathbf{1}\left\{(s_h,e_h)=(s,a)\right\} & \text{in option 1,} \\ \widehat{N}_h^b(s,a) & \text{in option 2,} \end{cases}$$

*Proof.* First, we observe that Claim (i) can be obtained by repeating a similar argument as in Lemma E.1 and Claim (ii) can also be directly derived from Lemma F.1. And claim (iii) has been shown in the proof of (Li et al., 2023, Theorem 2).

Next, we focus on (iv). For claim (iv), in option 1, we have

$$\mathbb{P}\left(\widehat{N}_h^e(s,a)=0\right) = \left(1-\pi_h^{\mathsf{E}}(a|s)\right)^{\widehat{N}_h^b(s,a)} \le \exp\left(\widehat{N}_h^b(s,a)\log\left(1-\eta\right)\right)$$

$$\le \exp\left(\log\frac{\delta}{4HSA}\right) = \frac{\delta}{4HSA},$$

for all $(h,s,a)\in[H]\times\mathcal{S}\times\mathcal{A}$. The last line is valid since

$$\widehat{N}_h^b(s,a)\log\left(1-\eta\right) \le C_2\log\frac{\delta}{HSA}\cdot\frac{\log\left(1-\eta\right)}{\eta} \le \log\frac{\delta}{4HSA},$$

holds for sufficiently large constant $C_2$. In option 2, we have

$$\widehat{N}_h^e(s,a) = \widehat{N}_h^b(s,a) \ge \max\left\{C_2\eta\iota,1\right\} \ge 1,$$

for all $(h,s,a)\in[H]\times\mathcal{S}\times\mathcal{A}$. This completes the proof.

$\square$

### F.3 PROOF OF THEOREM 5.1

Define

$$\mathcal{I}_h = \left\{(s,a)\in\mathcal{S}\times\mathcal{A} \,|\, \mathbb{E}_{\pi'\sim\mu_{\mathsf{b}}}\left[\widehat{d}_h^{\pi'}(s,a)\right] \ge \frac{\xi}{N} + \frac{4(C_2\eta+3)\iota}{K}\right\}, \tag{F.5}$$

for all $h\in[H]$. Then for $(s,a)\in\mathcal{I}_h$, we have

$$\widehat{N}_h^b(s,a) \ge \frac{K}{4}\mathbb{E}_{\pi'\sim\mu_{\mathsf{b}}}\left[\widehat{d}_h^{\pi'}(s,a)\right] - \frac{K\xi}{8N} - 3\iota \ge C_2\eta\iota. \tag{F.6}$$

By concentration event $\mathcal{E}$ (iv), we have

$$\widehat{N}_h^e(s,a) \ge 1,$$

By construction of $\widehat{\pi}^{\mathsf{E}}$ in Algorithm 1, we deduce that

$$\left|\mathbf{1}\left\{a\in\mathrm{supp}\left(\pi_h^{\mathsf{E}}(\cdot|s)\right)\right\} - \mathbf{1}\left\{a\in\mathrm{supp}\left(\widehat{\pi}_h^{\mathsf{E}}(\cdot|s)\right)\right\}\right| = 0. \tag{F.7}$$

for all $(s,a)\in\mathcal{I}_h$.

With $\mathcal{I}_h$ at hand, we can decompose the $d^\pi\left(r^\theta,\widehat{r}^\theta\right)$ for any $\pi$ and $\theta\in\Theta$ as follows:

$$d^\pi\left(r_h^\theta,\widehat{r}_h^\theta\right) \le \sum_{(h,s,a)\in[H]\times\mathcal{S}\times\mathcal{A}} d_h^\pi(s,a)\cdot\left|r_h^\theta(s,a)-\widehat{r}_h^\theta(s,a)\right|$$

$$\le \underbrace{\sum_{h\in[H]}\sum_{(s,a)\notin\mathcal{I}_h} d_h^\pi(s,a)\cdot\left|r_h^\theta(s,a)-\widehat{r}_h^\theta(s,a)\right|}_{(\mathrm{I})} + \underbrace{\sum_{h\in[H]}\sum_{(s,a)\in\mathcal{I}_h} d_h^\pi(s,a)\cdot\left|r^\theta(s,a)-\widehat{r}_h^\theta(s,a)\right|}_{(\mathrm{II})},$$

$$\tag{F.8}$$

where the first line follows the same argument in the proof of Lemma E.2. We then study the terms (I) and (II) separately. For the term (I), by the construction of Algorithm 1, we obtain that

$$(\text{I}) = \sum_{h \in [H]} \sum_{(s,a) \notin \mathcal{I}_h} d_h^\pi(s,a) \cdot \left| r^\theta(s,a) - \widehat{r}^\theta(s,a) \right|$$

$$= \sum_{h \in [H]} \sum_{(s,a) \notin \mathcal{I}_h} d_h^\pi(s,a) \cdot \left| - A_h(s,a) \left( \mathbf{1} \left\{ a \in \text{supp} \left( \pi_h^{\mathsf{E}}(\cdot|s) \right) \right\} - \mathbf{1} \left\{ a \in \text{supp} \left( \widehat{\pi}_h^{\mathsf{E}}(\cdot|s) \right) \right\} \right) - \left[ \left( \mathbb{P}_h - \widehat{\mathbb{P}}_h \right) V_{h+1} \right](s,a) - b_h^\theta \right.$$

$$\leq \sum_{h \in [H]} \sum_{(s,a) \notin \mathcal{I}_h} d_h^\pi(s,a) \cdot \left| A_h(s,a) \cdot \left( \mathbf{1} \left\{ a \in \text{supp} \left( \widehat{\pi}_h^{\mathsf{E}}(\cdot|s) \right) \right\} - \cdot \mathbf{1} \left\{ a \in \text{supp} \left( \pi_h^{\mathsf{E}}(\cdot|s) \right) \right\} \right) \right|$$

$$+ \left| \left[ (\mathbb{P}_h - \widehat{\mathbb{P}}_h) V_{h+1} \right](s,a) \right| + b_h^\theta(s,a) \qquad\qquad \text{(by triangle inequality)}$$

$$\overset{(i)}{\lesssim} H \cdot \sum_{h \in [H]} \sum_{(s,a) \notin \mathcal{I}_h} d_h^\pi(s,a)$$

$$\overset{(ii)}{\lesssim} H \cdot \sum_{h \in [H]} \sum_{(s,a) \notin \mathcal{I}_h} \left( 2\widehat{d}_h^\pi(s,a) + 2e_h^\pi(s,a) + \frac{\xi}{4N} \right)$$

$$\overset{(iii)}{\lesssim} H \cdot \sum_{h \in [H]} \sum_{(s,a) \notin \mathcal{I}_h} \widehat{d}_h^\pi(s,a) + \frac{\xi H^2 SA}{N} + \sqrt{\frac{HSA}{K}}$$

$$= H \cdot \sum_{h \in [H]} \sum_{(s,a) \notin \mathcal{I}_h} \frac{\widehat{d}_h^\pi(s,a)}{\mathbb{E}_{\pi' \sim \mu^{\mathsf{b}}} \left[ \widehat{d}_h^{\pi'}(s,a) \right] + \frac{1}{KH}} \cdot \left( \mathbb{E}_{\pi' \sim \mu^{\mathsf{b}}} \left[ \widehat{d}_h^{\pi'}(s,a) \right] + \frac{1}{KH} \right) + \frac{\xi H^2 SA}{N} + \sqrt{\frac{HSA}{K}}$$

$$\overset{(iv)}{\lesssim} \left( \frac{\xi H}{N} + \frac{4H(C_2\eta + 3)\iota}{K} + \frac{1}{K} \right) \sum_{h \in [H]} \sum_{(s,a) \notin \mathcal{I}_h} \frac{\widehat{d}_h^\pi(s,a)}{\mathbb{E}_{\pi' \sim \mu^{\mathsf{b}}} \left[ \widehat{d}_h^{\pi'}(s,a) \right] + \frac{1}{KH}} + \frac{\xi H^2 SA}{N} + \sqrt{\frac{HSA}{K}}$$

$$\lesssim \left( \frac{H\xi}{N} + \frac{4H(C_2\eta + 3)\iota}{K} + \frac{1}{K} \right) \cdot HSA + \frac{\xi H^2 SA}{N} + \sqrt{\frac{HSA}{K}}$$

$$\asymp \frac{\xi H^2 SA}{N} + \frac{H^2 SA\eta\iota}{K} + \frac{HSA}{K} + \sqrt{\frac{HSA}{K}}, \qquad\qquad\qquad\qquad\qquad \text{(F.9)}$$

where (i) is by $\|A_h\|_\infty, \|V_{h+1}\|_\infty, b_h^\theta(s,a) \leq H$, (ii) comes from concentration $\mathcal{E}$(ii), (iii) comes from Eq.(F.2), and (iv) is by definition of $\mathcal{I}_h$. For the term (I), conditioning on the concentration event $\mathcal{E}$, we have

$$(\text{II}) = \sum_{h \in [H]} \sum_{(s,a) \in \mathcal{I}_h} d_h^\pi(s,a) \cdot \left| r_h^\theta(s,a) - \widehat{r}_h^\theta(s,a) \right|$$

$$\leq \sum_{h \in [H]} \sum_{(s,a) \in \mathcal{I}_h} d_h^\pi(s,a) \cdot \left| \left[ (\mathbb{P}_h - \widehat{\mathbb{P}}_h) V_{h+1} \right](s,a) - b_h^\theta(s,a) \right|$$

$$\leq 2 \sum_{h \in [H]} \sum_{(s,a) \in \mathcal{I}_h} d_h^\pi(s,a) \cdot b_h^\theta(s,a)$$

$$\leq \sum_{h \in [H]} \sum_{(s,a) \in \mathcal{I}_h} \left( 4\widehat{d}_h^\pi(s,a) + 4e_h^\pi(s,a) + \frac{\xi}{2N} \right) \cdot b_h^\theta(s,a)$$

$$\lesssim \sum_{h \in [H]} \sum_{(s,a) \in \mathcal{I}_h} \widehat{d}_h^\pi(s,a) \cdot b_h^\theta(s,a) + H \cdot \sum_{h \in [H]} \sum_{(s,a) \in \mathcal{I}_h} \left( \frac{\xi}{N} + e_h^\pi(s,a) \right)$$

$$\lesssim \frac{\xi H^2 SA}{N} + \sqrt{\frac{HSA}{K}} + \sum_{h \in [H]} \sum_{(s,a) \in \mathcal{I}_h} \widehat{d}_h^\pi(s,a) \cdot b_h^\theta(s,a), \qquad\qquad\qquad \text{(F.10)}$$

where the second line is by construction of Algorithm 1, the second last line is by $b_h^\theta(s,a) \lesssim H$, the last follows from (F.2). Further, we decompose the second term of Eq.(F.10) for any $\theta \in \Theta$ by

$$\sum_{h \in [H]} \sum_{(s,a) \in \mathcal{I}_h} \widehat{d}_h^\pi(s,a) \cdot b_h^\theta(s,a)$$

$$
= \sum_{h \in [H]} \sum_{(s,a) \in \mathcal{I}_h} \widehat{d}_h^\pi(s,a) \cdot \min \left\{ \sqrt{\frac{\log \mathcal{N}(\Theta; \epsilon/H) \iota}{\widehat{N}_h^b(s,a) \vee 1}} \left[ \widehat{\mathbb{V}}_h V_{h+1} \right](s,a) + \frac{H \log \mathcal{N}(\Theta; \epsilon/H) \iota}{\widehat{N}_h^b(s,a) \vee 1} \right.
$$

$$
\left. + \frac{\epsilon}{H} \left( 1 + \sqrt{\frac{\log \mathcal{N}(\Theta; \epsilon/H) \iota}{\widehat{N}_h^b(s,a) \vee 1}} \right), H \right\}
$$

$$
\overset{(i)}{\le} \sum_{h \in [H]} \sum_{(s,a) \in \mathcal{I}_h} \widehat{d}_h^\pi(s,a) \cdot \left\{ \min \left\{ \sqrt{\frac{\log \mathcal{N}(\Theta; \epsilon/H) \iota}{\widehat{N}_h^b(s,a) \vee 1}} \left[ \widehat{\mathbb{V}}_h V_{h+1} \right](s,a), H \right\} \right.
$$

$$
\left. + \frac{H \log \mathcal{N}(\Theta; \epsilon/H) \iota}{\widehat{N}_h^b(s,a) \vee 1} + \frac{\epsilon}{H} \left( 1 + \sqrt{\frac{\log \mathcal{N}(\Theta; \epsilon/H) \iota}{\widehat{N}_h^b(s,a) \vee 1}} \right) \right\}
$$

$$
\overset{(ii)}{\le} \sum_{h \in [H]} \sum_{(s,a) \in \mathcal{I}_h} \widehat{d}_h^\pi(s,a) \cdot \left\{ \sqrt{\frac{\log \mathcal{N}(\Theta; \epsilon/H) \iota \left[ \widehat{\mathbb{V}}_h V_{h+1} \right](s,a) + H}{\widehat{N}_h^b(s,a) \vee 1 + 1/H}} + \frac{H \log \mathcal{N}(\Theta; \epsilon/H) \iota}{\widehat{N}_h^b(s,a) \vee 1} \right.
$$

$$
\left. + \frac{\epsilon}{H} \left( 1 + \sqrt{\frac{\log \mathcal{N}(\Theta; \epsilon/H) \iota}{\widehat{N}_h^b(s,a) \vee 1}} \right) \right\}
$$

$$
\overset{(iii)}{=} \underbrace{\sum_{h \in [H]} \sum_{(s,a) \in \mathcal{I}_h} \widehat{d}_h^\pi(s,a) \cdot \sqrt{\frac{\log \mathcal{N}(\Theta; \epsilon/H) \iota \left[ \widehat{\mathbb{V}}_h V_{h+1} \right](s,a) + H}{K \mathbb{E}_{\pi' \sim \mu^b} \left[ \widehat{d}_h^{\pi'}(s,a) \right] + 1/H}}}_{(\mathrm{II.a})}
$$

$$
+ \underbrace{\sum_{h \in [H]} \sum_{(s,a) \in \mathcal{I}_h} \widehat{d}_h^\pi(s,a) \cdot \frac{H \log \mathcal{N}(\Theta; \epsilon/H) \iota}{K \mathbb{E}_{\pi' \sim \mu^b} \left[ \widehat{d}_h^{\pi'}(s,a) \right] + 1/H}}_{(\mathrm{II.b})}
$$

$$
+ \underbrace{\frac{\epsilon}{H} \sum_{h \in [H]} \sum_{(s,a) \in \mathcal{I}_h} \widehat{d}_h^\pi(s,a) \cdot \left( 1 + \sqrt{\frac{\log \mathcal{N}(\Theta; \epsilon/H) \iota}{K \mathbb{E}_{\pi' \sim \mu^b} \left[ \widehat{d}_h^{\pi'}(s,a) \right] + 1/H}} \right)}_{(\mathrm{II.c})} \tag{F.11}
$$

where the (i) is by inequality $\min \{a + b, c\} \le \min \{a, c\} + b \, (a, b, c \ge 0)$, (ii) comes from inequality $\min \left\{ \frac{x}{y}, \frac{z}{w} \right\} \le \frac{x+z}{y+w}$ and (iii) is valid since

$$
\widehat{N}_h^b(s,a) = \left[ \frac{K}{4}, \mathbb{E}_{\pi \sim \mu^b} [\widehat{d}_h^\pi(s,a)] - \frac{K \xi}{8N} - 3 \log \frac{HSA}{\delta} \right]_+ \gtrsim K \mathbb{E}_{\pi' \sim \mu^b} \left[ \widehat{d}_h^{\pi'}(s,a) \right] + 1/H
$$

holds for all $(s,a) \in \mathcal{I}_h$ according to definition of $\mathcal{I}$. We then study the three terms separately. For the term (II.a), by the Cauchy-Schwarz inequality, we have

$$
(\mathrm{II.a}) \le \underbrace{\left\{ \sum_{h \in [H]} \sum_{(s,a) \in \mathcal{S} \times \mathcal{A}} \widehat{d}_h^\pi(s,a) \cdot [\log \mathcal{N}(\Theta; \epsilon/H) \iota [\mathbb{V}_h V_{h+1}](s,a) + H] \right\}^{1/2}}_{(\mathrm{II.a.1})}
$$

$$
\times \underbrace{\left\{ \sum_{h \in [H]} \sum_{(s,a) \in \mathcal{S} \times \mathcal{A}} \frac{\widehat{d}_h^\pi(s,a)}{K \mathbb{E}_{\pi' \sim \mu^b} \left[ \widehat{d}_h^{\pi'}(s,a) \right] + 1/H} \right\}^{1/2}}_{(\mathrm{II.a.2})}.
$$

Observe that $\|V_{h+1}\|_\infty \le H$, then the term (II.a.1) can be upper bounded by

$$
(\mathrm{II.a.1}) = \sqrt{\sum_{h \in [H]} \sum_{(s,a) \in \mathcal{S} \times \mathcal{A}} \widehat{d}_h^\pi(s,a) \cdot [\log \mathcal{N}(\Theta; \epsilon/H) \iota [\mathbb{V}_h V_{h+1}](s,a) + H]}
$$

$$\leq \sqrt{[H^2 \log \mathcal{N}(\Theta; \epsilon/H)\iota + H] \cdot \sqrt{\sum_{h\in[H]} \sum_{(s,a)\in\mathcal{S}\times\mathcal{A}} \widehat{d_h^\pi}(s,a)}} \asymp \sqrt{H^3 \log \mathcal{N}(\Theta; \epsilon/H)\iota}.$$

(F.12)

For the term (II.a.2), we have

$$(\text{II.a.2}) = \sqrt{\sum_{h\in[H]} \sum_{(s,a)\in\mathcal{S}\times\mathcal{A}} \frac{\widehat{d_h^\pi}(s,a)}{K \mathbb{E}_{\pi'\sim\mu^\mathrm{b}}\left[\widehat{d_h^{\pi'}}(s,a)\right] + 1/H}}$$

$$= \sqrt{\frac{1}{K} \cdot \sum_{h\in[H]} \sum_{(s,a)\in\mathcal{S}\times\mathcal{A}} \frac{\widehat{d_h^\pi}(s,a)}{\mathbb{E}_{\pi'\sim\mu^\mathrm{b}}\left[\widehat{d_h^{\pi'}}(s,a)\right] + 1/KH}}$$

$$\lesssim \sqrt{\frac{HSA}{K}},$$

(F.13)

which the last line comes from Eq.(B.3). Combining Eq.(F.12) and (F.13), we conclude that

$$(\text{II.a}) \lesssim \sqrt{\frac{H^4 SA}{K}}.$$

(F.14)

For the term (II.b), by Eq.(B.3), we have

$$(\text{II.b}) = \sum_{h\in[H]} \sum_{(s,a)\in\mathcal{S}\times\mathcal{A}} \widehat{d_h^\pi}(s,a) \cdot \frac{H \log \mathcal{N}(\Theta; \epsilon/H)\iota}{K \mathbb{E}_{\pi'\sim\mu^\mathrm{b}}\left[\widehat{d_h^{\pi'}}(s,a)\right] + 1/H}$$

$$= \frac{1}{K} \cdot \sum_{h\in[H]} \sum_{(s,a)\in\mathcal{S}\times\mathcal{A}} \widehat{d_h^\pi}(s,a) \cdot \frac{H \log \mathcal{N}(\Theta; \epsilon/H)\iota}{\mathbb{E}_{\pi'\sim\mu^\mathrm{b}}\left[\widehat{d_h^{\pi'}}(s,a)\right] + 1/KH}$$

$$\lesssim \frac{H^2 SA \log \mathcal{N}(\Theta; \epsilon/H)\iota}{K}.$$

(F.15)

For the term (II.c), we have

$$(\text{II.c}) = \frac{\epsilon}{H} \sum_{h\in[H]} \sum_{(s,a)\in\mathcal{I}_h} \widehat{d_h^\pi}(s,a) \cdot \left(1 + \sqrt{\frac{\log \mathcal{N}(\Theta; \epsilon/H)\iota}{K \mathbb{E}_{\pi'\sim\mu^\mathrm{b}}\left[\widehat{d_h^{\pi'}}(s,a)\right] + 1/H}}\right)$$

$$= \epsilon + \frac{\epsilon}{H} \sum_{h\in[H]} \sum_{(s,a)\in\mathcal{I}_h} \sqrt{\widehat{d_h^\pi}(s,a)} \cdot \sqrt{\frac{\widehat{d_h^\pi}(s,a) \log \mathcal{N}(\Theta; \epsilon/H)\iota}{K \mathbb{E}_{\pi'\sim\mu^\mathrm{b}}\left[\widehat{d_h^{\pi'}}(s,a)\right] + 1/H}}$$

$$\leq \epsilon + \frac{\epsilon}{H} \sqrt{\sum_{h\in[H]} \sum_{(s,a)\in\mathcal{I}_h} \widehat{d_h^\pi}(s,a)} \cdot \sqrt{\sum_{h\in[H]} \sum_{(s,a)\in\mathcal{I}_h} \frac{\widehat{d_h^\pi}(s,a) \log \mathcal{N}(\Theta; \epsilon/H)\iota}{K \mathbb{E}_{\pi'\sim\mu^\mathrm{b}}\left[\widehat{d_h^{\pi'}}(s,a)\right] + 1/H}}$$

$$\leq \epsilon(1 + \sqrt{\frac{SA \log \mathcal{N}(\Theta; \epsilon/H)\iota}{K}}),$$

(F.16)

where the second last line is by the Cauchy-Schwarz inequality and the last line is due to Eq.(F.13).

Then combining Eq.(F.10), Eq.(F.14), Eq.(F.15), and Eq.(F.16), we obtain the bound for the term (II)

$$(\text{II}) \lesssim (\text{II.a}) + (\text{II.b}) + (\text{II.c})$$

$$\lesssim \sqrt{\frac{H^4 SA \log \mathcal{N}(\Theta; \epsilon/H)\iota}{K}} + \frac{H^2 SA \log \mathcal{N}(\Theta; \epsilon/H)\iota}{K} + \epsilon(1 + \sqrt{\frac{SA \log \mathcal{N}(\Theta; \epsilon/H)\iota}{K}})$$

$$\lesssim \sqrt{\frac{H^4 SA \log \mathcal{N}(\Theta; \epsilon/H)\iota}{K}} + \epsilon,$$

(F.17)

where the last line is from $\epsilon < 1$. Finally, combining Eq.(F.9) and (F.14), we get the final bound

$$D_\Theta^{\mathsf{all}}\left(\mathscr{R}^\star, \widehat{\mathscr{R}}\right) = \sup_{\pi,\theta\in\Theta} d^\pi\left(r_h^\theta, \widehat{r}_h^\theta\right) \leq \mathrm{I} + \mathrm{II}$$

$$\lesssim \frac{\xi H^2 SA}{N} + \frac{H^2 SA \eta \iota}{K} + \sqrt{\frac{H^4 SA \log \mathcal{N}(\Theta; \epsilon/H)\iota}{K}} + \epsilon.$$

Hence, we can guarantee $D_\Theta^{\mathsf{all}}\left(\mathscr{R}^\star, \widehat{\mathscr{R}}\right) \leq 2\epsilon$, provided that

$$K \geq \widetilde{\mathcal{O}}\left( \frac{H^4 SA \log \mathcal{N}(\Theta; \epsilon/H)}{\epsilon^2} + \frac{H^2 SA(\eta + \log \mathcal{N}(\Theta; \epsilon/H))}{\epsilon} \right), \qquad KH \geq N \geq \widetilde{\mathcal{O}}\left( \sqrt{H^9 S^7 A^7 K} \right).$$

Here $poly \log (H, S, A, 1/\delta)$ are omitted.

## G   LOWER BOUND IN THE ONLINE SETTING

### G.1   LOWER BOUND OF ONLINE IRL PROBLEMS

We focus on the cases where $\Theta = \overline{\mathcal{V}} \times \overline{\mathcal{A}}$. In this case $\log \mathcal{N}(\Theta; \epsilon/H) = \widetilde{\mathcal{O}}(S)$, the upper bound of the sample complexity of Algorithm 6 becomes $\widetilde{\mathcal{O}}\left(H^4 S^2 A/\epsilon^2\right)$ (we hide the burn-in term).

Similar to the offline setting, we define $(\epsilon, \delta)$-*PAC algorithm for online IRL problems* for all $\epsilon, \delta \in (0, 1)$ as follows.

**Definition G.1.** *Fix a parameter set $\Theta$, we say an online IRL algorithm $\mathfrak{A}$ is a $(\epsilon, \delta)$-PAC algorithm for online IRL problems, if for any IRL problem $(\mathcal{M}, \pi^{\mathsf{E}})$, with probability $1 - \delta$, $\mathfrak{A}$ outputs a reward mapping $\widehat{\mathscr{R}}$ such that*

$$D_\Theta^{\mathsf{all}}(\widehat{\mathscr{R}}, \mathscr{R}^\star) \leq \epsilon.$$

**Theorem G.2** (Lower bound for online IRL problems)**.** *Fix parameter set $\Theta = \overline{\mathcal{V}} \times \overline{\mathcal{A}}$ and let $\mathfrak{A}$ be an $(\epsilon, \delta)$-PAC algorithm for online IRL problems, where $\delta \leq 1/3$. Then, there exists an IRL problem $(\mathcal{M}, \pi^{\mathsf{E}})$ such that, if $H \geq 4, S \geq 130, A \geq 2$, there exists an absolute constant $c_0$ such that the expected sample complexity $N$ is lower bounded by*

$$N \geq \frac{c_0 H^3 SA \min\{S, A\}}{\epsilon^2},$$

*where $0 < \epsilon \leq (H - 2)/1024$;*

Note that when $S \leq A$, the lower bound scales with $\Omega\left(S^2 A\right)$, matching the $S^2 A$ factor dependence observed in the upper bound (Theorem 5.1).

### G.2   HARD INSTANCE CONSTRUCTION

**Hard Instance Construction**   Our construction is a modification of the hard instance constructed in the proof of Metelli et al. (2023, Theorem B.3). We construct the hard instance with $2S + 1$ states, $A + 1$ actions, and $2H + 2$ stages for any $H$, $S$, $A > 0$. (This rescaling only affects $S$, $H$ by at most a multiplicative constant and thus does not affect our result). We then define an integer $K$ by

$$K := \min\{S, A\}.$$

Each MDP $\mathcal{M}_{\mathbf{v}}$ is indexed by a vector $\mathbf{w} = \left( w_h^{(i,j,k)} \right)_{h \in [H], i \in [K], j \in [S], k \in [K]} \in \mathbb{R}^{HSKA}$ and is specified as follows:

- State space: $\mathcal{S} = \left\{ s_{\mathsf{start}}, s_{\mathsf{root}}, s_1, \ldots, s_S, \bar{s}_1, \ldots, \bar{s}_S \right\}$.
- Action space: $\mathcal{A} = \{a_0, a_1, ..., a_A\}$.
- Initial state: $s_{\mathsf{start}}$, that is
$$\mathbb{P}(s_1 = s_{\mathsf{start}}) = 1.$$
- Transitions:

– At stage 1, $s_{\text{start}}$ can only transition to itself or $s_i$. The transition probabilities are given by

$$
\begin{cases}
\mathbb{P}_1(s_{\text{start}} \mid s_{\text{start}}, a_0) = 1 \\
\mathbb{P}_1(s_i \mid s_{\text{start}}, a_i) = 1 & \text{for all } i \in [K], \\
\mathbb{P}_1(s_j \mid s_{\text{start}}, a_k) = \frac{1}{S} & \text{for all } j \in [S],\ k \geq K+1,
\end{cases}
$$

– At each stage $h \in \{2, \ldots, H+1\}$, $s_{\text{start}}$ can only transition to itself or $s_i$, $s_i$ can only transition to absorbing state $\bar{s}_j$. The transition probabilities are given by

$$
\begin{cases}
\mathbb{P}_h(s_{\text{start}} \mid s_{\text{start}}, a_0) = 1, \\
\mathbb{P}_h(s_i \mid s_{\text{start}}, a_i) = 1 & \text{for all } i \in [K], \\
\mathbb{P}_h(s_j \mid s_{\text{start}}, a_k) = \frac{1}{S} & \text{for all } j \in [S],\ k \geq K+1, \\
\mathbb{P}_h(\bar{s}_j \mid s_i, a_0) = \frac{1}{S} & \text{for all } i \geq K+1,\ j \in [S], \\
\mathbb{P}_h(\bar{s}_j \mid s_i, a_k) = \frac{1+\epsilon' \cdot w_{h-1}^{(i,j,k)}}{S} & \text{for all } i \in [K],\ j \in [S],\ k \in [A], \\
\mathbb{P}_h(\bar{s}_j \mid \bar{s}_j, a_k) = 1 & \text{for all } j \in [S],\ k \geq 0.
\end{cases}
\tag{G.1}
$$

– At each stage $h \in \{H+1, \ldots, 2H+2\}$ and $s_{\text{start}}$ can only transition to $s_i$ and $s_i$ can only transition to absorbing state $\bar{s}_j$. The transition probabilities are given by

$$
\begin{cases}
\mathbb{P}_h(s_i \mid s_{\text{start}}, a_0) = \frac{1}{S} & \text{for all } i \in [S], \\
\mathbb{P}_h(s_i \mid s_{\text{start}}, a_i) = 1 & \text{for } i \in [K], \\
\mathbb{P}_h(s_j \mid s_{\text{start}}, a_k) = \frac{1}{S} & \text{for all } j \in [S],\ k \geq K+1, \\
\mathbb{P}_h(\bar{s}_j \mid s_i, a_k) = \frac{1}{S} & \text{for all } i \in [K],\ j \in [S],\ k \geq 0, \\
\mathbb{P}_h(\bar{s}_j \mid \bar{s}_j, a_k) = 1 & \text{for all } i \in [S],\ k \geq 0.
\end{cases}
$$

• Expert policy: expert policy $\pi^{\mathsf{E}}$ plays action $a_0$ at every stage $h \in [H]$ and state $s \in \mathcal{S}$. That is

$$
\pi_h^{\mathsf{E}}(a_0 | s) = 1, \qquad \text{for all } h \in [2H+2],\ s \in \mathcal{S}.
\tag{G.2}
$$

To ensure the definition of $\mathcal{M}_{\mathbf{w}}$ is valid, we enforce the following condition:

$$
\sum_{j \in [S]} w_h^{(i,j,k)} = 0,
$$

for any $h \in [H]$, $i \in [K]$, $k \in [A]$. We define a vector space $\mathcal{W}$ by

$$
\mathcal{W} := \left\{ w = (w_j)_{j \in [S]} \in \{1, -1\}^S : \sum_{j \in [S]} w_j = 0 \right\}.
$$

Let $\mathcal{I}$ denote $[H] \times [K] \times [A]$, the Eq.(G.2) is equivalent to

$$
\mathbf{w} \in \mathcal{W}^{\mathcal{I}}.
$$

Further, we let $\mathbb{P}^{(\mathbf{w})} = \left\{ \mathbb{P}_h^{(\mathbf{w})} \right\}_{h \in [H]}$ to be the transition kernel of MDP\R $\mathcal{M}_{\mathbf{w}}$. In addition, Given $\mathbf{w} \in \mathcal{W}^{\mathcal{I}}$, $w \in \mathcal{W}$ and index $a \in \mathcal{I}$, we use the notation $\mathbf{w} \overset{a}{\leftarrow} w$ to represent vector obtained by replacing $a$ component of $\mathbf{w}$ with $w$. For example, let $\mathbf{w} = (w_h^{(i,j,k)})_{h \in [H], i \in [K], j \in [S], k \in [K]}$, $w = (w_j)_{j \in [S]}$, $a = (h_a, i_a, j_a)$ and $\overline{\mathbf{w}} = \mathbf{w} \overset{a}{\leftarrow} w$ and then $\overline{\mathbf{w}}$ can be expressed as follows:

$$
\overline{w}_h^{(i,j,k)} = \begin{cases} w_j & (h, i, k) = (h_a, i_a, k_a), \\ w_h^{(i,j,k)} & \text{otherwise.} \end{cases}
\tag{G.3}
$$

By Metelli et al. (2023, Lemma E.6), there exists a $\overline{\mathcal{W}} \subseteq \mathcal{W}$ such that

$$
\sum_{i \in [n]} (w_i - v_i)^2 \geq \frac{S}{8}, \qquad \forall v, w \in \overline{\mathcal{W}}, \qquad \log |\overline{\mathcal{W}}| \geq \frac{S}{10}.
\tag{G.4}
$$

**Notations.** To distinguish with different MDP\Rs, we denote $V_h^\pi\big(\cdot; r, \mathbb{P}^{(\mathbf{w})}\big)$ be the value function of $\pi$ in MDP $\mathcal{M}_\mathbf{w} \cup r$. Given two rewards $r$ $r'$, we define $d^{\mathsf{all}}(r, r'; \mathbb{P}^{(\mathbf{w})})$ to be the $d^{\mathsf{all}}$ metric evaluated in $\mathcal{M}_\mathbf{w}$:

$$d^{\mathsf{all}}(r, r'; \mathbb{P}^{(\mathbf{w})}) := \sup_{\pi, h \in [H]} \mathbb{E}_{\mathbb{P}^{(\mathbf{w})}, \pi} \left| V_h^\pi(s_h; r, \mathbb{P}^{(\mathbf{w})}) - V_h^\pi(s_h; r', \mathbb{P}^{(\mathbf{w})}) \right|.$$

Correspondingly, given a parameter set $\Theta$, two reward mappings $\mathscr{R}$, $\mathscr{R}'$, we can define $D_\Theta^{\mathsf{all}}(\mathscr{R}, \mathscr{R}'; \mathbb{P}^{(\mathbf{w})})$ by

$$D_\Theta^{\mathsf{all}}(\mathscr{R}, \mathscr{R}'; \mathbb{P}^{(\mathbf{w})}) := \sup_{(V, A) \in \Theta} d^{\mathsf{all}}\Big(\mathscr{R}(V, A), \mathscr{R}'(V, A); \mathbb{P}^{(\mathbf{w})}\Big).$$

In the following, we always assume that $\mathbf{w} \in \bar{\mathcal{W}}$. We then present the following lemma which shows the difference between two MDP\Rs $\mathcal{M}_{\mathbf{w} \overset{a}{\leftarrow} v}$ and $\mathcal{M}_{\mathbf{w} \overset{a}{\leftarrow} w}$ for any $\mathbf{w} \in \overline{\mathcal{W}}^\mathcal{I}$ and $v \neq w \in \overline{\mathcal{W}}$.

**Lemma G.3.** *Given any* $\mathbf{w} \in \overline{\mathcal{W}}^\mathcal{I}$, $w \neq v \in \overline{\mathcal{W}}$, *and index* $a = (h_a, i_a, k_a) \in \mathcal{I}$, *let* $\mathscr{R}^{(\mathbf{w} \overset{a}{\leftarrow} w)}$, $\mathscr{R}^{(\mathbf{w} \overset{a}{\leftarrow} v)}$ *be the ground truth reward mapping induced by* $\mathcal{M}_{\mathbf{w} \overset{a}{\leftarrow} w}$, $\mathcal{M}_{\mathbf{w} \overset{a}{\leftarrow} v}$, *respectively. Set* $\Theta = \overline{\mathcal{V}} \times \overline{\mathcal{A}}$. *For any* $\epsilon' \in (0, 1/2]$ *and any reward mapping* $\mathscr{R} : \overline{\mathcal{V}} \times \overline{\mathcal{A}} \to \mathcal{R}^{\mathsf{all}}$, *we have*

$$7D_\Theta^{\mathsf{all}}\Big(\mathscr{R}^{(\mathbf{w} \overset{a}{\leftarrow} w)}, \mathscr{R}; \mathbb{P}^{(\mathbf{w} \overset{a}{\leftarrow} w)}\Big) + D_\Theta^{\mathsf{all}}\Big(\mathscr{R}^{(\mathbf{w} \overset{a}{\leftarrow} v)}, \mathscr{R}; \mathbb{P}^{(\mathbf{w} \overset{a}{\leftarrow} v)}\Big) \geq \frac{H\epsilon'}{16},$$

*where* $\epsilon'$ *is specified in Eq.*(G.1).

*Proof.* **Step 1: Construct bad parameter** $(V^{\mathsf{bad}}, A^{\mathsf{bad}})$. We construct the bad parameter $(V^{\mathsf{bad}}, A^{\mathsf{bad}}) \in \overline{\mathcal{V}} \times \overline{\mathcal{A}}$ as follows:

- We set $A_h^{\mathsf{bad}}(s, a) = 0$ for all $(h, s, a) \in [2H + 2] \times \mathcal{S} \times \mathcal{A}$.

- We set $V_h^{\mathsf{bad}}$ by

$$V_h^{\mathsf{bad}}(s) := \begin{cases} \frac{(2H+2-h) \cdot (w_i - v_i)}{2} & \text{if } s = \bar{s}_i, h = h_a + 2, \\ 0 & \text{other.} \end{cases} \tag{G.5}$$

Directly by the construction of $(V^{\mathsf{bad}}, A^{\mathsf{bad}})$, we obtain that

$$\sum_{i \in [S]} (w_i - v_i) \cdot V_h^{\mathsf{bad}}(\bar{s}_i) = \sum_{i \in [S]} \frac{(2H - h_a)(w_i - v_i)^2}{2} \geq \frac{H(w_i - v_i)^2}{2} \geq \frac{HS}{16}, \tag{G.6}$$

where the last inequality is due to Eq.(G.4). We then denote $\mathscr{R}^{(\mathbf{w} \overset{a}{\leftarrow} w)}(V^{\mathsf{bad}}, A^{\mathsf{bad}})$, $\mathscr{R}^{(\mathbf{w} \overset{a}{\leftarrow} v)}(V^{\mathsf{bad}}, A^{\mathsf{bad}})$ as $r_w^{\mathsf{bad}}$, $r_v^{\mathsf{bad}}$, respectively.

Since $A^{\mathsf{bad}} \equiv \mathbf{0}$, any policy $\pi \in \Pi^\star_{\mathcal{M}_{\mathbf{w} \overset{a}{\leftarrow} w} \cup r_w^{\mathsf{bad}}}, \Pi^\star_{\mathcal{M}_{\mathbf{w} \overset{a}{\leftarrow} v} \cup r_v^{\mathsf{bad}}}$. More explicitly, any policy is optimal in $\mathcal{M}_{\mathbf{w} \overset{a}{\leftarrow} w} \cup r_w^{\mathsf{bad}}$ and $\mathcal{M}_{\mathbf{w} \overset{a}{\leftarrow} v} \cup r_v^{\mathsf{bad}}$.

**Step 2: Construct test policies** $\pi^{\mathsf{test},(1)}$, $\pi^{\mathsf{test},(2)}$. Let $r = \mathscr{R}(V^{\mathsf{bad}}, A^{\mathsf{bad}})$. Let $\pi^{\mathsf{g}}$ be the greedy policy of $\mathcal{M}_{\mathbf{w} \overset{a}{\leftarrow} w} \cup r$. By Lemma 2.2, there exist a pair $(V, A) \in \overline{\mathcal{V}} \times \overline{\mathcal{A}}$ such that

$$r_h(s, a) = -A_h(s, a) \cdot \mathbf{1}\{a \notin \mathrm{supp}(\pi_h^{\mathsf{g}}(\cdot \mid s))\} + V_h(s) - \left[\mathbb{P}_h^{(\mathbf{w} \overset{a}{\leftarrow} w)} V_{h+1}\right](s, a), \tag{G.7}$$

We then construct test policy $\pi^{\mathsf{test},(1)}$ by

$$\begin{cases} \pi_h^{\mathsf{test},(1)}(a_0 \mid s_{\mathsf{start}}) = 1 & h \leq h_a - 1 \\ \pi_h^{\mathsf{test},(1)}(a_{i_a} \mid s_{\mathsf{start}}) = 1 & h = h_a \\ \pi_h^{\mathsf{test},(1)}(a_{k_a} \mid s_{i_a}) = 1 & h = h_a + 1 \\ \pi_h^{\mathsf{test},(1)} = \pi_h^{\mathsf{g}} & h \geq h_a + 2 \end{cases}$$

which implies that at stage $h \leq h_a - 1$, $\pi^{\text{test},(1)}$ always plays $a_0$, at stage $h_a$, $\pi^{\text{test},(1)}$ plays $a_{i_a}$, then transition to $s_{i_a}$, at stage $h_a + 1$, $\pi^{\text{test},(1)}$ plays $a_{k_a}$, then at stage $h \geq h_a + 2$, $\pi^{\text{test},(1)}$ is equal to the greedy policy $\pi^{\text{g}}$. By construction, we can conclude that

$$d_{h_a+1}^{\pi^{\text{test},(1)}}\left(s_{i_a}; \mathbb{P}^{\left(\mathbf{w} \overset{a}{\leftarrow} w\right)}\right) = 1, \qquad V_{h_a+2}^{\pi^{\text{test},(1)}}\left(\cdot \mid r, \mathbb{P}^{\left(\mathbf{w} \overset{a}{\leftarrow} w\right)}\right) = V_{h_a+2}(\cdot)., \tag{G.8}$$

the second equality is due to $\pi_h^{\text{test},(1)} = \pi_h^{\text{g}}$ for any $h \geq h_a + 2$.

Further, we have

$$V_{h_a+1}^{\pi^{\text{test},(1)}}\left(s_{i_a}; r, \mathbb{P}^{\left(\mathbf{w} \overset{a}{\leftarrow} w\right)}\right) = r_{h_a+1}(s_{i_a}, a_{k_a}) + \left[\mathbb{P}_{h_a+1}^{\left(\mathbf{w} \overset{a}{\leftarrow} w\right)} V_{h_a+2}^{\pi^{\text{test},(1)}}\left(\cdot \mid r, \mathbb{P}^{\left(\mathbf{w} \overset{a}{\leftarrow} w\right)}\right)\right](s_{i_a}, a_{k_a})$$

$$= -A_{h_a+1}(s_{i_a}, a_{k_a}) \cdot \mathbf{1}\left\{a_{k_a} \notin \text{supp}\left(\pi_{h_a+1}^{\text{g}}(\cdot \mid s_{i_a})\right)\right\}$$

$$+ V_{h_a+1}(s) - \left[\mathbb{P}_{h_a+1}^{\left(\mathbf{w} \overset{a}{\leftarrow} w\right)} V_{h_a+2}\right](s_{i_a}, a_{k_a}) + \left[\mathbb{P}_{h_a+1}^{\left(\mathbf{w} \overset{a}{\leftarrow} w\right)} V_{h_a+2}\right]$$

$$= V_{h_a+1}(s_{i_a}) - \text{gap}, \tag{G.9}$$

where the first line is by the Bellman equation, the second line is due to Eq.(G.7) and Eq.(G.8). Here gap is the advantage function at $(h_a + 1, s_{i_a}, a_{k_a})$, i.e, $\text{gap} := A_{h_a+1}(s_{i_a}, a_{k_a}) \cdot \mathbf{1}\left\{a_{k_a} \in \text{supp}\left(\pi_{h_a+1}^{\text{g}}(\cdot \mid s_{i_a})\right)\right\}$. Then by definition of $D_{\Theta}^{\text{all}}(\mathscr{R}^{\left(\mathbf{w} \overset{a}{\leftarrow} w\right)}, \mathscr{R}; \mathbb{P}^{\left(\mathbf{w} \overset{a}{\leftarrow} w\right)})$, we can obtain that

$$D_{\Theta}^{\text{all}}\left(\mathscr{R}^{\left(\mathbf{w} \overset{a}{\leftarrow} w\right)}, \mathscr{R}; \mathbb{P}^{\left(\mathbf{w} \overset{a}{\leftarrow} w\right)}\right)$$

$$\geq d^{\text{all}}\left(\mathscr{R}^{\left(\mathbf{w} \overset{a}{\leftarrow} w\right)}(V^{\text{bad}}, A^{\text{bad}}), \mathscr{R}(V^{\text{bad}}, A^{\text{bad}}); \mathbb{P}^{\left(\mathbf{w} \overset{a}{\leftarrow} w\right)}\right)$$

$$= d^{\text{all}}\left(r_w^{\text{bad}}, r; \mathbb{P}^{\left(\mathbf{w} \overset{a}{\leftarrow} w\right)}\right)$$

$$\geq \mathbb{E}_{\mathbb{P}^{\left(\mathbf{w} \overset{a}{\leftarrow} w\right)}, \pi^{\text{test},(1)}}\left|V_{h_a+1}^{\pi^{\text{test},(1)}}\left(s; r_w^{\text{bad}}, \mathbb{P}^{\left(\mathbf{w} \overset{a}{\leftarrow} w\right)}\right) - V_{h_a+1}^{\pi^{\text{test},(1)}}\left(s; r, \mathbb{P}^{\left(\mathbf{w} \overset{a}{\leftarrow} w\right)}\right)\right|$$

$$= \left|V_{h_a+1}^{\pi^{\text{test},(1)}}\left(s_{i_a}; r_w^{\text{bad}}, \mathbb{P}^{\left(\mathbf{w} \overset{a}{\leftarrow} w\right)}\right) - V_{h_a+1}^{\pi^{\text{test},(1)}}\left(s_{i_a}; r, \mathbb{P}^{\left(\mathbf{w} \overset{a}{\leftarrow} w\right)}\right)\right|$$

$$= \left|V_{h_a+1}^{\text{bad}}(s_{i_a}) - V_{h_a+1}(s_{i_a}) + \text{gap}\right|, \tag{G.10}$$

where the second last line is due to Eq.(G.8) and the last line is by Eq.(G.9) and $\pi^{\text{test},(1)} \in \Pi_{\mathcal{M}_{\mathbf{w} \overset{a}{\leftarrow} w} \cup r_w^{\text{bad}}}^{\star}$: $V_{h_a+1}^{\pi^{\text{test},(1)}}\left(s_{i_a}; r_w^{\text{bad}}, \mathbb{P}^{\left(\mathbf{w} \overset{a}{\leftarrow} w\right)}\right) = V_{h_a+1}^{\text{bad}}(s_{i_a})$.

Next, we construct another test policy $\pi^{\text{test},(2)}$ as follows:

$$\begin{cases} \pi_h^{\text{test},(2)}(a_0 \mid s_{\text{start}}) = 1 & h \leq h_a - 1 \\ \pi_h^{\text{test},(2)}(a_{i_a} \mid s_{\text{start}}) = 1 & h = h_a \\ \pi_h^{\text{test},(2)} = \pi_h^{\text{g}} & h \geq h_a + 1. \end{cases}$$

The difference between $\pi^{\text{test},(2)}$ and $\pi^{\text{test},(1)}$ is that at stage $h_a$ $\pi^{\text{test},(2)}$ play the $\pi_{h_a+1}^{\text{g}}(s_{i_a})$ instead of $a_{k_a}$. Similar to Eq.(G.8), we have

$$d_{h_a+1}^{\pi^{\text{test},(2)}}\left(s_{i_a}; \mathbb{P}^{\left(\mathbf{w} \overset{a}{\leftarrow} w\right)}\right) = 1, \qquad V_{h_a+1}^{\pi^{\text{test},(2)}}\left(s_{i_a}; r, \mathbb{P}^{\left(\mathbf{w} \overset{a}{\leftarrow} w\right)}\right) = V_{h_a+1}(s_{i_a}) \tag{G.11}$$

where the seconed equality is valid since $\pi_h^{\text{test},(2)} = \pi_h^{\text{g}}$ for any $h \geq h_a + 1$.

Similar to Eq.(G.10), we have

$$D_{\Theta}^{\text{all}}\left(\mathscr{R}^{\left(\mathbf{w} \overset{a}{\leftarrow} w\right)}, \mathscr{R}; \mathbb{P}^{\left(\mathbf{w} \overset{a}{\leftarrow} w\right)}\right) \geq d^{\text{all}}\left(r_w^{\text{bad}}, r; \mathbb{P}^{\left(\mathbf{w} \overset{a}{\leftarrow} w\right)}\right)$$

$$\geq \mathbb{E}_{\mathbb{P}^{\left(\mathbf{w} \overset{a}{\leftarrow} w\right)}, \pi^{\text{test},(2)}}\left|V_{h_a+1}^{\pi^{\text{test},(2)}}\left(s; r_w^{\text{bad}}, \mathbb{P}^{\left(\mathbf{w} \overset{a}{\leftarrow} w\right)}\right) - V_{h_a+1}^{\pi^{\text{test},(2)}}\left(s; r, \mathbb{P}^{\left(\mathbf{w} \overset{a}{\leftarrow} w\right)}\right)\right|$$

$$= \left|V_{h_a+1}^{\pi^{\text{test},(2)}}\left(s_{i_a}; r_w^{\text{bad}}, \mathbb{P}^{\left(\mathbf{w} \overset{a}{\leftarrow} w\right)}\right) - V_{h_a+1}^{\pi^{\text{test},(2)}}\left(s_{i_a}; r, \mathbb{P}^{\left(\mathbf{w} \overset{a}{\leftarrow} w\right)}\right)\right|$$

$$= \left|V_{h_a+1}^{\text{bad}}(s_{i_a}) - V_{h_a+1}(s_{i_a})\right|, \tag{G.12}$$

where the second last is due to Eq.(G.11), the last line follows from $\pi^{\text{test},(2)} \in \Pi^{\star}_{\mathcal{M}_{\mathbf{w}\overset{a}{\leftarrow}w} \cup r^{\text{bad}}_w}$:
$V^{\pi^{\text{test},(2)}}_{h_a+1}(s_{i_a}; r^{\text{bad}}_w, \mathbb{P}^{(\mathbf{w}\overset{a}{\leftarrow}w)}) = V^{\text{bad}}_{h_a+1}(s_{i_a})$. Combing Eq.(G.10) and Eq.(G.12), we have

$$2D^{\text{all}}_{\Theta}\left(\mathscr{R}^{(\mathbf{w}\overset{a}{\leftarrow}w)}, \mathscr{R}; \mathbb{P}^{(\mathbf{w}\overset{a}{\leftarrow}w)}\right) \geq \left|V^{\text{bad}}_{h_a+1}(s_{i_a}) - V_{h_a+1}(s_{i_a})\right| + \left|V^{\text{bad}}_{h_a}(s_{i_a}) - V_{h_a+1}(s_{i_a}) + \text{gap}\right|$$
$$\geq \text{gap}, \tag{G.13}$$

where the second line comes from the triangle inequality.

**Step 3: lower bound** $D^{\text{all}}_{\Theta}\left(\mathscr{R}^{(\mathbf{w}\overset{a}{\leftarrow}v)}, \mathscr{R}; \mathbb{P}^{(\mathbf{w}\overset{a}{\leftarrow}v)}\right)$. We still use the test policy $\pi^{\text{test},(1)}$ in $\mathcal{M}_{(\mathbf{w}\overset{a}{\leftarrow}v)}$.
Since $\mathbb{P}^{(\mathbf{w}\overset{a}{\leftarrow}v)}_h = \mathbb{P}^{(\mathbf{w}\overset{a}{\leftarrow}w)}_h$ for any $h \geq h_a + 2$, we have

$$V^{\pi^{\text{test},(1)}}_{h_a+2}(\bar{s}_i|r, \mathbb{P}^{(\mathbf{w}\overset{a}{\leftarrow}v)}) = V^{\pi^{\text{test},(1)}}_{h_a+2}(\bar{s}_i|r, \mathbb{P}^{(\mathbf{w}\overset{a}{\leftarrow}w)}) = V_{h_a+2}(\bar{s}_i), \qquad \text{for all } i \in [S], \tag{G.14}$$

where the second equality comes from Eq.(G.8).

By the definition of $D^{\text{all}}_{\Theta}\left(\mathscr{R}^{(\mathbf{w}\overset{a}{\leftarrow}v)}, \mathscr{R}; \mathbb{P}^{(\mathbf{w}\overset{a}{\leftarrow}v)}\right)$, we have

$$D^{\text{all}}_{\Theta}\left(\mathscr{R}^{(\mathbf{w}\overset{a}{\leftarrow}v)}, \mathscr{R}; \mathbb{P}^{(\mathbf{w}\overset{a}{\leftarrow}v)}\right)$$
$$\geq d^{\text{all}}\left(r^{\text{bad}}_v, r; \mathbb{P}^{(\mathbf{w}\overset{a}{\leftarrow}v)}\right)$$
$$\geq \mathbb{E}_{\mathbb{P}^{(\mathbf{w}\overset{a}{\leftarrow}v)}, \pi^{\text{test},(1)}} \left|V^{\pi^{\text{test},(1)}}_{h_a+1}(s; r^{\text{bad}}_v, \mathbb{P}^{(\mathbf{w}\overset{a}{\leftarrow}v)}) - V^{\pi^{\text{test},(1)}}_{h_a+1}(s; r, \mathbb{P}^{(\mathbf{w}\overset{a}{\leftarrow}v)})\right|$$
$$= \left|V^{\pi^{\text{test},(1)}}_{h_a+1}(s_{i_a}; r^{\text{bad}}_v, \mathbb{P}^{(\mathbf{w}\overset{a}{\leftarrow}v)}) - V^{\pi^{\text{test},(1)}}_{h_a+1}(s_{i_a}; r, \mathbb{P}^{(\mathbf{w}\overset{a}{\leftarrow}v)})\right| \quad \text{(by construction of policy } \pi^{\text{test},(1)}.)$$
$$= \left|V^{\text{bad}}_{h_a+1}(s_{i_a}) - V^{\pi^{\text{test},(1)}}_{h_a+1}(s_{i_a}; r, \mathbb{P}^{(\mathbf{w}\overset{a}{\leftarrow}v)})\right|$$
$$\overset{(i)}{=} \left|V^{\text{bad}}_{h_a+1}(s_{i_a}) - r_{h_a+1}(s_{i_a}, a_{k_a}) - \left[\mathbb{P}^{(\mathbf{w}\overset{a}{\leftarrow}v)}_{h_a+1} V_{h_a+2}\right](s_{i_a}, a_{k_a})\right|$$
$$= \left|V^{\text{bad}}_{h_a+1}(s_{i_a}) - r_{h_a+1}(s_{i_a}, a_{k_a}) - \left[\mathbb{P}^{(\mathbf{w}\overset{a}{\leftarrow}w)}_{h_a+1} V_{h_a+2}\right](s_{i_a}, a_{k_a}) - \left[\left(\mathbb{P}^{(\mathbf{w}\overset{a}{\leftarrow}v)}_{h_a+1} - \mathbb{P}^{(\mathbf{w}\overset{a}{\leftarrow}w)}_{h_a+1}\right) V_{h_a+2}\right](s_{i_a}, a_{k_a})\right|$$
$$\geq \left|\left[\left(\mathbb{P}^{(\mathbf{w}\overset{a}{\leftarrow}v)}_{h_a+1} - \mathbb{P}^{(\mathbf{w}\overset{a}{\leftarrow}w)}_{h_a+1}\right) V_{h_a+2}\right](s_{i_a}, a_{k_a})\right| - \left|V^{\text{bad}}_{h_a+1}(s_{i_a}) - r_{h_a+1}(s_{i_a}, a_{k_a}) - \left[\mathbb{P}^{(\mathbf{w}\overset{a}{\leftarrow}w)}_{h_a+1} V_{h_a+2}\right](s_{i_a}, a_{k_a})\right|$$
$$\text{(by triangle inequality)}$$
$$\overset{(ii)}{=} \left|\left[\left(\mathbb{P}^{(\mathbf{w}\overset{a}{\leftarrow}v)}_{h_a+1} - \mathbb{P}^{(\mathbf{w}\overset{a}{\leftarrow}w)}_{h_a+1}\right) V_{h_a+2}\right](s_{i_a}, a_{k_a})\right| - \left|V^{\text{bad}}_{h_a+1}(s_{i_a}) - V_{h_a+1}(s_{i_a}) + \text{gap}\right|$$
$$\geq \left|\left[\left(\mathbb{P}^{(\mathbf{w}\overset{a}{\leftarrow}v)}_{h_a+1} - \mathbb{P}^{(\mathbf{w}\overset{a}{\leftarrow}w)}_{h_a+1}\right) V_{h_a+2}\right](s_{i_a}, a_{k_a})\right| - \left|V^{\text{bad}}_{h_a+1}(s_{i_a}) - V_{h_a+1}(s_{i_a})\right| - \text{gap}$$
$$\overset{(iii)}{\geq} \left|\left[\left(\mathbb{P}^{(\mathbf{w}\overset{a}{\leftarrow}v)}_{s_a+1} - \mathbb{P}^{(\mathbf{w}\overset{a}{\leftarrow}w)}_{h_a+1}\right) V_{h_a+2}\right](s_{i_a}, a_{k_a})\right| - 3D^{\text{all}}_{\Theta}\left(\mathscr{R}^{(\mathbf{w}\overset{a}{\leftarrow}w)}, \mathscr{R}; \mathbb{P}^{(\mathbf{w}\overset{a}{\leftarrow}w)}\right), \tag{G.15}$$

where (i) is by the Bellman equation, (ii) is valid since

$$r_{h_a+1}(s_{i_a}, a_{k_a}) + \left[\mathbb{P}^{(\mathbf{w}\overset{a}{\leftarrow}w)}_{h_a+1} V_{h_a+2}\right](s_{i_a}, a_{k_a})$$
$$= -A_{h_a+1}(s_{i_a}, a_{k_a}) \cdot \mathbf{1}\left\{a_{k_a} \in \text{supp}\left(\pi^g_{h_a+1}(\cdot|s_{i_a})\right)\right\} + V_{h_a+1}(s_{i_a})$$
$$- \left[\mathbb{P}^{(\mathbf{w}\overset{a}{\leftarrow}w)}_{h_a+1} V_{h_a+2}\right](s_{i_a}, a_{k_a}) + \left[\mathbb{P}^{(\mathbf{w}\overset{a}{\leftarrow}w)}_{h_a+1} V_{h_a+2}\right](s_{i_a}, a_{k_a}) \quad \text{(by Eq.(G.7))}$$
$$= -\text{gap} + V_{h_a+1}(s_{i_a})$$

and (iii) is due to Eq.(G.12) and Eq.(G.13). We next analyse $\left|\left[\left(\mathbb{P}^{(\mathbf{w}\overset{a}{\leftarrow}v)}_{h_a+1} - \mathbb{P}^{(\mathbf{w}\overset{a}{\leftarrow}w)}_{h_a+1}\right) V_{h_a+2}\right](s_{i_a}, a_{k_a})\right|$. We move back to $\pi^{\text{test},(1)}$. By the construction of $\pi^{\text{test},(1)}$ and the transition probabilities of $\mathcal{M}_{\mathbf{w}\overset{a}{\leftarrow}w}$, we have

$$d^{\pi^{\text{test},(1)}}_{h_a+2}(\bar{s}_i; \mathbb{P}^{(\mathbf{w}\overset{a}{\leftarrow}w)}) = \frac{1 + \epsilon' w_i}{S}, \qquad V^{\pi^{\text{test},(1)}}_{h_a+2}(\bar{s}_i; r, \mathbb{P}^{(\mathbf{w}\overset{a}{\leftarrow}w)}) = V_{h_a+2}(\bar{s}_i), \qquad \forall i \in [S]. \tag{G.16}$$

By definition of $D_\Theta^{\text{all}}(\mathscr{R}^{(\mathbf{w}\overset{a}{\leftarrow}w)}, \mathscr{R}; \mathbb{P}^{(\mathbf{w}\overset{a}{\leftarrow}w)})$, we have

$$
D_\Theta^{\text{all}}\left(\mathscr{R}^{(\mathbf{w}\overset{a}{\leftarrow}w)}, \mathscr{R}; \mathbb{P}^{(\mathbf{w}\overset{a}{\leftarrow}w)}\right)
$$

$$
\geq \mathbb{E}_{\mathbb{P}^{(\mathbf{w}\overset{a}{\leftarrow}w)}, \pi^{\text{test},(1)}} \left| V_{h_a+2}^{\pi^{\text{test},(1)}}(s; r_w^{\text{bad}}, \mathbb{P}^{(\mathbf{w}\overset{a}{\leftarrow}w)}) - V_{h_a+2}^{\pi^{\text{test},(1)}}(s; r, \mathbb{P}^{(\mathbf{w}\overset{a}{\leftarrow}w)}) \right|
$$

$$
\geq \sum_{i\in[S]} d_{h_a+2}^{\pi^{\text{test},(1)}}(\bar{s}_i) \cdot \left| V_{h_a+2}^{\pi^{\text{test},(1)}}(\bar{s}_i; r_w^{\text{bad}}, \mathbb{P}^{(\mathbf{w}\overset{a}{\leftarrow}w)}) - V_{h_a+2}^{\pi^{\text{test},(1)}}(\bar{s}_i; r, \mathbb{P}^{(\mathbf{w}\overset{a}{\leftarrow}w)}) \right|
$$

$$
= \sum_{i\in[S]} \frac{1 + \epsilon' w_i}{S} \cdot \left| V_{h_a+2}^{\text{bad}}(\bar{s}_i) - V_{h_a+2}(\bar{s}_i) \right|
$$

$$
\geq \sum_{i\in[S]} \frac{1}{2S} \cdot \left| V_{h_a+2}^{\text{bad}}(\bar{s}_i) - V_{h_a+2}(\bar{s}_i) \right|, \tag{G.17}
$$

where the last second is by Eq.(G.16) and the last line comes from $\epsilon' \in (0, 1/2]$. Applying Eq.(G.17), we obtain that

$$
\left| \left[ \left( \mathbb{P}^{(\mathbf{w}\overset{a}{\leftarrow}v)} - \mathbb{P}^{(\mathbf{w}\overset{a}{\leftarrow}w)} \right) V_{h_a+2} \right] (s_{i_a}, a_{k_a}) \right|
$$

$$
= \left| \frac{\epsilon'}{S} \cdot \sum_{i\in[S]} V_{h_a+2}(\bar{s}_i) \cdot (w_i - v_i) \right|
$$

$$
\geq \left| \frac{\epsilon'}{S} \cdot \sum_{i\in[S]} V_{h_a+2}^{\text{bad}}(\bar{s}_i) \cdot (w_i - v_i) \right| - \frac{\epsilon'}{S} \cdot \sum_{i\in[S]} \left| V_{h_a+2}^{\text{bad}}(\bar{s}_i) - V_{h_a+2}(\bar{s}_i) \right| \cdot |(w_i - v_i)|
$$

$$
\text{(by triangle inequality)}
$$

$$
\geq \left| \frac{\epsilon'}{S} \cdot \sum_{i\in[S]} V_{h_a+2}^{\text{bad}}(\bar{s}_i) \cdot (w_i - v_i) \right| - \frac{2\epsilon'}{S} \cdot \sum_{i\in[S]} \left| V_{h_a+2}^{\text{bad}}(\bar{s}_i) - V_{h_a+2}(\bar{s}_i) \right|
$$

$$
\geq \frac{H\epsilon'}{16} - 2D_\Theta^{\text{all}}\left(\mathscr{R}^{(\mathbf{w}\overset{a}{\leftarrow}w)}, \mathscr{R}; \mathbb{P}^{(\mathbf{w}\overset{a}{\leftarrow}w)}\right), \tag{G.18}
$$

where the second line is by the triangle inequality and the last line comes from Eq.(G.6) and Eq.(G.17). Combining Eq.(G.15) and Eq.(G.18), we complete the proof.

$\square$

### G.3 PROOF OF THEOREM G.2

*Proof of Theorem G.2.* Our method is similar to the one used for the proof of Metelli et al. (2023, Theorem B.3). For any $\epsilon \in (0, 1/2]$, $\delta \in (0, 1)$, We consider an online algorithm $\mathfrak{A}$ such that for any IRL problem $(\mathcal{M}, \pi^{\mathsf{E}})$, we have

$$
\mathbb{P}_{(\mathcal{M}, \pi^{\mathsf{E}}), \mathfrak{A}}\left( D_\Theta^{\text{all}}\left(\mathscr{R}^\star, \widehat{\widehat{\mathscr{R}}}\right) \leq \epsilon \right) \geq 1 - \delta, \tag{G.19}
$$

where $\mathbb{P}_{(\mathcal{M}, \pi^{\mathsf{E}}), \mathfrak{A}}$ denotes the probability measure induced by executing the algorithm $\mathfrak{A}$ in the IRL

problem $(\mathcal{M}, \pi^{\mathsf{E}})$, $\mathscr{R}^\star$ is the ground truth reward mapping and $\widehat{\widehat{\mathscr{R}}}$ is the estimated reward mapping outputted by executing $\mathfrak{A}$ in $(\mathcal{M}, \pi^{\mathsf{E}})$. We define the the identification function for any $(a, \mathbf{w}) \in \mathcal{I} \times \overline{\mathcal{W}}^\mathcal{I}$ by

$$
\mathbf{\Phi}_{a,\mathbf{w}} := \underset{v\in\overline{\mathcal{W}}}{\arg\min}\, D_\Theta^{\text{all}}\left(\mathscr{R}^{(\mathbf{w}\overset{a}{\leftarrow}v)}, \widehat{\widehat{\mathscr{R}}}; \mathbb{P}^{(\mathbf{w}\overset{a}{\leftarrow}v)}\right),
$$

where $\mathscr{R}^{(\mathbf{w})}$ is the ground truth reward mapping induced by $(\mathcal{M}_\mathbf{w}, \pi^{\mathsf{E}})$. Let $v^\star = \mathbf{\Phi}_{a,\mathbf{w}}$. For any $v \neq v^\star \in \overline{\mathcal{W}}$, by definition of $v^\star$, we have

$$
D_\Theta^{\text{all}}\left(\mathscr{R}^{(\mathbf{w}\overset{a}{\leftarrow}v^\star)}, \widehat{\widehat{\mathscr{R}}}; \mathbb{P}^{(\mathbf{w}\overset{a}{\leftarrow}v^\star)}\right) \leq D_\Theta^{\text{all}}\left(\mathscr{R}^{(\mathbf{w}\overset{a}{\leftarrow}v)}, \widehat{\widehat{\mathscr{R}}}; \mathbb{P}^{(\mathbf{w}\overset{a}{\leftarrow}v)}\right).
$$

By applying Lemma G.3, we obtain that

$$\frac{H\epsilon'}{16} \le D_\Theta^{\mathsf{all}}\left(\mathscr{R}^{(\mathbf{w}\xleftarrow{a}v)}, \widehat{\mathscr{R}}; \mathbb{P}^{(\mathbf{w}\xleftarrow{a}v)}\right) + 7D_\Theta^{\mathsf{all}}\left(\mathscr{R}^{(\mathbf{w}\xleftarrow{a}v^\star)}, \widehat{\mathscr{R}}; \mathbb{P}^{(\mathbf{w}\xleftarrow{a}v^\star)}\right) \le 8D_\Theta^{\mathsf{all}}\left(\mathscr{R}^{(\mathbf{w}\xleftarrow{a}v)}, \widehat{\mathscr{R}}; \mathbb{P}^{(\mathbf{w}\xleftarrow{a}v)}\right).$$

Next, we set $\epsilon' = \frac{256\epsilon}{H}$ which implies that

$$\frac{H\epsilon'}{16} \ge 16\epsilon. \tag{G.20}$$

Here, to employ Lemma G.3, we need $\epsilon' \in (0, 1/2]$ which is equivalent to $0 < \epsilon \le H/512$. Then, it holds that

$$D_\Theta^{\mathsf{all}}\left(\mathscr{R}^{(\mathbf{w}\xleftarrow{a}v)}, \widehat{\mathscr{R}}; \mathbb{P}^{(\mathbf{w}\xleftarrow{a}v)}\right) \ge 2\epsilon > \epsilon,$$

which implies that

$$\left\{v \ne \boldsymbol{\Phi}_{a,\mathbf{w}}\right\} \subseteq \left\{D_\Theta^{\mathsf{all}}\left(\mathscr{R}^{(\mathbf{w}\xleftarrow{a}v)}, \widehat{\mathscr{R}}; \mathbb{P}^{(\mathbf{w}\xleftarrow{a}v)}\right) > \epsilon\right\}. \tag{G.21}$$

By Eq.(G.21), we have the following lower bound for the probability

$$\begin{aligned}
\delta &\ge \sup_{v\in\overline{\mathcal{W}}} \mathop{\mathbb{P}}_{(\mathcal{M}_{\mathbf{w}\xleftarrow{a}v}, \pi^{\mathsf{E}}), \mathfrak{A}}\left(D_\Theta^{\mathsf{all}}\left(\mathscr{R}^{(\mathbf{w}\xleftarrow{a}v)}, \widehat{\mathscr{R}}; \mathbb{P}^{(\mathbf{w}\xleftarrow{a}v)}\right) > \epsilon\right) \\
&\ge \sup_{v\in\overline{\mathcal{W}}} \mathop{\mathbb{P}}_{(\mathcal{M}_{\mathbf{w}\xleftarrow{a}v}, \pi^{\mathsf{E}}), \mathfrak{A}}(v \ne \boldsymbol{\Phi}_{a,\mathbf{w}}) \\
&\ge \frac{1}{|\overline{\mathcal{W}}|} \sum_{v\in\overline{\mathcal{W}}} \mathop{\mathbb{P}}_{(\mathcal{M}_{\mathbf{w}\xleftarrow{a}v}, \pi^{\mathsf{E}}), \mathfrak{A}}(v \ne \boldsymbol{\Phi}_{a,\mathbf{w}}),
\end{aligned} \tag{G.22}$$

By applying Theorem D.3 with $\mathbb{P}_0 = \mathop{\mathbb{P}}_{(\mathcal{M}_{\mathbf{w}\xleftarrow{a}0}, \pi^{\mathsf{E}}), \mathfrak{A}}$, $\mathbb{P}_w = \mathop{\mathbb{P}}_{(\mathcal{M}_{\mathbf{w}\xleftarrow{a}w}, \pi^{\mathsf{E}}), \mathfrak{A}}$, we have

$$\frac{1}{|\overline{\mathcal{W}}|} \sum_{(\mathcal{M}_{\mathbf{w}\xleftarrow{a}v}, \pi^{\mathsf{E}}), \mathfrak{A}}(v \ne \boldsymbol{\Phi}_{a,\mathbf{w}}) \ge 1 - \frac{1}{\log|\overline{\mathcal{W}}|}\left(\frac{1}{|\overline{\mathcal{W}}|} \sum_{v\in\overline{\mathcal{W}}} D_{\mathrm{KL}}(\mathop{\mathbb{P}}_{(\mathcal{M}_{\mathbf{w}\xleftarrow{a}v}, \pi^{\mathsf{E}}), \mathfrak{A}}, \mathop{\mathbb{P}}_{(\mathcal{M}_{\mathbf{w}\xleftarrow{a}0}, \pi^{\mathsf{E}}), \mathfrak{A}}) - \log 2\right). \tag{G.23}$$

Our next step is to bound the KL divergence. Using the same scheme in the proof (Metelli et al., 2021, Theorem B.3), we can compute the KL-divergence as follows:

$$\begin{aligned}
&D_{\mathrm{KL}}(\mathop{\mathbb{P}}_{(\mathcal{M}_{\mathbf{w}\xleftarrow{a}v}, \pi^{\mathsf{E}}), \mathfrak{A}}, \mathop{\mathbb{P}}_{(\mathcal{M}_{\mathbf{w}\xleftarrow{a}0}, \pi^{\mathsf{E}}), \mathfrak{A}}) \\
&= \mathbb{E}_{(\mathcal{M}_{\mathbf{w}\xleftarrow{a}v}, \pi^{\mathsf{E}}), \mathfrak{A}}\left[\sum_{t=1}^N D_{\mathrm{KL}}\left(\mathbb{P}_{h_t}^{(\mathbf{w}\xleftarrow{a}w)}(\cdot \mid s_t, a_t), \mathbb{P}_{h_t}^{(\mathbf{w}\xleftarrow{a}0)}(\cdot \mid s_t, a_t)\right)\right] \\
&\le \mathbb{E}_{(\mathcal{M}_{\mathbf{w}\xleftarrow{a}v}, \pi^{\mathsf{E}}), \mathfrak{A}}[N_{h_a}(s_{i_a}, a_{k_a})]D_{\mathrm{KL}}\left(\mathbb{P}_{h_a}^{((\mathbf{w}\xleftarrow{a}v))}(\cdot \mid s_{i_a}, a_{k_a}), \mathbb{P}_{h_a}^{(\mathbf{w}\xleftarrow{a}0)}(\cdot \mid s_{i_a}, a_{k_a})\right) \\
&\le 2(\epsilon')^2\mathbb{E}_{(\mathcal{M}_{\mathbf{w}\xleftarrow{a}v}, \pi^{\mathsf{E}}), \mathfrak{A}}[N_{h_a}(s_{i_a}, a_{k_a})],
\end{aligned} \tag{G.24}$$

where $N_h(s, a) := \sum_{t=1}^N \mathbf{1}\left\{(h_t, s_t, a_t) = (h, s, a)\right\}$ for any given $(h, s, a) \in [H] \times \mathcal{S} \times \mathcal{A}$ and the last inequality comes from Metelli et al. (2021, Lemma E.4). Combining Eq.(G.22) and Eq.(G.23), we have

$$\delta \ge 1 - \frac{1}{\log(|\overline{\mathcal{W}}|)}\left(\frac{1}{|\overline{\mathcal{W}}|} \sum_{v\in\overline{\mathcal{W}}} 2(\epsilon')^2\mathbb{E}_{(\mathcal{M}_{\mathbf{w}\xleftarrow{a}v}, \pi^{\mathsf{E}}), \mathfrak{A}}[N_{h_a}(s_{i_a}, a_{k_a})] - \log 2\right)$$

for any $\mathbf{w}$. It also holds for any $a \in \mathcal{I}$ that

$$\frac{1}{|\overline{\mathcal{W}}|} \sum_{v\in\overline{\mathcal{W}}} \mathbb{E}_{(\mathcal{M}_{\mathbf{w}\xleftarrow{a}v}, \pi^{\mathsf{E}}), \mathfrak{A}}[N_{h_a}(s_{i_a}, a_{k_a})] \ge \frac{(1-\delta)\log|\overline{\mathcal{W}}| - \log 2}{2(\epsilon')^2}. \tag{G.25}$$

By summing Eq.(G.25) over all $\mathbf{w}$, we obtain that

$$\sum_{a \in \mathcal{I}} \frac{1}{|\overline{\mathcal{W}}^{\mathcal{I}}|} \sum_{\mathbf{w} \in \overline{\mathcal{W}}^{\mathcal{I}}} \frac{1}{|\overline{\mathcal{W}}|} \sum_{v \in \overline{\mathcal{W}}} \mathbb{E}_{(\mathcal{M}_{\mathbf{w} \xleftarrow{a} v}, \pi^{\mathsf{E}}), \mathfrak{A}}[N_{h_a}(s_{i_a}, a_{k_a})]$$

$$= \frac{1}{|\overline{\mathcal{W}}^{\mathcal{I}}|} \sum_{\mathbf{w} \in \overline{\mathcal{W}}^{\mathcal{I}}} \sum_{a \in \mathcal{I}} \mathbb{E}_{(\mathcal{M}_{\mathbf{w}}, \pi^{\mathsf{E}}), \mathfrak{A}}[N_{h_a}(s_{i_a}, a_{k_a})]$$

$$\geq HKA \frac{(1-\delta) \log |\overline{\mathcal{W}}| - \log 2}{2(\epsilon')^2}. \tag{G.26}$$

Hence, there exists a $\mathbf{w}^{\mathsf{bad}} \in \overline{\mathcal{W}}^{\mathcal{I}}$ such that

$$\mathbb{E}_{(\mathcal{M}_{\mathbf{w}^{\mathsf{bad}}}, \pi^{\mathsf{E}}), \mathfrak{A}}[N] \geq \sum_{a \in \mathcal{I}} \mathbb{E}_{(\mathcal{M}_{\mathbf{w}^{\mathsf{bad}}}, \pi^{\mathsf{E}}), \mathfrak{A}}\left[N_{h_a}^t(s_{i_a}, a_{k_a})\right] \geq HKA \frac{(1-\delta) \log |\overline{\mathcal{W}}| - \log 2}{2(\epsilon')^2}$$

$$= H^3 KA \frac{(1-\delta) \log |\overline{\mathcal{W}}| - \log 2}{131072 \epsilon^2}, \tag{G.27}$$

where the last line is by $\epsilon' = \frac{\epsilon}{256H}$. By taking $\delta = 1/3$, we obtain that

$$\mathbb{E}_{(\mathcal{M}_{\mathbf{w}^{\mathsf{bad}}}, \pi^{\mathsf{E}}), \mathfrak{A}}[N] \geq H^3 KA \frac{(1-\delta) \log |\overline{\mathcal{W}}| - \log 2}{131072 \epsilon^2} = H^3 KA \frac{2 \log |\overline{\mathcal{W}}| - 3 \log 2}{393216 \epsilon^2}$$

$$= \Omega\left(\frac{H^3 SKA}{\epsilon^2}\right) = \Omega\left(\frac{H^3 SA \min\{S, A\}}{\epsilon^2}\right), \tag{G.28}$$

where the last line follows from Eq.(H.24) and $\log |\overline{\mathcal{W}}| \geq \frac{S}{10}$. $\qquad\square$

## H  LOWER BOUND IN THE OFFLINE SETTING

### H.1  LOWER BOUND OF OFFLINE IRL PROBLEMS

We direct our attention towards the lower bound analysis of the offline IRL problems, under the performance metric described in Section 3.3, particularly in scenarios where $\Theta = \overline{\mathcal{V}} \times \overline{\mathcal{A}}$. In this case $\log \mathcal{N}(\Theta; \epsilon/H)$ is upper-bounded by $\widetilde{\mathcal{O}}(S)$, and the corresponding upper bound of the sample complexity becomes $\widetilde{\mathcal{O}}\left(\frac{C^\star H^4 S^2}{\epsilon^2}\right)$.

Following Metelli et al. (2023) we define the $(\epsilon, \delta)$-PAC algorithm for offline IRL problems for all $\epsilon, \delta \in (0, 1)$.

**Definition H.1** ($(\epsilon, \delta)$-PAC algorithm for offline IRL problems). *We say an offline IRL algorithm $\mathfrak{A}$ is an $(\epsilon, \delta)$-PAC algorithm for offline IRL problems if for any offline IRL problem $(\mathcal{M}, \pi^{\mathsf{E}}, \pi^{\mathsf{b}}, \pi^{\mathsf{val}})$ and any parameter set $\Theta$, with probability $1 - \delta$, $\mathfrak{A}$ outputs a reward mapping $\widehat{\mathscr{R}}$ such that*

$$D_{\Theta}^{\pi^{\mathsf{val}}}(\widehat{\mathscr{R}}, \mathscr{R}^\star) \leq \epsilon.$$

**Theorem H.2** (Lower bound for offline IRL problems). *Fix $\Theta = \overline{\mathcal{V}} \times \overline{\mathcal{A}}$ and let $\mathfrak{A}$ be an $(\epsilon, \delta)$-PAC algorithm for offline IRL problems, where $\delta \leq 1/3$. Then, there exists an offline IRL problem $(\mathcal{M}, \pi^{\mathsf{E}}, \pi^{\mathsf{b}}, \pi^{\mathsf{val}})$ such that, if $H, S \geq 4, A \geq 2, C^\star \geq 2$, there exists an absolute constant $c_0$ such that the sample complexity $N$ is lower bounded by*

$$N \geq \frac{c_0 C^\star H^2 S \min\{S, A\}}{\epsilon^2}.$$

*where $0 < \epsilon \leq (H-2)/1034$.*

The hard instance construction and the proof of Theorem H.2 and the hard instance construction are deferred to Section H.2 and Section H.3, respectively. Our proof involves a modification of the challenging instance constructed in Metelli et al. (2023). Specifically, when $S \leq A$, the lower bound scales with $\Omega(C^\star S^2)$, matching the $C^\star S^2$ factor dependence observed in the upper bound (Theorem 4.2).

## H.2 Hard instance construction

We consider the MDP\R $\mathcal{M}_{\mathbf{w}}$ indexed by vector $\mathbf{w} \in \overline{\mathcal{W}}^{\mathcal{I}}$, defined in Section G. We assume $C^\star \geq 2$. Fix $i^\star \in [K]$, we construct the behavior policy $\pi^{\mathsf{b}}$ as follows:

$$\begin{cases} \pi^{\mathsf{b}}_h(a_0|s_{\text{start}}) = 1 & \text{for all } i \in [K] \text{ and } h \in [H-1], \\ \pi^{\mathsf{b}}_H(a_i|s_{\text{start}}) = \frac{1}{K} & \text{for all } i \in [K], \\ \pi^{\mathsf{b}}_{H+1}(a_0|s_i) = 1 & \text{for all } i \neq i^\star, \\ \pi^{\mathsf{b}}_{H+1}(a_0|s_{i^\star}) = 1 - \frac{1}{C^\star}, \qquad \pi^{\mathsf{b}}_{H+1}(a_1|s_{i^\star}) = \frac{1}{C^\star}, \\ \pi^{\mathsf{b}}_h(a_0|\bar{s}_i) = 1 & \text{for all } i \in [S] \text{ and } h \geq H+2. \end{cases} \qquad \text{(H.1)}$$

And evaluation policy $\pi^{\mathsf{val}}$ is defined by

$$\begin{cases} \pi^{\mathsf{val}}_h(a_0|s_{\text{start}}) = 1 & \text{for all } i \in [K] \text{ and } h \in [H-1], \\ \pi^{\mathsf{val}}_H(a_{i^\star}|s_{\text{start}}) = 1, \\ \pi^{\mathsf{val}}_{H+1}(a_0|s_i) = 1 & \text{for all } i \neq i^\star, \\ \pi^{\mathsf{val}}_{H+1}(a_1|s_{i^\star}) = 1, \\ \pi^{\mathsf{val}}_h(a_0|\bar{s}_i) & \text{for all } i \in [S] \text{ and } h \geq H+2. \end{cases} \qquad \text{(H.2)}$$

For all $\mathbf{w} \in \overline{\mathcal{W}}^{\mathcal{I}}$, we can show that $\pi^{\mathsf{val}}$ has $C^\star$-concentrability in $\mathcal{M}_{\mathbf{w}}$.

**Lemma H.3.** *Suppose that $\epsilon' \in (0, 1/2]$. For any $\mathbf{w} \in \overline{\mathcal{W}}^{\mathcal{I}}$, it holds that*

$$\sum_{(h,s,a)\in[2H+2]\times\mathcal{S}\times\mathcal{A}} \frac{d^{\pi^{\mathsf{val}}}(s,a)}{d^{\pi^{\mathsf{b}}}(s,a)} \leq 3C^\star(H+2)S.$$

*Proof.* By the construction of behavior policy $\pi^{\mathsf{b}}$, we have

$$\text{supp}\left(d^{\pi^{\mathsf{val}}}_h(\cdot,\cdot)\right) \subseteq \{(s_{\text{start}}, a_0), (s_{\text{start}}, a_{k^\star}), (s_{i^\star}, a_1), (\bar{s}_1, a_0), \ldots, (\bar{s}_S, a_0)\}.$$

Since $\pi^{\mathsf{b}}_h = \pi^{\mathsf{val}}_h$ for all $h \in [H-1]$, then

$$d^{\pi^{\mathsf{b}}}_h(s_{\text{start}}, a_0) = d^{\pi^{\mathsf{val}}}_h(s_{\text{start}}, a_0) = 1 \qquad \text{(H.3)}$$

for all $h \in [H-1]$.

At stage $h = H$, we have

$$d^{\pi^{\mathsf{b}}}_H(s_{\text{start}}, a_{i^\star}) = \frac{1}{K}, \qquad d^{\pi^{\mathsf{val}}}_H(s_{\text{start}}, a_{i^\star}) = 1. \qquad \text{(H.4)}$$

At stage $h = H+1$, we have

$$d^{\pi^{\mathsf{b}}}_{H+1}(s_{i^\star}, a_1) = \frac{1}{C^\star K}, \qquad d^{\pi^{\mathsf{val}}}_{H+1}(s_{i^\star}, a_1) = 1. \qquad \text{(H.5)}$$

At stage $h \in \{H+2, \ldots, 2H+2\}$, by direct computation, we obtain that

$$d^{\pi^{\mathsf{b}}}_h(\bar{s}_j, a_0) = \frac{C^\star K - 1}{C^\star SK} + \frac{1 + \epsilon' w^{(i^\star,j,1)}_{H+1}}{C^\star SK}, \qquad d^{\pi^{\mathsf{val}}}_h(\bar{s}_j, a_1) = \frac{1 + \epsilon' w^{(i^\star,j,1)}_{H+1}}{S}, \qquad \text{(H.6)}$$

for all $j \in [S]$. Since $0 < \epsilon \leq 1/2$ and $C^\star \geq 1$, we have

$$\begin{aligned} d^{\pi^{\mathsf{b}}}_h(\bar{s}_j, a_0) &= \frac{C^\star K - 1}{SK} + \frac{1 + \epsilon' w^{(i^\star,j,1)}_{H+1}}{C^\star SK} \\ &\geq \frac{C^\star K - 1}{SK} + \frac{1}{2C^\star SK} = \frac{1}{S}\left(1 - \frac{1}{2C^\star K}\right) \geq \frac{1}{2S} \end{aligned} \qquad \text{(H.7)}$$

and

$$d^{\pi^{\mathsf{val}}}_h(s_{i^\star}, a_1) = \frac{1 + \epsilon' w^{(i^\star,j,1)}_{H+1}}{S} \leq \frac{3}{2S}, \qquad \text{(H.8)}$$

for all $h \geq H + 2$. By Eq.(H.7) and (H.7), we obtain that

$$\frac{d_h^{\pi^{\text{val}}}(\bar{s}_j, a_0)}{d_h^{\pi^{\text{b}}}(\bar{s}_j, a_0)} \leq 3, \tag{H.9}$$

for all $h \geq H + 2$.

Combining Eq.(H.3), Eq.(H.4) and Eq.(H.5), we have

$$\sum_{h=1}^{2H+2} \sum_{(s,a) \in \mathcal{S} \times \mathcal{A}} \frac{d^{\pi^{\text{val}}}(s,a)}{d^{\pi^{\text{b}}}(s,a)} = \sum_{h \in [H-1]} \frac{d_h^{\pi^{\text{val}}}(s_{\text{start}}, a_0)}{d_h^{\pi^{\text{b}}}(s_{\text{start}}, a_0)} + \frac{d_H^{\pi^{\text{val}}}(s_{\text{start}}, a_{i^\star})}{d_H^{\pi^{\text{val}}}(s_{\text{start}}, a_{i\star})} \tag{H.10}$$

$$+ \frac{d_{H+1}^{\pi^{\text{val}}}(s_{i^\star}, a_1)}{d_{H+1}^{\pi^{\text{b}}}(s_{i^\star}, a_1)} + \sum_{h \geq H+2} \sum_{i \in [S]} \frac{d_h^{\pi^{\text{val}}}(\bar{s}_i, a_0)}{d_h^{\pi^{\text{b}}}(\bar{s}_i, a_0)} \tag{H.11}$$

$$= H - 1 + K + C^\star K + \sum_{h \geq H+2} \sum_{i \in [S]} \frac{d_h^{\pi^{\text{val}}}(\bar{s}_i, a_0)}{d_h^{\pi^{\text{b}}}(\bar{s}_i, a_0)} \tag{H.12}$$

$$\leq H - 1 + K + C^\star K + 3(H+1)S \leq C^\star(2H+2)(2S+1), \tag{H.13}$$

where the last second inequality is by Eq.(H.9) and the last inequality is by $C^\star \geq 2$. This completes the proof. $\qquad \square$

Lemma H.3 demonstrate that $\pi^{\text{b}}$ and $\pi^{\text{val}}$ satisfies $C^\star$-concentrability (Assumption 4.1) in any $\mathcal{M}_{\mathbf{w}}$.

**Notations.** To distinguish with different MDP\Rs, we still use $V_h^\pi\big(\cdot; r, \mathbb{P}^{(\mathbf{w})}\big)$ to denote the value function of $\pi$ in MDP $\mathcal{M}_{\mathbf{w}} \cup r$. Given two rewards $r$ $r'$ and $\mathbf{w} \in \overline{\mathcal{W}}^{\mathcal{I}}$, we define $d^{\pi^{\text{val}}}(r, r'; \mathbb{P}^{(\mathbf{w})})$ by:

$$d^{\pi^{\text{val}}}(r, r'; \mathbb{P}^{(\mathbf{w})}) := \sup_{\pi, h \in [H]} \mathbb{E}_{\mathbb{P}^{(\mathbf{w})}} \left| V_h^{\pi^{\text{val}}}(s_h; r, \mathbb{P}^{(\mathbf{w})}) - V_h^{\pi^{\text{val}}}(s_h; r', \mathbb{P}^{(\mathbf{w})}) \right|.$$

Correspondingly, given a parameter set $\Theta$, two reward mappings $\mathscr{R}$, $\mathscr{R}'$, we define $D_\Theta^{\pi^{\text{val}}}(\mathscr{R}, \mathscr{R}'; \mathbb{P}^{(\mathbf{w})})$ by

$$D_\Theta^{\pi^{\text{val}}}(\mathscr{R}, \mathscr{R}'; \mathbb{P}^{(\mathbf{w})}) := \sup_{(V,A) \in \Theta} d^{\pi^{\text{val}}}\Big(\mathscr{R}(V, A), \mathscr{R}'(V, A); \mathbb{P}^{(\mathbf{w})}\Big).$$

In this section, we only consider the case that $\Theta = \overline{\mathcal{V}} \times \overline{\mathcal{A}}$.

**Lemma H.4.** *Given any* $\mathbf{w} \in \overline{\mathcal{W}}^{\mathcal{I}}$, $w \neq v \in \overline{\mathcal{W}}$, *and* $i^\star \in [K]$. *Let* $\mathscr{R}^{(\mathbf{w} \overset{a}{\leftarrow} w)}$, $\mathscr{R}^{(\mathbf{w} \overset{a}{\leftarrow} v)}$ *be the ground truth reward mappings induced by* $\mathcal{M}_{\mathbf{w} \overset{a}{\leftarrow} w}$, $\mathcal{M}_{\mathbf{w} \overset{a}{\leftarrow} v}$ *where* $a = (i^\star, H+1, 1) \in \mathcal{I}$. *Set* $\Theta = \overline{\mathcal{V}} \times \overline{\mathcal{A}}$. *For any rewarding mapping* $\mathscr{R}$ *and* $\epsilon' \in (0, 1/2]$, *we have*

$$7 D_\Theta^{\pi^{\text{val}}}\Big(\mathscr{R}^{(\mathbf{w} \overset{a}{\leftarrow} w)}, \mathscr{R}; \mathbb{P}^{\mathbf{w} \overset{a}{\leftarrow} w}\Big) + D_\Theta^{\pi^{\text{val}}}\Big(\mathscr{R}^{(\mathbf{w} \overset{a}{\leftarrow} v)}, \mathscr{R}; \mathbb{P}^{\mathbf{w} \overset{a}{\leftarrow} v}\Big) \geq \frac{H\epsilon'}{16}.$$

*Proof.* We consider similar construction of bad parameter $V^{\text{bad}}$, $A^{\text{bad}}$ in the Proof of Lemma G.3. To summarize, $\big(V^{\text{bad}}, A^{\text{bad}}\big)$ is given by

- We set $A_h^{\text{bad}}(s, a) = 0$ for all $(h, s, a) \in [2H+2] \times \mathcal{S} \times \mathcal{A}$.

- We set $V_h^{\text{bad}}$ by

$$V_h^{\text{bad}}(s) := \begin{cases} \frac{(2H+2-h) \cdot (w_i - v_i)}{2} & \text{if } s = \bar{s}_i, h = H+2, \\ 0 & \text{otherwise.} \end{cases} \tag{H.14}$$

Similarly, we define $r_w^{\text{bad}}$, $r_v^{\text{bad}}$ and $r$ by

$$r_w^{\text{bad}} := \mathscr{R}^{(\mathbf{w} \overset{a}{\leftarrow} w)}\big(V^{\text{bad}}, A^{\text{bad}}\big), \qquad r_w^{\text{bad}} := \mathscr{R}^{(\mathbf{w} \overset{a}{\leftarrow} v)}\big(V^{\text{bad}}, A^{\text{bad}}\big), \qquad r := \mathscr{R}\big(V^{\text{bad}}, A^{\text{bad}}\big).$$

By definition of $\mathscr{R}^{\left(\mathbf{w}\xleftarrow{a}w\right)}, \mathscr{R}^{\left(\mathbf{w}\xleftarrow{a}v\right)}$, we have

$$
\begin{aligned}
\left|r_{w,H+1}^{\mathsf{bad}}(s_{i^\star}, a_1) - r_{v,H+1}^{\mathsf{bad}}(s_{i^\star}, a_1)\right| &= \left|\left[\left(\mathbb{P}_{H+2}^{\left(\mathbf{w}\xleftarrow{a}w\right)} - \mathbb{P}_{H+2}^{\left(\mathbf{w}\xleftarrow{a}v\right)}\right)V_{H+1}^{\mathsf{bad}}\right](s_{i^\star}, a_1)\right| \\
&= \epsilon' \cdot \left|\sum_{i\in[S]} \frac{(w_i - v_i)V_{H+2}^{\mathsf{bad}}}{S}\right| \\
&= \frac{H\epsilon'}{2S} \cdot \sum_{i\in[S]} (w_i - v_i)^2 \geq \frac{H\epsilon'}{16}, \quad\quad\quad (\text{H.15})
\end{aligned}
$$

where the last inequality follows from Eq.(G.4). By definition of $D_\Theta^{\pi^{\mathsf{val}}}$, we have

$$
\begin{aligned}
D_\Theta^{\pi^{\mathsf{val}}}\left(\mathscr{R}^{\left(\mathbf{w}\xleftarrow{a}w\right)}, \mathscr{R}; \mathbb{P}^{\left(\mathbf{w}\xleftarrow{a}w\right)}\right) &\geq d^{\pi^{\mathsf{val}}}\left(r_w^{\mathsf{bad}}, r; \mathbb{P}^{\left(\mathbf{w}\xleftarrow{a}w\right)}\right) \\
&\geq \mathbb{E}_{\mathbb{P}^{\left(\mathbf{w}\xleftarrow{a}w\right)}, \pi^{\mathsf{val}}} \left|V_{H+2}^{\pi^{\mathsf{val}}}(s; r_w^{\mathsf{bad}}, \mathbb{P}^{\left(\mathbf{w}\xleftarrow{a}w\right)}) - V_{H+2}^{\pi^{\mathsf{val}}}(s; r, \mathbb{P}^{\left(\mathbf{w}\xleftarrow{a}w\right)})\right| \\
&= \sum_{i\in[S]} \frac{1 + \epsilon' \cdot w_i}{S} \left|V_{H+2}^{\pi^{\mathsf{val}}}(\bar{s}_i; r_w^{\mathsf{bad}}, \mathbb{P}^{\left(\mathbf{w}\xleftarrow{a}w\right)}) - V_{H+2}^{\pi^{\mathsf{val}}}(\bar{s}_i; r, \mathbb{P}^{\left(\mathbf{w}\xleftarrow{a}w\right)})\right| \\
&\geq \sum_{i\in[S]} \frac{\epsilon'}{2S} \left|V_{H+2}^{\pi^{\mathsf{val}}}(s; r_w^{\mathsf{bad}}, \mathbb{P}^{\left(\mathbf{w}\xleftarrow{a}w\right)}) - V_{H+2}^{\pi^{\mathsf{val}}}(s; r, \mathbb{P}^{\left(\mathbf{w}\xleftarrow{a}w\right)})\right|,
\end{aligned}
$$
$$(\text{H.16})$$

where the last line is due to $\epsilon' \in (0, 1/2]$. By construction of $\pi^{\mathsf{val}}$, in MDP\R $\mathcal{M}_{\mathbf{w}\xleftarrow{a}v}$, the visiting probability $d_{H+1}^{\pi^{\mathsf{val}}}$ is given by

$$
d_{H+1}^{\pi^{\mathsf{val}}}\left(s_{i^\star}, a_1; \mathbb{P}^{\left(\mathbf{w}\xleftarrow{a}w\right)}\right) = 1.
$$

For $D_\Theta^{\pi^{\mathsf{val}}}\left(\mathscr{R}^{\left(\mathbf{w}\xleftarrow{a}v\right)}, \mathscr{R}; \mathbb{P}^{\left(\mathbf{w}\xleftarrow{a}v\right)}\right)$, we also have

$$
\begin{aligned}
&D_\Theta^{\pi^{\mathsf{val}}}\left(\mathscr{R}^{\left(\mathbf{w}\xleftarrow{a}v\right)}, \mathscr{R}; \mathbb{P}^{\left(\mathbf{w}\xleftarrow{a}v\right)}\right) \geq d^{\pi^{\mathsf{val}}}\left(r_v^{\mathsf{bad}}, r; \mathbb{P}^{\left(\mathbf{w}\xleftarrow{a}v\right)}\right) \\
&\geq \mathbb{E}_{\mathbb{P}^{\left(\mathbf{w}\xleftarrow{a}v\right)}, \pi^{\mathsf{val}}} \left|V_{H+1}^{\pi^{\mathsf{val}}}(s; r_v^{\mathsf{bad}}, \mathbb{P}^{\left(\mathbf{w}\xleftarrow{a}v\right)}) - V_{H+1}^{\pi^{\mathsf{val}}}(s; r, \mathbb{P}^{\left(\mathbf{w}\xleftarrow{a}v\right)})\right| \\
&= \left|V_{H+1}^{\pi^{\mathsf{val}}}(s_{i^\star}; r_v^{\mathsf{bad}}, \mathbb{P}^{\left(\mathbf{w}\xleftarrow{a}v\right)}) - V_{H+1}^{\pi^{\mathsf{val}}}(s_{i^\star}; r, \mathbb{P}^{\left(\mathbf{w}\xleftarrow{a}v\right)})\right| \\
&= \left|r_{v,H+1}^{\mathsf{bad}}(s_{i^\star}, a_1) - r_{H+1}(s_{i^\star}, a_1)\right. \\
&\quad \left. - \sum_{i\in[S]} \mathbb{P}_{H+1}^{\left(\mathbf{w}\xleftarrow{a}v\right)}(\bar{s}_i | s_{i^\star}, a_1) \cdot \left(V_{H+2}^{\pi^{\mathsf{val}}}(\bar{s}_i; r_v^{\mathsf{bad}}, \mathbb{P}^{\left(\mathbf{w}\xleftarrow{a}v\right)}) - V_{H+2}^{\pi^{\mathsf{val}}}(\bar{s}_i; r, \mathbb{P}^{\left(\mathbf{w}\xleftarrow{a}v\right)})\right)\right| \\
&\geq \left|r_{v,H+1}^{\mathsf{bad}}(s_{i^\star}, a_1) - r_{H+1}(s_{i^\star}, a_1)\right| \\
&\quad - \sum_{i\in[S]} \frac{1 + \epsilon' \cdot v_i}{S} \cdot \left|V_{H+2}^{\pi^{\mathsf{val}}}(\bar{s}_i; r_v^{\mathsf{bad}}, \mathbb{P}^{\left(\mathbf{w}\xleftarrow{a}v\right)}) - V_{H+2}^{\pi^{\mathsf{val}}}(\bar{s}_i; r, \mathbb{P}^{\left(\mathbf{w}\xleftarrow{a}v\right)})\right|, \quad (\text{H.17})
\end{aligned}
$$

where the second last line is by the bellman equation and the last line is due to the triangle inequality. Since $\mathbb{P}_h^{\left(\mathbf{w}\xleftarrow{a}w\right)} = \mathbb{P}_h^{\left(\mathbf{w}\xleftarrow{a}w\right)}$ and $r_{w,h}^{\mathsf{bad}} = r_{v,h}^{\mathsf{bad}}$ for all $h \geq H + 2$, we have

$$
V_{H+2}^{\pi^{\mathsf{val}}}(\bar{s}_i; r, \mathbb{P}^{\left(\mathbf{w}\xleftarrow{a}v\right)}) = V_{H+2}^{\pi^{\mathsf{val}}}(\bar{s}_i; r, \mathbb{P}^{\left(\mathbf{w}\xleftarrow{a}w\right)}), \quad\quad V_{H+2}^{\pi^{\mathsf{val}}}(\bar{s}_i; r_v^{\mathsf{bad}}, \mathbb{P}^{\left(\mathbf{w}\xleftarrow{a}v\right)}) = V_{H+2}^{\pi^{\mathsf{val}}}(\bar{s}_i; r_w^{\mathsf{bad}}, \mathbb{P}^{\left(\mathbf{w}\xleftarrow{a}w\right)}).
$$
$$(\text{H.18})$$

Apply Eq.(H.18) to Eq.(H.17), we have

$$
D_\Theta^{\pi^{\mathsf{val}}}\left(\mathscr{R}^{\left(\mathbf{w}\xleftarrow{a}v\right)}, \mathscr{R}; \mathbb{P}^{\left(\mathbf{w}\xleftarrow{a}v\right)}\right)
$$

$$\geq \left| r^{\mathsf{bad}}_{v,H+1}(s_{i^\star},a_1) - r_{H+1}(s_{i^\star},a_1) \right|$$

$$- \sum_{i\in[S]} \frac{1+\epsilon'\cdot v_i}{S}\cdot \left| V^{\pi^{\mathsf{val}}}_{H+2}(\bar{s}_i; r^{\mathsf{bad}}_v, \mathbb{P}^{\left(\mathbf{w}\overset{a}{\leftarrow}v\right)}) - V^{\pi^{\mathsf{val}}}_{H+2}(\bar{s}_i; r, \mathbb{P}^{\left(\mathbf{w}\overset{a}{\leftarrow}v\right)}) \right|$$

$$= \left| r^{\mathsf{bad}}_{v,H+1}(s_{i^\star},a_1) - r_{H+1}(s_{i^\star},a_1) \right|$$

$$- \sum_{i\in[S]} \frac{1+\epsilon'\cdot v_i}{S}\cdot \left| V^{\pi^{\mathsf{val}}}_{H+2}(\bar{s}_i; r^{\mathsf{bad}}_w, \mathbb{P}^{\left(\mathbf{w}\overset{a}{\leftarrow}w\right)}) - V^{\pi^{\mathsf{val}}}_{H+2}(\bar{s}_i; r, \mathbb{P}^{\left(\mathbf{w}\overset{a}{\leftarrow}w\right)}) \right|$$

$$\geq \left| r^{\mathsf{bad}}_{v,H+1}(s_{i^\star},a_1) - r_{H+1}(s_{i^\star},a_1) \right|$$

$$- \sum_{i\in[S]} \frac{3}{2S}\cdot \left| V^{\pi^{\mathsf{val}}}_{H+2}(\bar{s}_i; r^{\mathsf{bad}}_w, \mathbb{P}^{\left(\mathbf{w}\overset{a}{\leftarrow}w\right)}) - V^{\pi^{\mathsf{val}}}_{H+2}(\bar{s}_i; r, \mathbb{P}^{\left(\mathbf{w}\overset{a}{\leftarrow}w\right)}) \right|$$

$$\geq \left| r^{\mathsf{bad}}_{v,H+1}(s_{i^\star},a_1) - r_{H+1}(s_{i^\star},a_1) \right| - 3D^{\pi^{\mathsf{val}}}_{\Theta}\left( \mathscr{R}^{\left(\mathbf{w}\overset{a}{\leftarrow}w\right)}, \mathscr{R}; \mathbb{P}^{\left(\mathbf{w}\overset{a}{\leftarrow}w\right)} \right), \tag{H.19}$$

where the last second inequality comes from $\epsilon' \in (0,1/2]$ and the last inequality comes from Eq.(H.16).

We next bound $\left| r^{\mathsf{bad}}_{w,H+1}(s_{i^\star},a_1) - r_{H+1}(s_{i^\star},a_1) \right|$ by $D^{\pi^{\mathsf{val}}}_{\Theta}\left( \mathscr{R}^{\left(\mathbf{w}\overset{a}{\leftarrow}w\right)}, \mathscr{R}; \mathbb{P}^{\left(\mathbf{w}\overset{a}{\leftarrow}w\right)} \right)$.

$$D^{\pi^{\mathsf{val}}}_{\Theta}\left( \mathscr{R}^{\left(\mathbf{w}\overset{a}{\leftarrow}w\right)}, \mathscr{R}; \mathbb{P}^{\left(\mathbf{w}\overset{a}{\leftarrow}w\right)} \right) \geq d^{\pi^{\mathsf{val}}}\left( r^{\mathsf{bad}}_w, r; \mathbb{P}^{\left(\mathbf{w}\overset{a}{\leftarrow}w\right)} \right)$$

$$\geq \mathbb{E}_{\mathbb{P}^{\left(\mathbf{w}\overset{a}{\leftarrow}w\right)},\pi^{\mathsf{val}}} \left| V^{\pi^{\mathsf{val}}}_{H+1}(s; r^{\mathsf{bad}}_v, \mathbb{P}^{\left(\mathbf{w}\overset{a}{\leftarrow}w\right)}) - V^{\pi^{\mathsf{val}}}_{H+1}(s; r, \mathbb{P}^{\left(\mathbf{w}\overset{a}{\leftarrow}w\right)}) \right|$$

$$= \left| V^{\pi^{\mathsf{val}}}_{H+1}(s_{i^\star}; r^{\mathsf{bad}}_w, \mathbb{P}^{\left(\mathbf{w}\overset{a}{\leftarrow}w\right)}) - V^{\pi^{\mathsf{val}}}_{H+1}(s_{i^\star}; r, \mathbb{P}^{\left(\mathbf{w}\overset{a}{\leftarrow}w\right)}) \right|$$

$$= \Bigg| r^{\mathsf{bad}}_{w,H+1}(s_{i^\star},a_1) - r_{H+1}(s_{i^\star},a_1)$$

$$- \sum_{i\in[S]} \mathbb{P}^{\left(\mathbf{w}\overset{a}{\leftarrow}w\right)}_{H+1}(\bar{s}_i|s_{i^\star},a_1)\cdot \left( V^{\pi^{\mathsf{val}}}_{H+2}(\bar{s}_i; r^{\mathsf{bad}}_w, \mathbb{P}^{\left(\mathbf{w}\overset{a}{\leftarrow}w\right)}) - V^{\pi^{\mathsf{val}}}_{H+2}(\bar{s}_i; r, \mathbb{P}^{\left(\mathbf{w}\overset{a}{\leftarrow}w\right)}) \right) \Bigg|$$

$$\geq \left| r^{\mathsf{bad}}_{w,H+1}(s_{i^\star},a_1) - r_{H+1}(s_{i^\star},a_1) \right|$$

$$- \sum_{i\in[S]} \frac{1+\epsilon'\cdot w_i}{S}\cdot \left| V^{\pi^{\mathsf{val}}}_{H+2}(\bar{s}_i; r^{\mathsf{bad}}_w, \mathbb{P}^{\left(\mathbf{w}\overset{a}{\leftarrow}w\right)}) - V^{\pi^{\mathsf{val}}}_{H+2}(\bar{s}_i; r, \mathbb{P}^{\left(\mathbf{w}\overset{a}{\leftarrow}w\right)}) \right|$$

$$\geq \left| r^{\mathsf{bad}}_{w,H+1}(s_{i^\star},a_1) - r_{H+1}(s_{i^\star},a_1) \right|$$

$$- \sum_{i\in[S]} \frac{3}{2S}\cdot \left| V^{\pi^{\mathsf{val}}}_{H+2}(\bar{s}_i; r^{\mathsf{bad}}_w, \mathbb{P}^{\left(\mathbf{w}\overset{a}{\leftarrow}w\right)}) - V^{\pi^{\mathsf{val}}}_{H+2}(\bar{s}_i; r, \mathbb{P}^{\left(\mathbf{w}\overset{a}{\leftarrow}w\right)}) \right|$$

$$\geq \left| r^{\mathsf{bad}}_{w,H+1}(s_{i^\star},a_1) - r_{H+1}(s_{i^\star},a_1) \right| - 3D^{\pi^{\mathsf{val}}}_{\Theta}\left( \mathscr{R}^{\left(\mathbf{w}\overset{a}{\leftarrow}w\right)}, \mathscr{R}; \mathbb{P}^{\left(\mathbf{w}\overset{a}{\leftarrow}w\right)} \right), \tag{H.20}$$

where the last second inequality comes from $\epsilon' \in (0,1/2]$ and the last inequality is by Eq.(H.16). Eq.(H.20) is equivalent to

$$4D^{\pi^{\mathsf{val}}}_{\Theta}\left( \mathscr{R}^{\left(\mathbf{w}\overset{a}{\leftarrow}w\right)}, \mathscr{R}; \mathbb{P}^{\left(\mathbf{w}\overset{a}{\leftarrow}w\right)} \right) \geq \left| r^{\mathsf{bad}}_{w,H+1}(s_{i^\star},a_1) - r_{H+1}(s_{i^\star},a_1) \right|. \tag{H.21}$$

Combining Eq.(H.19) and Eq.(H.21), we conclude that

$$7D^{\pi^{\mathsf{val}}}_{\Theta}\left( \mathscr{R}^{\left(\mathbf{w}\overset{a}{\leftarrow}w\right)}, \mathscr{R}; \mathbb{P}^{\left(\mathbf{w}\overset{a}{\leftarrow}w\right)} \right) + D^{\pi^{\mathsf{val}}}_{\Theta}\left( \mathscr{R}^{\left(\mathbf{w}\overset{a}{\leftarrow}v\right)}, \mathscr{R}; \mathbb{P}^{\left(\mathbf{w}\overset{a}{\leftarrow}v\right)} \right)$$

$$\geq \left| r^{\mathsf{bad}}_{w,H+1}(s_{i^\star},a_1) - r_{H+1}(s_{i^\star},a_1) \right| + \left| r^{\mathsf{bad}}_{v,H+1}(s_{i^\star},a_1) - r_{H+1}(s_{i^\star},a_1) \right|$$

$$\geq \left| r^{\mathsf{bad}}_{v,H+1}(s_{i^\star},a_1) - r^{\mathsf{bad}}_{w,H+1}(s_{i^\star},a_1) \right| \geq \frac{H\epsilon'}{16}, \tag{H.22}$$

where the last inequality comes from Eq.(H.15). This completes the proof.

$$\square$$

## H.3 Proof for Theorem H.2

Our proof is similar to the proof of Theorem G.2 in Section G.

*Proof of Theorem H.2.* For any $\epsilon \in (0, 1/2]$, $\delta \in (0, 1)$, We consider an offline IRL algorithm $\mathfrak{A}$ such that for any IRL problem $(\mathcal{M}, \pi^{\mathsf{E}})$, we have

$$\mathbb{P}_{(\mathcal{M}, \pi^{\mathsf{E}}), \mathfrak{A}}\left(D_{\Theta}^{\pi^{\text{val}}}\left(\mathscr{R}^{\star}, \widehat{\mathscr{R}}\right) \le \epsilon\right) \ge 1 - \delta, \tag{H.23}$$

where $\mathbb{P}_{(\mathcal{M}, \pi^{\mathsf{E}}), \mathfrak{A}}$ denotes the probability measure induced by executing the algorithm $\mathfrak{A}$ in the IRL problem $(\mathcal{M}, \pi^{\mathsf{E}})$, $\mathscr{R}^{\star}$ is the ground truth reward mapping and $\widehat{\mathscr{R}}$ is the estimated reward mapping outputted by executing $\mathfrak{A}$ in $(\mathcal{M}, \pi^{\mathsf{E}})$. Fix $i^{\star} \in [S]$, We define the the identification function for any $\mathbf{w} \in \overline{\mathcal{W}}$ by

$$\mathbf{\Phi_w} := \arg\min_{v \in \overline{\mathcal{W}}} D_{\Theta}^{\pi^{\text{val}}}\left(\mathscr{R}^{\left(\mathbf{w} \overset{a}{\leftarrow} v\right)}, \widehat{\mathscr{R}}; \mathbb{P}^{\left(\mathbf{w} \overset{a}{\leftarrow} v\right)}\right),$$

where $a = (i^{\star}, H + 1, 1)$, $\mathscr{R}^{\left(\mathbf{w} \overset{a}{\leftarrow} v\right)}$ is the ground truth reward mapping induced by $(\mathcal{M}_{\mathbf{w} \overset{a}{\leftarrow} v}, \pi^{\mathsf{E}})$. Let $v^{\star} = \mathbf{\Phi}_{a, \mathbf{w}}$. For any $v \ne v^{\star} \in \overline{\mathcal{W}}$, by definition of $v^{\star}$, we have

$$D_{\Theta}^{\pi^{\text{val}}}\left(\mathscr{R}^{\left(\mathbf{w} \overset{a}{\leftarrow} v^{\star}\right)}, \widehat{\mathscr{R}}; \mathbb{P}^{\left(\mathbf{w} \overset{a}{\leftarrow} v^{\star}\right)}\right) \le D_{\Theta}^{\pi^{\text{val}}}\left(\mathscr{R}^{\left(\mathbf{w} \overset{a}{\leftarrow} v\right)}, \widehat{\mathscr{R}}; \mathbb{P}^{\left(\mathbf{w} \overset{a}{\leftarrow} v\right)}\right).$$

By applying Lemma G.3, we obtain that

$$\frac{H\epsilon'}{16} \le D_{\Theta}^{\pi^{\text{val}}}\left(\mathscr{R}^{\left(\mathbf{w} \overset{a}{\leftarrow} v\right)}, \widehat{\mathscr{R}}; \mathbb{P}^{\left(\mathbf{w} \overset{a}{\leftarrow} v\right)}\right) + 7 D_{\Theta}^{\pi^{\text{val}}}\left(\mathscr{R}^{\left(\mathbf{w} \overset{a}{\leftarrow} v^{\star}\right)}, \widehat{\mathscr{R}}; \mathbb{P}^{\left(\mathbf{w} \overset{a}{\leftarrow} v^{\star}\right)}\right) \le 8 D_{\Theta}^{\text{all}}\left(\mathscr{R}^{\left(\mathbf{w} \overset{a}{\leftarrow} v\right)}, \widehat{\mathscr{R}}; \mathbb{P}^{\left(\mathbf{w} \overset{a}{\leftarrow} v\right)}\right).$$

Next, we set $\epsilon' = \frac{256\epsilon}{H}$ which implies that

$$\frac{H\epsilon'}{16} \ge 16\epsilon. \tag{H.24}$$

Here, to employ Lemma H.4, we need $\epsilon' \in (0, 1/2]$ which is equivalent to $0 < \epsilon \le H/512$. Then, it holds that

$$D_{\Theta}^{\pi^{\text{val}}}\left(\mathscr{R}^{\left(\mathbf{w} \overset{a}{\leftarrow} v\right)}, \widehat{\mathscr{R}}; \mathbb{P}^{\left(\mathbf{w} \overset{a}{\leftarrow} v\right)}\right) \ge 2\epsilon > \epsilon,$$

which implies that

$$\left\{v \ne \mathbf{\Phi_w}\right\} \subseteq \left\{D_{\Theta}^{\pi^{\text{val}}}\left(\mathscr{R}^{\left(\mathbf{w} \overset{a}{\leftarrow} v\right)}, \widehat{\mathscr{R}}; \mathbb{P}^{\left(\mathbf{w} \overset{a}{\leftarrow} v\right)}\right) \ge \epsilon\right\}.$$

By Eq.(H.23), we have the following lower bound for the probability

$$\begin{aligned}
\delta &\ge \sup_{v \in \overline{\mathcal{W}}} \mathbb{P}_{(\mathcal{M}_{\mathbf{w} \overset{a}{\leftarrow} v}, \pi^{\mathsf{E}}), \mathfrak{A}}\left(D_{\Theta}^{\pi^{\text{val}}}\left(\mathscr{R}^{\left(\mathbf{w} \overset{a}{\leftarrow} v\right)}, \widehat{\mathscr{R}}; \mathbb{P}^{\left(\mathbf{w} \overset{a}{\leftarrow} v\right)}\right) \ge \epsilon\right) \\
&\ge \sup_{v \in \overline{\mathcal{W}}} \mathbb{P}_{(\mathcal{M}_{\mathbf{w} \overset{a}{\leftarrow} v}, \pi^{\mathsf{E}}), \mathfrak{A}}\left(v \ne \mathbf{\Phi_w}\right) \\
&\ge \frac{1}{|\overline{\mathcal{W}}|} \sum_{v \in \overline{\mathcal{W}}} \mathbb{P}_{(\mathcal{M}_{\mathbf{w} \overset{a}{\leftarrow} v}, \pi^{\mathsf{E}}), \mathfrak{A}}\left(v \ne \mathbf{\Phi_w}\right). \tag{H.25}
\end{aligned}$$

By applying Theorem D.3 with $\mathbb{P}_0 = \mathbb{P}_{(\mathcal{M}_{\mathbf{w} \overset{a}{\leftarrow} 0}, \pi^{\mathsf{E}}), \mathfrak{A}}$, $\mathbb{P}_w = \mathbb{P}_{(\mathcal{M}_{\mathbf{w} \overset{a}{\leftarrow} w}, \pi^{\mathsf{E}}), \mathfrak{A}}$, we have

$$\frac{1}{|\overline{\mathcal{W}}|} \sum_{v \in \overline{\mathcal{W}}} \mathbb{P}_{(\mathcal{M}_{\mathbf{w} \overset{a}{\leftarrow} w}, \pi^{\mathsf{E}}), \mathfrak{A}}\left(v \ne \mathbf{\Phi_w}\right) \ge 1 - \frac{1}{\log|\overline{\mathcal{W}}|}\left(\frac{1}{|\overline{\mathcal{W}}|} \sum_{v \in \overline{\mathcal{W}}} D_{\text{KL}}\left(\mathbb{P}_{(\mathcal{M}_{\mathbf{w} \overset{a}{\leftarrow} v}, \pi^{\mathsf{E}}), \mathfrak{A}}, \mathbb{P}_{(\mathcal{M}_{\mathbf{w} \overset{a}{\leftarrow} 0}, \pi^{\mathsf{E}}), \mathfrak{A}}\right) - \log 2\right). \tag{H.26}$$

Our next step is to bound the KL divergence. Using the same scheme in the proof (Metelli et al., 2021, Theorem B.3), we can compute the KL-divergence as follows:

$$D_{\text{KL}}\left(\mathbb{P}_{(\mathcal{M}_{\mathbf{w} \overset{a}{\leftarrow} v}, \pi^{\mathsf{E}}), \mathfrak{A}}, \mathbb{P}_{(\mathcal{M}_{\mathbf{w} \overset{a}{\leftarrow} 0}, \pi^{\mathsf{E}}), \mathfrak{A}}\right)$$

$$
= \mathbb{E}_{(\mathcal{M}_{\mathbf{w} \overset{a}{\leftarrow} v}, \pi^{\mathsf{E}}), \mathfrak{A}} \left[ \sum_{t=1}^{N} D_{\mathrm{KL}} \left( \mathbb{P}^{(\mathbf{w} \overset{a}{\leftarrow} v)} h_t(\cdot \mid s_t, a_t), \mathbb{P}^{(\mathbf{w} \overset{a}{\leftarrow} 0)}_{h_t}(\cdot \mid s_t, a_t) \right) \right]
$$

$$
\leq \mathbb{E}_{(\mathcal{M}_{\mathbf{w} \overset{a}{\leftarrow} v}, \pi^{\mathsf{E}}), \mathfrak{A}} [N_{h_a}(s_{i_a}, a_{k_a})] D_{\mathrm{KL}} \left( \mathbb{P}^{(\mathbf{w} \overset{a}{\leftarrow} v)}_{H+1}(\cdot \mid s_{i^\star}, a_1), \mathbb{P}^{(\mathbf{w} \overset{a}{\leftarrow} 0)}_{H+1}(\cdot \mid s_{i^\star}, a_1) \right)
$$

$$
\leq 2(\epsilon')^2 \mathbb{E}_{(\mathcal{M}_{\mathbf{w} \overset{a}{\leftarrow} v}, \pi^{\mathsf{E}}), \mathfrak{A}} [N_{H+1}(s_{i^\star}, a_1)], \tag{H.27}
$$

where $N_h(s, a) := \sum_{t=1}^{N} \mathbf{1}\{(h_t, s_t, a_t) = (h, s, a)\}$ for any given $(h, s, a) \in [2H + 2] \times \mathcal{S} \times \mathcal{A}$ and the last inequality comes from (Metelli et al., 2021, Lemma E.4). Combining Eq.(H.25) and (H.26), we have

$$
\delta \geq 1 - \frac{1}{\log(|\overline{\mathcal{W}}|)} \left( \frac{1}{|\overline{\mathcal{W}}|} \sum_{v \in \overline{\mathcal{W}}} 2(\epsilon')^2 \mathbb{E}_{(\mathcal{M}_{\mathbf{w} \overset{a}{\leftarrow} v}, \pi^{\mathsf{E}}), \mathfrak{A}} [N_{h_a}(s_{i^\star}, a_1)] - \log 2 \right)
$$

for any $\mathbf{w}$. It also holds that

$$
\frac{1}{|\overline{\mathcal{W}}|} \sum_{v \in \overline{\mathcal{W}}} \mathbb{E}_{(\mathcal{M}_{\mathbf{w} \overset{a}{\leftarrow} v}, \pi^{\mathsf{E}}), \mathfrak{A}} [N_{H+1}(s_{i^\star}, a_1)] \geq \frac{(1 - \delta) \log |\overline{\mathcal{W}}| - \log 2}{2(\epsilon')^2}. \tag{H.28}
$$

Hence, there exists a $\mathbf{w}^{\mathrm{hard}} \in \overline{\mathcal{W}}$ such that

$$
\mathbb{E}_{(\mathcal{M}_{\mathbf{w}^{\mathrm{hard}}}, \pi^{\mathsf{E}}), \mathfrak{A}} [N_{H+1}(s_{i^\star}, a_1)] \geq \frac{(1 - \delta) \log |\overline{\mathcal{W}}| - \log 2}{2(\epsilon')^2}. \tag{H.29}
$$

By taking $\delta = 1/3$, we have

$$
\mathbb{E}_{(\mathcal{M}_{\mathbf{w}^{\mathrm{hard}}}, \pi^{\mathsf{E}}), \mathfrak{A}} [N_{H+1}(s_{i^\star}, a_1)] \geq \frac{(1 - \delta) \log |\overline{\mathcal{W}}| - \log 2}{2(\epsilon')^2} = \frac{2 \log |\overline{\mathcal{W}}| - 3 \log 2}{6(\epsilon')^2} = \Omega \left( \frac{H^2 S}{\epsilon^2} \right), \tag{H.30}
$$

where the last equality follows from $\epsilon' = \frac{256\epsilon}{H}$ and $\log |\overline{\mathcal{W}}| \geq \frac{S}{10}$. By construction of $\pi^{\mathsf{b}}$, it holds that $N_{H+1}(s_{i^\star}, a_1) \sim \mathrm{Bin}\left(K, \frac{1}{C^\star K}\right)$, which implies that

$$
\mathbb{E}_{(\mathcal{M}_{\mathbf{w}^{\mathrm{hard}}}, \pi^{\mathsf{E}}), \mathfrak{A}} [N] \geq C^\star K \cdot \Omega \left( \frac{H^2 S}{\epsilon} \right) = \Omega \left( \frac{C^\star H^2 S K}{\epsilon^2} \right) = \Omega \left( \frac{C^\star H^2 S \min\{S, A\}}{\epsilon^2} \right).
$$

$\square$

# I  TRANSFER LEARNING

In this section, we explore the application of IRL in the context of transfer learning. Specifically, we apply the rewards learned by Algorithm 1 and Algorithm 6 to do the same tasks in a different environment.

To distinguish different environments, given a transition dynanmics $\mathbb{P}$ and policy $\pi$, we introduce the following notations: $\left\{ d_h^{\mathbb{P}, \pi} \right\}_{h \in [H]}$ represents the visitation probability induced by $\mathbb{P}$ and $\pi$, $d^{\mathbb{P}, \pi}$ signifies the metric $d^\pi$ evaluated on $\mathbb{P}$, and correspondingly $D_\Theta^{\mathbb{P}, \pi}$ denotes the metric $D_\Theta^\pi$ evaluated on $\mathbb{P}$.

## I.1  TRANSFER LEARNING BETWEEN IRL PROBLEMS

We introduce the transfer learning setting outlined in Metelli et al. (2021), where we consider two IRL problems: $(\mathcal{M}, \pi^{\mathsf{E}})$ (the source IRL problem), $(\mathcal{M}', (\pi')^{\mathsf{E}})$ (the target IRL problem). Here, $\mathcal{M}$, $\mathcal{M}'$ share the same state space and action space, but different dynamics. Suppose that we can learn the source MDP and obtain a reward $r$. However, $r$ is not necessarily a solution for $(\mathcal{M}', (\pi')^E)$, hence, in order to facilitate the transfer learning, we enforce the following assumption.

**Assumption I.1.** *If $r$ represents a solution to the source IRL problem $(\mathcal{M}, \pi^{\mathsf{E}})$, it also stands as a solution to the target IRL problem $(\mathcal{M}', (\pi')^E)$.*

We remark that in numerous practical scenarios, Assumption I.1 may not be precisely met, but could be approximated: within a certain distance, the solutions to the two IRL problems are very close to each other.

### I.2 TRANSFER BETWEEN TWO MDP\RS

In this section, we consider a more general setting, where we focus solely on a source IRL problem and a target MDP\R.

We consider two MDP\Rs $\mathcal{M} = (\mathcal{S}, \mathcal{A}, H, \mathbb{P})$ (source MDP\R) and $\mathcal{M}' = (\mathcal{S}, \mathcal{A}, H, \mathbb{P}')$ (the target MDP\R), which share the same state space and action space, but different dynamics and an expert policy $\pi^{\mathsf{E}}$. Let $\mathscr{R}^{\star}$ be the ground truth reward mapping of the IRL problem $(\mathcal{M}, \pi^{\mathsf{E}})$ and $\widehat{\mathscr{R}}$ be the estimated reward mapping learned from $(\mathcal{M}, \pi^{\mathsf{E}})$. In the precious setting, we consider $D^{\mathbb{P}, \pi}\left(\mathscr{R}^{\star}, \widehat{\mathscr{R}}\right)$ as performance metric. However, in transfer learning, we use $D^{\mathbb{P}', \pi}_{\Theta}\left(\mathscr{R}^{\star}, \widehat{\mathscr{R}}\right)$ as our measure of performance. That is we use the rewards learned by the source IRL problem to perform the same tasks in a different environment and evaluate them in the new environment.

As we see in Section 1, Inverse reinforcement learning(IRL) and behavioral cloning(BC) are highly related. As mentioned in Metelli et al. (2021), transfer learning makes IRL more powerful than BC, and a lot of literature has used IRL to do transfer learning (Syed & Schapire, 2007; Metelli et al., 2021; Abbeel & Ng, 2004; Fu et al., 2017; Levine et al., 2011).

Inspired by the single policy concentrability of policies, we propose the following transferability assumption.

**Definition I.2** (Weak-transferability). *Given transitions $(\mathbb{P}, \mathbb{P}')$, and policies $(\pi, \pi')$, we say $(\mathbb{P}', \pi')$ is $C^{\mathsf{wtran}}$-weakly transferable from $(\mathbb{P}, \pi)$ if it holds that*

$$\sup_{s,a} \frac{d_h^{\mathbb{P}', \pi'}(s,a)}{d_h^{\mathbb{P}, \pi}(s,a)} \leq C^{\mathsf{wtran}}.$$

**Definition I.3** (Transferability). *Given source and target transitions $\mathbb{P}$, $\mathbb{P}'$, and target policy $\pi'$, we say $\pi'$ is $C^{\mathsf{tran}}$-transferable from $\mathbb{P}$ to $\mathbb{P}'$ if it holds that*

$$\inf_{\pi} \sup_{s,a} \frac{d_h^{\mathbb{P}', \pi'}(s,a)}{d_h^{\mathbb{P}, \pi}(s,a)} \leq C^{\mathsf{tran}}.$$

Here, we introduce two notions: transferability and weak-transferability. We remark that given a policy $\pi$ and a dynamics $(\mathbb{P}, \mathbb{P}')$, transferability measures how hard one can lean the states $\pi'$ frequently goes to in $\mathbb{P}$ in a different environment $\mathbb{P}'$ while given a policy pair $(\pi, \pi')$ and a dynamics pair $(\mathbb{P}, \mathbb{P}')$, weak-transferability measures how hard one learn the states $\pi$ frequently visits in $\mathbb{P}$ via policy $\pi'$ in $\mathbb{P}'$. Without transferability, we can't obtain information on the policy of interest in the target MDP, which makes transfer learning hard to perform.

### I.3 THEORETICAL GUARANTEE

We then present the main theorem in this section.

**Theorem I.4** (Transfer learning in the offline setting). *Suppose $(\mathbb{P}', \pi^{\mathsf{val}})$ is $C^{\mathsf{wtran}}$-weakly transferable from $(\mathbb{P}, \pi^{\mathsf{b}})$ (Definition I.2). In addition, we assume $\pi^{\mathsf{E}}$ is well-posed (Definition 3.3) when we receive feedback in option 1. Then with probability at least $1 - \delta$, RLP (Algorithm 1) outputs a reward mapping $\widehat{\mathscr{R}}$ such that $\left[\widehat{\mathscr{R}}(V, A)\right]_h(s,a) \leq [\mathscr{R}^{\star}(V, A)]_h(s,a)$ for all $(V, A) \in \Theta$ and $(h, s, a) \in [H] \times \mathcal{S} \times \mathcal{A}$, and $D^{\mathbb{P}', \pi^{\mathsf{val}}}_{\Theta}\left(\mathscr{R}^{\star}, \widehat{\mathscr{R}}\right) \leq 2\epsilon$, within*

$$\widetilde{\mathcal{O}}\left( \frac{C^{\mathsf{wtran}} H^4 SA \log \mathcal{N}(\Theta; \epsilon/H)}{\epsilon^2} + \frac{C^{\mathsf{wtran}} H^2 SA (\eta + H \log \mathcal{N}(\Theta; \epsilon/H))}{\epsilon} \right), \qquad \text{(I.1)}$$

*samples, where $\text{poly} \log (H, S, A, 1/\delta)$ are omitted and $\eta$ is defined in Eq.(3.3).*

**Theorem I.5** (Transfer learning in the online setting)**.** *Suppose $\pi^{\mathsf{E}}$ is well-posed (Definition 3.4) when we receive feedback in option* 1*. Let $\mathscr{R}^{\star}$ be the ground truth reward mapping of IRL problem $(\mathcal{M}, \pi^{\mathsf{E}})$. Then with probability at least $1 - \delta$, RLE (Algorithm 6) outputs a reward mapping $\widehat{\mathscr{R}}$ such that $\left[\widehat{\mathscr{R}}(V, A)\right]_h (s, a) \leq [\mathscr{R}^{\star}(V, A)]_h (s, a)$ for all $(V, A) \in \Theta$ and $(h, s, a) \in [H] \times \mathcal{S} \times \mathcal{A}$, and*

$$\sup_{\pi^{\mathsf{val}} is \ C^{\mathsf{tran}}\text{-}transferable \ from \ \mathbb{P} \ to \ \mathbb{P}'} D_{\Theta}^{\mathbb{P}', \pi^{\mathsf{val}}}(\mathscr{R}^{\star}, \widehat{\mathscr{R}}) \leq 2\epsilon,$$

*provided that*

$$KH \geq N \geq \widetilde{\mathcal{O}}\left(\sqrt{H^9 S^7 A^7 K}\right),$$

$$K \geq \widetilde{\mathcal{O}}\left(\frac{C^{\mathsf{tran}} HSA\left(C^{\mathsf{tran}} + H^3 \log \mathcal{N}(\Theta; \epsilon/H)\right)}{\epsilon^2} + \frac{C^{\mathsf{tran}} H^2 SA(\eta + \log \mathcal{N}(\Theta; \epsilon/H))}{\epsilon}\right)$$

*where $\eta$ is defined in Eq.(3.3) and $\widetilde{\mathcal{O}}$ hides $poly \log (H, S, A, 1/\delta)$.*

**Application: performing RL algorithms in different environments** With Theorem I.4 and Theorem I.5 in place, as a concrete application, we consider to utilize rewards learned by IRL algorithms to execute RL algorithms in a different environment ($\mathcal{M}'$). The following two corollaries provide guarantees for the performance of learned rewards in executing RL algorithms in the offline and online setting, respectively. Both of these corollaries are direct consequences of Proposition C.6.

**Corollary I.6** (Performing RL algorithms with learned rewards in the offline setting)**.** *Fix $\theta = (V, A) \in \Theta$, let $r^{\theta} := \mathscr{R}^{\star}(V, A)$ and $\widehat{r}^{\theta} := \widehat{\mathscr{R}}(V, A)$, where $\widehat{\mathscr{R}}$ are recovered reward mapping outputted by Algorithm 1. suppose that there exists a policy $\pi$ such that $\pi$ is $\bar{\epsilon}$-optimal in MDP $\mathcal{M}' \cup r^{\theta}$ and $(\mathbb{P}', \pi)$ is $C^{\mathsf{wtran}}$-weakly transferable from $(\mathbb{P}, \pi^{\mathsf{b}})$ (Definition I.2). Let $\widehat{\pi}$ be an $\epsilon'$-optimal policy in $\mathcal{M}' \cup \widehat{r}^{\theta}$ (learned by some RL algorithms with $\widehat{r}^{\theta}$). Under the same assumption of Theorem I.4 , we have $V_1^{\star}(s_1; \mathcal{M}' \cup r^{\theta}) - V_1^{\widehat{\pi}}(s_1; \mathcal{M}' \cup r^{\theta}) \leq 2\epsilon + \epsilon' + 2\bar{\epsilon}$, provided that Algorithm 1 takes*

$$\widetilde{\mathcal{O}}\left(\frac{C^{\mathsf{wtran}} H^4 SA \log \mathcal{N}(\Theta; \epsilon/H)}{\epsilon^2} + \frac{C^{\mathsf{wtran}} H^2 SA(\eta + H \log \mathcal{N}(\Theta; \epsilon/H))}{\epsilon}\right), \qquad (\text{I.2})$$

*samples.*

**Corollary I.7** (Performing RL algorithms with learned rewards in the online setting)**.** *Fix $\theta = (V, A) \in \Theta$, let $r^{\theta} := \mathscr{R}^{\star}(V, A)$ and $\widehat{r}^{\theta} := \widehat{\mathscr{R}}(V, A)$, where $\widehat{\mathscr{R}}$ are recovered reward mapping outputted by Algorithm 6. suppose that there exists a policy $\pi$ such that $\pi$ is $\bar{\epsilon}$-optimal in MDP $\mathcal{M}' \cup r^{\theta}$ and $\pi$ is $C^{\mathsf{tran}}$-transferable from $\mathbb{P}$ to $\mathbb{P}'$ (Definition I.3). Let $\widehat{\pi}$ be an $\epsilon'$-optimal policy in $\mathcal{M}' \cup \widehat{r}^{\theta}$ (learned by some RL algorithms with $\widehat{r}^{\theta}$), then we have $V_1^{\star}(s_1; \mathcal{M}' \cup r^{\theta}) - V_1^{\widehat{\pi}}(s_1; \mathcal{M}' \cup r^{\theta}) \leq 2\epsilon + \epsilon' + 2\bar{\epsilon}$, provided that*

$$KH \geq N \geq \widetilde{\mathcal{O}}\left(\sqrt{H^9 S^7 A^7 K}\right),$$

$$K \geq \widetilde{\mathcal{O}}\left(\frac{C^{\mathsf{tran}} HSA\left(C^{\mathsf{tran}} + H^3 \log \mathcal{N}(\Theta; \epsilon/H)\right)}{\epsilon^2} + \frac{C^{\mathsf{tran}} H^2 SA(\eta + \log \mathcal{N}(\Theta; \epsilon/H))}{\epsilon}\right)$$

**Application: learning IRL problems by transfer learning** We return to the topic of transfer learning between IRL problems. We note that our findings related to transfer learning between MDP\Rs can also be employed in the context of transfer learning between IRL problems. As the illustrated in Theorem I.4 and Theorem I.5, we can efficiently learn a $\widehat{\mathscr{R}}$ such that the distance $D_{\Theta}^{\pi^{\mathsf{val}}}(\widehat{\mathscr{R}}, \mathscr{R}^{\star}) \leq 2\epsilon$, where $\mathscr{R}^{\star}$ is the ground truth reward mapping of $(\mathcal{M}, \pi^{\mathsf{E}})$. By Assumption I.1, the rewards induced by $\mathscr{R}^{\star}$ are solutions of $\left(\mathcal{M}', (\pi')^{\mathsf{E}}\right)$, hence the rewards induced by $\widehat{\mathscr{R}}$ approximate the solutions of $\left(\mathcal{M}', (\pi')^{\mathsf{E}}\right)$.

### I.4 PROOF OF THEOREM I.4

Note that under the same assumptions in Theorem I.4, the concentration event $\mathcal{E}$ defined in Lemma E.1 still holds with $1 - \delta$. By the week-transferablity of $(\pi^{\mathsf{val}}, \pi^{\mathsf{b}})$, we have

$$\sum_{h\in[H]} \sum_{(s,a)\in\mathcal{S}\times\mathcal{A}} \frac{d_h^{\mathbb{P}',\pi^{\mathsf{val}}}(s,a)}{d_h^{\mathbb{P},\pi^{\mathsf{b}}}(s,a)} \leq C^{\mathsf{wtran}} \sum_{h\in[H]} \sum_{(s,a)\in\mathcal{S}\times\mathcal{A}} \mathbf{1}\left\{d_h^{\mathbb{P}',\pi^{\mathsf{val}}}(s,a) \neq 0\right\}$$
$$\leq C^{\mathsf{wtran}} \sum_{h\in[H]} \sum_{(s,a)\in\mathcal{S}\times\mathcal{A}} \mathbf{1}\left\{a \in \pi_h^{\mathsf{val}}(\cdot|s)\right\} \leq C^{\mathsf{wtran}} HSA. \quad \text{(I.3)}$$

For any $\theta = (V, A) \in \Theta$, define $r^\theta = \mathscr{R}^\star(V, A)$, and $\widehat{r}^\theta = \widehat{\mathscr{R}}(V, A)$. With Eq.(I.3) at hand, we can repeat the proof of Lemma E.2, thereby obtaining that

$$d^{\mathbb{P}',\pi^{\mathsf{val}}}\left(r^\theta, \widehat{r}^\theta\right) \lesssim \frac{C^{\mathsf{wtran}} H^2 SA\eta\iota}{K} + \underbrace{\sum_{h\in[H]} \sum_{(s,a)\in\mathcal{S}\times\mathcal{A}} d_h^{\mathbb{P}',\pi^{\mathsf{val}}}(s,a)b_h^\theta(s,a)}_{\text{(I)}}, \quad \text{(I.4)}$$

holds on the event $\mathcal{E}$. where $\eta, b_h^\theta(s,a)$ are specified in Lemma E.1.

Furthermore, similar to Eq.(E.11), and through the application of the triangle inequality, we can decompose $\sum_{(s,a)\in\mathcal{S}\times\mathcal{A}} d_h^{\mathbb{P}',\pi^{\mathsf{val}}}(s,a)b_h^\theta(s,a)$ as follows:

$$\text{(I)} = \sum_{h\in[H]} \sum_{(s,a)\in\mathcal{S}\times\mathcal{A}} d_h^{\mathbb{P}',\pi^{\mathsf{val}}}(s,a)b_h^\theta(s,a)$$

$$\lesssim \sum_{h\in[H]} \sum_{(s,a)\in\mathcal{S}\times\mathcal{A}} d_h^{\pi^{\mathsf{val}}}(s,a) \cdot \left\{\sqrt{\frac{\log\mathcal{N}(\Theta;\epsilon/H)\iota}{N_h^b(s,a) \vee 1}\left[\widehat{\mathbb{V}}_h V_{h+1}\right](s,a)} + \frac{H\log\mathcal{N}(\Theta;\epsilon/H)\iota}{N_h^b(s,a) \vee 1}\right\}$$

$$+ \sum_{h\in[H]} \sum_{(s,a)\in\mathcal{S}\times\mathcal{A}} d_h^{\pi^{\mathsf{val}}}(s,a) \cdot \frac{\epsilon}{H}\left(1 + \sqrt{\frac{\log\mathcal{N}(\Theta;\epsilon/H)\iota}{N_h^b(s,a) \vee 1}}\right)$$

$$\leq \underbrace{\sum_{h\in[H]} \sum_{(s,a)\in\mathcal{S}\times\mathcal{A}} d_h^{\mathbb{P}',\pi^{\mathsf{val}}}(s,a) \cdot \sqrt{\frac{\log\mathcal{N}(\Theta;\epsilon/H)\iota}{N_h^b(s,a) \vee 1}[\mathbb{V}_h V_{h+1}](s,a)}}_{\text{(I.a)}}$$

$$+ \underbrace{\sum_{h\in[H]} \sum_{(s,a)\in\mathcal{S}\times\mathcal{A}} d_h^{\mathbb{P}',\pi^{\mathsf{val}}}(s,a) \cdot \sqrt{\frac{\log\mathcal{N}(\Theta;\epsilon/H)\iota}{N_h^b(s,a) \vee 1}\left[\left(\widehat{\mathbb{V}}_h - \mathbb{V}_h\right)V_{h+1}\right](s,a)}}_{\text{(I.b)}}$$

$$+ \underbrace{\sum_{h\in[H]} \sum_{(s,a)\in\mathcal{S}\times\mathcal{A}} d_h^{\mathbb{P}',\pi^{\mathsf{val}}}(s,a) \cdot \frac{H\log\mathcal{N}(\Theta;\epsilon/H)\iota}{N_h^b(s,a) \vee 1}}_{\text{(I.c)}}$$

$$+ \underbrace{\epsilon \cdot \sum_{h\in[H]} \sum_{(s,a)\in\mathcal{S}\times\mathcal{A}} d_h^{\mathbb{P}',\pi^{\mathsf{val}}}(s,a) \cdot \left(\frac{1}{H} + \sqrt{\frac{\log\mathcal{N}(\Theta;\epsilon/H)\iota}{H^2 \cdot N_h^b(s,a) \vee 1}}\right)}_{\text{(I.d)}}. \quad \text{(I.5)}$$

Thanks to Eq.(I.3), we can employ a similar argument as in the proof of Eq.(E.12), Eq.(E.17), Eq.(E.18), and Eq.(E.19), which allows us to deduce that

$$\text{(I.a)} \lesssim \sqrt{\frac{C^{\mathsf{wtran}} H^4 SA n^{\mathsf{E}} \log\mathcal{N}(\Theta;\epsilon/H)\iota}{K}},$$

$$\text{(I.b)} \lesssim \sqrt{\frac{C^{\mathsf{wtran}} H^2 SA \log\mathcal{N}(\Theta;\epsilon/H)}{K}} + \frac{C^{\mathsf{wtran}} H^3 SA \log\mathcal{N}(\Theta;\epsilon/H)\iota^{5/2}}{K},$$

$$(\text{I.c}) \lesssim \frac{C^{\text{wtran}} H^2 SA \log \mathcal{N}(\Theta; \epsilon/H) \iota}{K}, \epsilon \cdot (1 + \sqrt{\frac{C^{\text{wtran}} SA \log \mathcal{N}(\Theta; \epsilon/H) \iota}{K}}), \qquad (\text{I.6})$$

Combining Eq.(I.4), Eq.(I.5) and Eq.(I.6), we conclude that

$$D_{\Theta}^{\mathbb{P}', \pi^{\text{val}}} \left( \mathscr{R}^\star, \widehat{\mathscr{R}} \right) = \sup_{\theta \in \Theta} d^{\mathbb{P}', \pi^{\text{val}}} \left( r^\theta, \widehat{r}^\theta \right) \lesssim \frac{C^{\text{wtran}} H^2 SA \eta \iota}{K} + \sqrt{\frac{C^{\text{wtran}} H^4 SA n^{\text{E}} \log \mathcal{N}(\Theta; \epsilon/H) \iota}{K}}$$

$$+ \frac{C^{\text{wtran}} H^3 SA n^{\text{E}} \log \mathcal{N}(\Theta; \epsilon/H) \iota^{5/2}}{K} + \epsilon. \qquad (\text{I.7})$$

The right-hand-side is upper bounded by $2\epsilon$ as long as

$$K \geq \widetilde{\mathcal{O}} \left( \frac{C^{\text{wtran}} H^4 SA \log \mathcal{N}(\Theta; \epsilon/H)}{\epsilon^2} + \frac{C^{\text{wtran}} H^2 SA (\eta + H \log \mathcal{N}(\Theta; \epsilon/H))}{\epsilon} \right).$$

Here $poly \log (H, S, A, 1/\delta)$ are omitted.

## I.5   PROOF OF THEOREM I.5

Under the assumptions in Theorem I.5, the concentration event $\mathcal{E}$ defined in Lemma F.2 still holds with $1 - \delta$.   Fix $\pi$ such that $\pi$ satisfies $C^{\text{tran}}$-concentrability from $\mathbb{P}$ to $\mathbb{P}'$. We define

$$\bar{\mathcal{I}}_h := \left\{ (s,a) \in \mathcal{S} \times \mathcal{A} \mid \widehat{d}_h^{\mathbb{P}', \pi}(s,a) \geq \frac{\xi}{N} + e_h^\pi(s,a) \right\},$$

for all $h \in [H]$. Similar to Eq.(F.8), we have the following decomposition:

$$d^{\mathbb{P}', \pi} \left( r_h^\theta, \widehat{r}_h^\theta \right) \leq \sum_{(h,s,a) \in [H] \times \mathcal{S} \times \mathcal{A}} d_h^{\mathbb{P}', \pi}(s,a) \cdot \left| r_h^\theta(s,a) - \widehat{r}_h^\theta(s,a) \right|$$

$$\leq \underbrace{\sum_{h \in [H]} \sum_{(s,a) \notin \mathcal{I}_h \cup \bar{\mathcal{I}}_h} d_h^{\mathbb{P}', \pi}(s,a) \cdot \left| r_h^\theta(s,a) - \widehat{r}_h^\theta(s,a) \right|}_{(\text{I})} + \underbrace{\sum_{h \in [H]} \sum_{(s,a) \in \mathcal{I}_h \cup \bar{\mathcal{I}}_h} d_h^{\mathbb{P}', \pi}(s,a) \cdot \left| r^\theta(s,a) - \widehat{r}_h^\theta(s,a) \right|}_{(\text{II})},$$

$$(\text{I.8})$$

where set $\mathcal{I}_h$ is defined in Eq.(F.5).

We further decompose the term (I) as follows:

$$(\text{I}) \leq \underbrace{\sum_{h \in [H]} \sum_{(s,a) \notin \mathcal{I}_h} d_h^{\mathbb{P}', \pi}(s,a) \cdot \left| r_h^\theta(s,a) - \widehat{r}_h^\theta(s,a) \right|}_{(\text{I.a})} + \underbrace{\sum_{h \in [H]} \sum_{(s,a) \notin \bar{\mathcal{I}}_h} d_h^{\mathbb{P}', \pi}(s,a) \cdot \left| r_h^\theta(s,a) - \widehat{r}_h^\theta(s,a) \right|}_{(\text{I.b})}.$$

$$(\text{I.9})$$

By the definition of transferability, there exists a policy $\pi'$ such that

$$d^{\mathbb{P}', \pi}_h(s,a) \leq 2 C^{\text{tran}} d_h^{\mathbb{P}, \pi'}(s,a),$$

for any $(h,s,a) \in [H] \times \mathcal{S} \times \mathcal{A}$. For the term (I.a), we have

$$(\text{I.a}) = \sum_{h \in [H]} \sum_{(s,a) \notin \mathcal{I}_h \cup \bar{\mathcal{I}}_h} d_h^{\mathbb{P}', \pi}(s,a) \cdot \left| r_h^\theta(s,a) - \widehat{r}_h^\theta(s,a) \right| \leq 2 C^{\text{tran}} \sum_{(s,a) \notin \mathcal{I}_h \cup \bar{\mathcal{I}}_h} d_h^{\mathbb{P}, \pi'}(s,a) \cdot \left| r_h^\theta(s,a) - \widehat{r}_h^\theta(s,a) \right|$$

$$(\text{I.10})$$

Similar to Eq.(F.9), on the event $\mathcal{E}$, we have

$$\sum_{h \in [H]} \sum_{(s,a) \notin \mathcal{I}_h \cup \bar{\mathcal{I}}_h} d_h^{\mathbb{P}, \pi'}(s,a) \cdot \left| r_h^\theta(s,a) - \widehat{r}_h^\theta(s,a) \right| \leq \sum_{h \in [H]} \sum_{(s,a) \notin \bar{\mathcal{I}}_h} d_h^{\mathbb{P}, \pi'}(s,a) \cdot \left| r_h^\theta(s,a) - \widehat{r}_h^\theta(s,a) \right|$$

$$\lesssim \frac{\xi H^2 SA}{N} + \frac{H^2 SA \eta}{K} + \sqrt{\frac{HSA}{K}},$$

which allows us to bound the term (I.a) as follows:

$$\text{(I.a)} \lesssim \frac{C^{\mathsf{tran}}\xi H^2 SA}{N} + \frac{C^{\mathsf{tran}}H^2 SA\eta}{K} + C^{\mathsf{tran}}\sqrt{\frac{HSA}{K}}. \tag{I.11}$$

For the term (I.b), on the event $\mathcal{E}$, we have

$$\text{(I.b)} = \sum_{h \in [H]} \sum_{(s,a) \notin \bar{\mathcal{I}}_h} d_h^{\mathbb{P}',\pi}(s,a) \cdot \left| r^\theta(s,a) - \hat{r}^\theta(s,a) \right|$$

$$= \sum_{h \in [H]} \sum_{(s,a) \notin \bar{\mathcal{I}}_h} d_h^{\mathbb{P}',\pi}(s,a) \cdot \bigg| - A_h(s,a) \big( \mathbf{1}\left\{ a \in \mathrm{supp}\left(\pi_h^{\mathsf{E}}(\cdot|s)\right) \right\} - \mathbf{1}\left\{ a \in \mathrm{supp}\left(\hat{\pi}_h^{\mathsf{E}}(\cdot|s)\right) \right\} \big)$$

$$- \left[\left(\mathbb{P}_h - \widehat{\mathbb{P}}_h\right) V_{h+1}\right](s,a)(s,a) - b_h^\theta(s,a) \bigg|$$

$$\leq \sum_{h \in [H]} \sum_{(s,a) \notin \bar{\mathcal{I}}_h} d_h^{\mathbb{P}',\pi}(s,a) \cdot \bigg\{ \left| A_h(s,a) \cdot \left( \mathbf{1}\left\{ a \in \mathrm{supp}\left(\hat{\pi}_h^{\mathsf{E}}(\cdot|s)\right) \right\} - \cdot\mathbf{1}\left\{ a \in \mathrm{supp}\left(\pi_h^{\mathsf{E}}(\cdot|s)\right) \right\} \right) \right|$$

$$+ \left| \left[(\mathbb{P}_h - \widehat{\mathbb{P}}_h) V_{h+1}\right](s,a) \right| + b_h^\theta(s,a) \bigg\} \qquad \text{(by triangle inequality)}$$

$$\overset{(i)}{\lesssim} H \cdot \sum_{h \in [H]} \sum_{(s,a) \notin \bar{\mathcal{I}}_h} d_h^{\mathbb{P}',\pi}(s,a)$$

$$\overset{(ii)}{\leq} 2C^{\mathsf{tran}} H \cdot \sum_{h \in [H]} \sum_{(s,a) \notin \bar{\mathcal{I}}_h} \left( \widehat{d}_h^{\mathbb{P},\pi'}(s,a) + e_h^{\pi'}(s,a) + \frac{\xi}{N} \right)$$

$$\overset{(iii)}{\lesssim} C^{\mathsf{tran}} H \cdot \sum_{h \in [H]} \sum_{(s,a) \notin \bar{\mathcal{I}}_h} \left( e_h^{\pi'}(s,a) + \frac{\xi}{N} \right)$$

$$\lesssim \frac{C^{\mathsf{tran}}\xi H^2 SA}{N} + C^{\mathsf{tran}}\sqrt{\frac{HSA}{K}}, \tag{I.12}$$

where (i) is by $\|A_h\|_\infty$, $\|V_{h+1}\|_\infty$, $b_h^\theta(s,a) \leq H$, (ii) comes from Eq.(F.2) and the concentration event $\mathcal{E}(ii)$, and (iii) follows from the definition of $\bar{\mathcal{I}}_h$.

Combining Eq.(I.11) and Eq.(I.12), we can conclude that

$$\text{(I)} \lesssim \frac{C^{\mathsf{tran}}\xi H^2 SA}{N} + \frac{C^{\mathsf{tran}}H^2 SA\eta}{K} + C^{\mathsf{tran}}\sqrt{\frac{HSA}{K}}. \tag{I.13}$$

For the term (II), following a similar approach as in Eq.(F.10), we have

$$\text{(II)} = \sum_{h \in [H]} \sum_{(s,a) \in \mathcal{I}_h \cup \bar{\mathcal{I}}_h} d_h^{\mathbb{P}',\pi}(s,a) \cdot b_h^\theta(s,a)$$

$$= \sum_{h \in [H]} \sum_{(s,a) \in \mathcal{I}_h \cup \bar{\mathcal{I}}_h} d_h^{\mathbb{P}',\pi}(s,a) \cdot \min\left\{ \sqrt{\frac{\log \mathcal{N}(\Theta; \epsilon/H)\iota}{\widehat{N}_h^b(s,a) \vee 1}} \left[\widehat{\mathbb{V}}_h V_{h+1}\right](s,a) + \frac{H \log \mathcal{N}(\Theta; \epsilon/H)\iota}{\widehat{N}_h^b(s,a) \vee 1} \right.$$

$$\left. + \frac{\epsilon}{H}\left(1 + \sqrt{\frac{\log \mathcal{N}(\Theta; \epsilon/H)\iota}{\widehat{N}_h^b(s,a) \vee 1}}\right), H \right\}$$

$$\overset{(i)}{\leq} \sum_{h \in [H]} \sum_{(s,a) \in \mathcal{I}_h \cup \bar{\mathcal{I}}_h} d_h^{\mathbb{P}',\pi}(s,a) \cdot \left\{ \min\left\{ \sqrt{\frac{\log \mathcal{N}(\Theta; \epsilon/H)\iota}{\widehat{N}_h^b(s,a) \vee 1}} \left[\widehat{\mathbb{V}}_h V_{h+1}\right](s,a), H \right\} + \frac{H \log \mathcal{N}(\Theta; \epsilon/H)\iota}{\widehat{N}_h^b(s,a) \vee 1} \right.$$

$$\tag{I.14}$$

$$\left. + \frac{\epsilon}{H}\left(1 + \sqrt{\frac{\log \mathcal{N}(\Theta; \epsilon/H)\iota}{\widehat{N}_h^b(s,a) \vee 1}}\right) \right\}$$

$$
\stackrel{(ii)}{\leq} \sum_{h \in [H]} \sum_{(s,a) \in \mathcal{I}_h \cup \bar{\mathcal{I}}_h} d_h^{\mathbb{P}',\pi}(s,a) \cdot \left\{ \sqrt{\frac{\log \mathcal{N}(\Theta; \epsilon/H) \iota \left[\widehat{\mathbb{V}}_h V_{h+1}\right](s,a) + H}{\widehat{N}_h^b(s,a) \vee 1 + 1/H}} + \frac{H \log \mathcal{N}(\Theta; \epsilon/H) \iota}{\widehat{N}_h^b(s,a) \vee 1} \right.
$$

$$
\left. + \frac{\epsilon}{H} \left( 1 + \sqrt{\frac{\log \mathcal{N}(\Theta; \epsilon/H) \iota}{\widehat{N}_h^b(s,a) \vee 1}} \right) \right\}
$$

$$
\stackrel{(iii)}{=} \underbrace{\sum_{h \in [H]} \sum_{(s,a) \in \mathcal{I}_h \cup \bar{\mathcal{I}}_h} d_h^{\mathbb{P}',\pi}(s,a) \cdot \sqrt{\frac{\log \mathcal{N}(\Theta; \epsilon/H) \iota \left[\widehat{\mathbb{V}}_h V_{h+1}\right](s,a) + H}{K \mathbb{E}_{\pi' \sim \mu^b} \left[\widehat{d}_h^{\pi'}(s,a)\right] + 1/H}}}_{(II.a)}
$$

$$
+ \underbrace{\sum_{h \in [H]} \sum_{(s,a) \in \mathcal{I}_h \cup \bar{\mathcal{I}}_h} d_h^{\mathbb{P}',\pi}(s,a) \cdot \frac{H \log \mathcal{N}(\Theta; \epsilon/H) \iota}{K \mathbb{E}_{\pi' \sim \mu^b} \left[\widehat{d}_h^{\pi'}(s,a)\right] + 1/H}}_{(II.b)}
$$

$$
+ \underbrace{\frac{\epsilon}{H} \sum_{h \in [H]} \sum_{(s,a) \in \mathcal{I}_h \cup \bar{\mathcal{I}}_h} d_h^{\mathbb{P}',\pi}(s,a) \cdot \left( 1 + \sqrt{\frac{\log \mathcal{N}(\Theta; \epsilon/H) \iota}{K \mathbb{E}_{\pi' \sim \mu^b} \left[\widehat{d}_h^{\pi'}(s,a)\right] + 1/H}} \right)}_{(II.c)}. \tag{I.15}
$$

For the term (II.a), by the Cauchy-Schwarz inequality, we have

$$
(II.a) \leq \sqrt{C^{\text{tran}}} \sum_{h \in [H]} \sum_{(s,a) \in \mathcal{I}_h \cup \bar{\mathcal{I}}_h} \sqrt{d_h^{\mathbb{P}',\pi}(s,a) \cdot (\log \mathcal{N}(\Theta; \epsilon/H) \iota [\mathbb{V}_h V_{h+1}](s,a) + H)}
$$

$$
\cdot \sqrt{\frac{d_h^{\mathbb{P},\pi'}(s,a)}{K \mathbb{E}_{\pi' \sim \mu^b} \left[\widehat{d}_h^{\pi'}(s,a)\right] + 1/H}}
$$

$$
\lesssim \sqrt{C^{\text{tran}}} \sum_{h \in [H]} \sum_{(s,a) \in \mathcal{I}_h \cup \bar{\mathcal{I}}_h} \sqrt{d_h^{\mathbb{P}',\pi}(s,a) \cdot (\log \mathcal{N}(\Theta; \epsilon/H) \iota [\mathbb{V}_h V_{h+1}](s,a) + H)}
$$

$$
\cdot \sqrt{\frac{2 \widehat{d}_h^{\mathbb{P},\pi'}(s,a) + 2 e_h^{\pi'}(s,a) + \frac{\xi}{2N}}{K \mathbb{E}_{\pi' \sim \mu^b} \left[\widehat{d}_h^{\pi'}(s,a)\right] + 1/H}}
$$

$$
\lesssim \sqrt{C^{\text{tran}}} \sum_{h \in [H]} \sum_{(s,a) \in \mathcal{I}_h \cup \bar{\mathcal{I}}_h} \sqrt{d_h^{\mathbb{P}',\pi}(s,a) \cdot (\log \mathcal{N}(\Theta; \epsilon/H) \iota [\mathbb{V}_h V_{h+1}](s,a) + H)}
$$

$$
\cdot \sqrt{\frac{\widehat{d}_h^{\mathbb{P},\pi'}(s,a)}{K \mathbb{E}_{\pi' \sim \mu^b} \left[\widehat{d}_h^{\pi'}(s,a)\right] + 1/H}}
$$

$$
\leq \sqrt{C^{\text{tran}}} \underbrace{\left\{ \sum_{h \in [H]} \sum_{(s,a) \in \mathcal{S} \times \mathcal{A}} d_h^{\mathbb{P}',\pi}(s,a) \cdot (\log \mathcal{N}(\Theta; \epsilon/H) \iota [\mathbb{V}_h V_{h+1}](s,a) + H) \right\}^{1/2}}_{(II.a.1)}
$$

$$
\times \underbrace{\left\{ \sum_{h \in [H]} \sum_{(s,a) \in \mathcal{S} \times \mathcal{A}} \frac{\widehat{d}_h^{\mathbb{P},\pi'}(s,a)}{K \mathbb{E}_{\pi' \sim \mu^b} \left[\widehat{d}_h^{\pi'}(s,a)\right] + 1/H} \right\}^{1/2}}_{(II.a.2)}.
$$

Following similar approaches as in Eq.(F.12) and Eq.(F.13), we have

$$
(II.a.1) \lesssim \sqrt{H^3 \log \mathcal{N}(\Theta; \epsilon/H) \iota}, \qquad (II.a.2) \lesssim \sqrt{\frac{HSA}{K}}, \tag{I.16}
$$

which implies that

$$\text{(II.a)} \lesssim \sqrt{\frac{C^{\text{tran}} H^4 SA \log \mathcal{N}(\Theta; \epsilon/H)\iota}{K}}. \tag{I.17}$$

For the term (II.b), by Eq.(B.3), we have

$$
\begin{aligned}
\text{(II.b)} &= \sum_{h \in [H]} \sum_{(s,a) \in \mathcal{S} \times \mathcal{A}} d_h^{\mathbb{P}', \pi}(s,a) \cdot \frac{H \log \mathcal{N}(\Theta; \epsilon/H)\iota}{K \mathbb{E}_{\pi' \sim \mu^{\text{b}}}\left[\widehat{d}_h^{\pi'}(s,a)\right] + 1/H} \\
&= \frac{C^{\text{tran}}}{K} \cdot \sum_{h \in [H]} \sum_{(s,a) \in \mathcal{S} \times \mathcal{A}} d_h^{\mathbb{P}, \pi'}(s,a) \cdot \frac{H \log \mathcal{N}(\Theta; \epsilon/H)\iota}{\mathbb{E}_{\pi' \sim \mu^{\text{b}}}\left[\widehat{d}_h^{\pi'}(s,a)\right] + 1/KH} \\
&\lesssim \frac{C^{\text{tran}}}{K} \cdot \sum_{h \in [H]} \sum_{(s,a) \in \mathcal{S} \times \mathcal{A}} \widehat{d}_h^{\mathbb{P}, \pi'}(s,a) \cdot \frac{H \log \mathcal{N}(\Theta; \epsilon/H)\iota}{\mathbb{E}_{\pi' \sim \mu^{\text{b}}}\left[\widehat{d}_h^{\pi'}(s,a)\right] + 1/KH} \\
&\lesssim \frac{C^{\text{tran}} H^2 SA \log \mathcal{N}(\Theta; \epsilon/H)\iota}{K}. \tag{I.18}
\end{aligned}
$$

For the term (II.c), we have

$$
\begin{aligned}
\text{(II.c)} &= \frac{\epsilon}{H} \sum_{h \in [H]} \sum_{(s,a) \in \mathcal{I}_h \cup \bar{\mathcal{I}}_h} d_h^{\mathbb{P}, \pi}(s,a) \cdot \left(1 + \sqrt{\frac{\log \mathcal{N}(\Theta; \epsilon/H)\iota}{K \mathbb{E}_{\pi' \sim \mu^{\text{b}}}\left[\widehat{d}_h^{\pi'}(s,a)\right] + 1/H}}\right) \\
&= \epsilon + \frac{\sqrt{C^{\text{tran}}}\epsilon}{H} \sum_{h \in [H]} \sum_{(s,a) \in \mathcal{I}_h \cup \bar{\mathcal{I}}_h} \sqrt{d_h^{\mathbb{P}', \pi}(s,a)} \cdot \sqrt{\frac{d_h^{\mathbb{P}, \pi'}(s,a) \log \mathcal{N}(\Theta; \epsilon/H)\iota}{K \mathbb{E}_{\pi' \sim \mu^{\text{b}}}\left[\widehat{d}_h^{\pi'}(s,a)\right] + 1/H}} \\
&\lesssim \epsilon + \frac{\sqrt{C^{\text{tran}}}\epsilon}{H} \sum_{h \in [H]} \sum_{(s,a) \in \mathcal{I}_h \cup \bar{\mathcal{I}}_h} \sqrt{d_h^{\mathbb{P}', \pi}(s,a)} \cdot \sqrt{\frac{\widehat{d}_h^{\mathbb{P}, \pi'}(s,a) \log \mathcal{N}(\Theta; \epsilon/H)\iota}{K \mathbb{E}_{\pi' \sim \mu^{\text{b}}}\left[\widehat{d}_h^{\pi'}(s,a)\right] + 1/H}} \\
&\leq \epsilon + \frac{\epsilon}{H} \sqrt{\sum_{h \in [H]} \sum_{(s,a) \in \mathcal{I}_h \cup \bar{\mathcal{I}}_h} d_h^{\mathbb{P}', \pi}(s,a)} \cdot \sqrt{\sum_{h \in [H]} \sum_{(s,a) \in \mathcal{I}_h} \frac{\widehat{d}_h^{\mathbb{P}, \pi'}(s,a) \log \mathcal{N}(\Theta; \epsilon/H)\iota}{K \mathbb{E}_{\pi' \sim \mu^{\text{b}}}\left[\widehat{d}_h^{\pi'}(s,a)\right] + 1/H}} \\
&\leq \epsilon \left(1 + \sqrt{\frac{C^{\text{tran}} SA \log \mathcal{N}(\Theta; \epsilon/H)\iota}{HK}}\right), \tag{I.19}
\end{aligned}
$$

where the second last line is by the Cauchy-Schwarz inequality and the last line is by Eq.(B.3).

Then combining Eq.(I.17) Eq.(I.18), and Eq.(I.19), we obtain the bound for the term (II)

$$
\begin{aligned}
\text{(II)} &\lesssim \text{(II.a)} + \text{(II.b)} + \text{(II.c)} \\
&\lesssim \sqrt{\frac{C^{\text{tran}} H^4 SA \log \mathcal{N}(\Theta; \epsilon/H)\iota}{K}} + \frac{C^{\text{tran}} H^2 SA \log \mathcal{N}(\Theta; \epsilon/H)\iota}{K} + \epsilon\left(1 + \sqrt{\frac{C^{\text{tran}} \log \mathcal{N}(\Theta; \epsilon/H)\iota}{HK}}\right) \\
&\lesssim \sqrt{\frac{C^{\text{tran}} H^4 SA \log \mathcal{N}(\Theta; \epsilon/H)\iota}{K}} + \epsilon, \tag{I.20}
\end{aligned}
$$

where the last line is from $\epsilon < 1$.

Finally, combining Eq.(I.13) and Eq.(I.20), we get the final bound

$$
\begin{aligned}
D_\Theta^{\text{all}}\left(\mathscr{R}^\star, \widehat{\mathscr{R}}\right) &= \sup_{\pi, \theta \in \Theta} d^\pi\left(r_h^\theta, \widehat{r}_h^\theta\right) \leq \text{(I)} + \text{(II)} \\
&\lesssim \frac{C^{\text{tran}} \xi H^2 SA}{N} + \sqrt{\frac{C^{\text{tran}} H^4 SA \log \mathcal{N}(\Theta; \epsilon/H)\iota}{K}} + \frac{C^{\text{tran}} H^2 SA \eta}{K} + C^{\text{tran}} \sqrt{\frac{HSA}{K}} + \epsilon
\end{aligned}
$$

Hence, we can guarantee $D_\Theta^{\text{all}}\left(\mathscr{R}^\star, \widehat{\mathscr{R}}\right) \leq 2\epsilon$, provided that

$$KH \geq N \geq \widetilde{\mathcal{O}}\left(\sqrt{H^9 S^7 A^7 K}\right),$$

$$K \geq \widetilde{\mathcal{O}}\left(\frac{C^{\mathrm{tran}}HSA\big(C^{\mathrm{tran}} + H^3 \log \mathcal{N}(\Theta; \epsilon/H)\big)}{\epsilon^2} + \frac{C^{\mathrm{tran}}H^2SA(\eta + \log \mathcal{N}(\Theta; \epsilon/H))}{\epsilon}\right)$$

Here $poly \log (H, S, A, 1/\delta)$ are omitted.

