$$- \left\{-A_1'(\bar{s}, \bar{a}) \cdot \mathbf{1}\left\{\bar{a} \in \text{supp}\left(\widehat{\pi}_1^{\mathsf{E}}(\cdot|\bar{s})\right)\right\} + V_1'(s) - [\mathbb{P}_1' V_2'](\bar{s}, \bar{a})\right\}\Big|$$

$$= |[\mathbb{P}_1' V_2'](\bar{s}, \bar{a}) - [\mathbb{P}_1 V_2](\bar{s}, \bar{a})|$$

$$= |V_2(s^{\star}) - [\mathbb{P}_1 V_2](\bar{s}, \bar{a})| = 0 \tag{C.12}$$

On the other hand, for any $(V', A') \in \bar{\mathcal{V}} \times \bar{\mathcal{A}}$, we define $(V, A) \in \bar{\mathcal{V}} \times \bar{\mathcal{A}}$ by

$$\begin{cases} V_2(s) = V_2'(s^{\star}), & s \in \mathcal{S}, \\ V_h(s) = V_h'(s) & h \neq 2, \end{cases} \tag{C.13}$$

which implies that

$$[\mathbb{P}_1 V_2](\bar{s}, \bar{a}) = V_2'(s^{\star}) = [\mathbb{P}_1' V_2'](\bar{s}, \bar{a}). \tag{C.14}$$

Hence, we have

$$d\Big(\mathscr{R}(V, A), \widehat{\mathscr{R}}(V', A')\Big) = |[\mathscr{R}(V, A)]_1(\bar{s}, \bar{a}) - [\mathscr{R}(V, A)]_1(\bar{s}, \bar{a})| \tag{C.15}$$

$$= \Big| - A_1(\bar{s}, \bar{a}) \cdot \mathbf{1}\left\{\bar{a} \in \text{supp}\left(\pi_1^{\mathsf{E}}(\cdot|\bar{s})\right)\right\} + V_1(s) - [\mathbb{P}_1 V_2](\bar{s}, \bar{a})$$

$$- \left\{-A_1'(\bar{s}, \bar{a}) \cdot \mathbf{1}\left\{\bar{a} \in \text{supp}\left(\widehat{\pi}_1^{\mathsf{E}}(\cdot|\bar{s})\right)\right\} + V_1'(s) - [\mathbb{P}_1' V_2'](\bar{s}, \bar{a})\right\}\Big|$$

$$= |[\mathbb{P}_1' V_2'](\bar{s}, \bar{a}) - [\mathbb{P}_1 V_2](\bar{s}, \bar{a})| = 0 \tag{C.16}$$

Combining Eq.(C.11) and Eq.(C.15), we have $D^{\mathsf{H}}(\mathcal{R}, \widehat{\mathcal{R}}) = 0$.

On the other hand, we define $(\widetilde{V}, \widetilde{A}) \in \bar{\mathcal{V}} \times \bar{\mathcal{A}}$ as follows:

$$\begin{cases} \widetilde{V}_2(s^{\star}) = H - 1, \\ \widetilde{V}_h(s) = 0, \quad (h, s) \neq (2, s^{\star}), \end{cases} \qquad \widetilde{A} \equiv \mathbf{0}. \tag{C.17}$$

Then we have

$$D^{\mathsf{M}}(\mathscr{R}, \widehat{\mathscr{R}}) \geq d\Big(\mathscr{R}(\widetilde{V}, \widetilde{A}), \widehat{\mathscr{R}}(\widetilde{V}, \widetilde{A})\Big) = \Big|\left[\mathbb{P}_1' \widetilde{V}_2\right](\bar{s}, \bar{a}) - \left[\mathbb{P}_1 \widetilde{V}_2\right](\

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

$$
d_h^{\pi^{\mathsf{b}}}(s_{\text{start}}, a_0) = d_h^{\pi^{\mathsf{val}}}(s_{\text{start}}, a_0) = 1
\tag{H.3}
$$

for all $h \in [H-1]$.

At stage $h = H$, we have

$$
d_H^{\pi^{\mathsf{b}}}(s_{\text{start}}, a_{i^\star}) = \frac{1}{K}, \qquad d_H^{\pi^{\mathsf{val}}}(s_{\text{start}}, a_{i^\star}) = 1.
\tag{H.4}
$$

At stage $h = H + 1$, we have

$$
d_{H+1}^{\pi^{\mathsf{b}}}(s_{i^\star}, a_1) = \frac{1}{C^\star K}, \qquad d_{H+1}^{\pi^{\mathsf{val}}}(s_{i^\star}, a_1) = 1.
\tag{H.5}
$$

At stage $h \in \{H+2, \ldots, 2H+2\}$, by direct computation, we obtain that

$$
d_h^{\pi^{\mathsf{b}}}(\bar{s}_j, a_0) = \frac{C^\star K - 1}{C^\star SK} + \frac{1 + \epsilon' w_{H+1}^{(i^\star, j, 1)}}{C^\star SK}, \qquad d_h^{\pi^{\mathsf{val}}}(\bar{s}_j, a_1) = \frac{1 + \epsilon' w_{H+1}^{(i^\star, j, 1)}}{S},
\tag{H.6}
$$

for all $j \in [S]$. Since $0 < \epsilon \leq 1/2$ and $C^\star \geq 1$, we have

$$
\begin{aligned}
d_h^{\pi^{\mathsf{b}}}(\bar{s}_j, a_0) &= \frac{C^\star K - 1}{SK} + \frac{1 + \epsilon' w_{H+1}^{(i^\star, j, 1)}}{C^\star SK} \\
&\geq \frac{C^\star K - 1}{SK} + \frac{1}{2C^\star SK} = \frac{1}{S}\left(1 - \frac{1}{2C^\star K}\right) \geq \frac{1}{2S}
\end{aligned}
\tag{H.7}
$$

and

$$
d_h^{\pi^{\mathsf{val}}}(s_{i^\star}, a_1) = \frac{1 + \epsilon' w_{H+1}^{(i^\star, j, 1)}}{S} \leq \frac{3}{2S},
\tag{H.8}
$$

for all $h \geq H + 2$. By Eq.(H.7) and (H.7), we obtain that

$$\frac{d_h^{\pi^{\mathrm{val}}}(\bar{s}_j, a_0)}{d_h^{\pi^{\mathrm{b}}}(\bar{s}_j, a_0)} \leq 3, \tag{H.9}$$

for all $h \geq H + 2$.

Combining Eq.(H.3), Eq.(H.4) and Eq.(H.5), we have

$$\sum_{h=1}^{2H+2} \sum_{(s,a) \in \mathcal{S} \times \mathcal{A}} \frac{d^{\pi^{\mathrm{val}}}(s,a)}{d^{\pi^{\mathrm{b}}}(s,a)} = \sum_{h \in [H-1]} \frac{d_h^{\pi^{\mathrm{val}}}(s_{\mathrm{start}}, a_0)}{d_h^{\pi^{\mathrm{b}}}(s_{\mathrm{start}}, a_0)} + \frac{d_H^{\pi^{\mathrm{val}}}(s_{\mathrm{start}}, a_{i^\star})}{d_H^{\pi^{\mathrm{val}}}(s_{\mathrm{