# OpenReview forum: "Is Inverse Reinforcement Learning Harder than Standard Reinforcement Learning?"
_ICLR.cc/2024/Conference — Submitted to ICLR 2024_

### Official Review · Reviewer_cGXj · 2023-10-30

**Soundness:** 3 good
**Presentation:** 2 fair
**Contribution:** 3 good
**Rating:** 6
**Confidence:** 3

**Summary:**

This paper studies the problem of inverse reinforcement learning: given observations of an expert policy deployed in a (tabular) MDP, what reward function can one learn which is consistent with the actions of the expert policy? They consider both the offline (batch) setting as well as the online setting. They propose a metric for inverse reinforcement learning, defined in terms of the difference between the learned and ground truth rewards.

For the offline setting, they propose a pessimistic algorithm and show it achieves poly(S, A, H) sample complexity, as well as some dependence on the concentrability coefficient. For the online setting, they propose an algorithm which also achieves poly(S, A, H) sample complexity (and uses the offline algorithm as a subroutine). There is also an extension to a "transfer learning" setting.

**Strengths:**

- The paper is thorough, detailed, and mathematically rigorous. (As a disclaimer, I only checked in detail the proofs and associated lemmas for Thm 4.2, their main result on the sample complexity of the offline algorithm; but I am familiar with most of the literature and techniques used for online/offline tabular RL, so the results seemed correct to me.)
- To me, the most interesting contribution is the new notion of "performance", which is measured in terms of a uniform distance for a set of "reward mappings" (Defn 2.1). I think this notion of metric merits future discussion and is a valuable contribution to this area. However, I have some comments/questions about it - see below.
- On the algorithmic side, this paper also shows how several existing techniques (i.e., pessimism, reachable state identification / reward free exploration) can also be adapted to the IRL problem. While the algorithms and analysis themselves are not particularly novel, it is nice that we can use well-studied techniques in RL for the IRL problem.

**Weaknesses:**

- Given that this paper proposes new metrics for IRL, I found the discussion / comparison with previous work a bit vague. At times, the language and writing was a bit informal, and sometimes confusing to interpret.
    - For example, in the appendix you write that "our method that considers is greater than theirs". What does it mean for a method to be greater than another method? Did you mean your metric?
    - You write in C.3 that the "metric can't capture transitions". What does this actually mean?
- I would urge the authors to rewrite their comparison with prior work in Appendix C, focusing on cases where one subsumes the other, and giving concrete examples when your algorithm(s) can achieve guarantees while previous algorithms cannot. What is currently lacking is a distinction in the writing between (1) comparing the quantities $d^\pi$ and $d^\mathrm{all}$ themselves to prior work; (2) comparing guarantees that your algorithms can achieve to guarantees that algorithms from prior work can achieve.
- (minor) weakness: the upper/lower bounds seem to be loose. In particular, it would be good if the authors commented on the fact that the lower bound has a $\min \{S, A\}$ term - where does this come from? can it be improved? In general, the regime that we care about is when $S \ge A$, so the fact that the rate is only sharp when $S \le A$ is not very meaningful.

Minor comments:
- Many typos throughout, especially in the appendix. Some examples:
    - "Broaderly" in the first paragraph.
    - e.g., at bottom of page 4, "Given a policy $\pi$, We" -> lower case the "we".
    - a missing citation before Corollary 4.3? I see an "()".
    - "week transferability" should be "weak transferability"?
    - some typo in the equation (C.6).
    - At the beginning of page 23, in the first display equation, should it be $a \in \mathrm{supp}$ instead of $a \notin \mathrm{supp}$?
- For lower bounds, use $\Omega$, not $O$.
- In the algorithm pseudocode, a quantifier over the set $\Theta$ is missing: The algorithm needs to be run separately for every $\theta \in \Theta$.

**Questions:**

1. I'm a bit confused about the discussion of the Hausdorff metric used by Metelli et al. The quantity that they propose seems to be some notion of diameter. I'm not sure why it should go to zero as you collect more samples, as even when $\mathcal{R} = \widehat{\mathcal{R}}$ the quantity doesn't seem like it should be zero since the set of possible rewards could be large. But your results imply that an upper bound on this quantity goes to zero, so what am I missing here?
2. While the definition of the metric seems mathematically well defined, it seems a bit weird: to define the reward mapping, you need a value and advantage function (V,A) (e.g., Eq. 3.2). However, inside the definition of the metric, you define a new value function of a particular policy and reward function (e.g., in Eq. 3.1). How do these two value functions relate?

---

> ### Author Response · Authors · 2023-11-20
> **Response Part 1**
>
> We thank the reviewer for the valuable feedback on our paper. We respond to the specific questions as follows.
>
> > At times, the language and writing was a bit informal and sometimes confusing to interpret.
>
> We appreciate the reviewer for bringing up this question. We have made extensive revisions and polished Appendix C significantly in the latest version. We have addressed the issues you mentioned, and you can review the latest version.  For example, regarding "our method that considers is greater than theirs", detailed explanations are provided in the discussion above Lemma C.1 in the most recent version. Concerning "metric can't capture transitions",  comprehensive explanations can be found in the discussion above Proposition C.4 in the latest version.
>
> > I would urge the authors to rewrite their comparison with prior work in Appendix C, focusing on cases where one subsumes the other, and giving concrete examples when your algorithm(s) can achieve guarantees while previous algorithms cannot. What is currently lacking is a distinction in the writing between (1) comparing the quantities $d^{\pi}$ and $d^{\sf all}$ themselves to prior work; (2) comparing guarantees that your algorithms can achieve to guarantees that algorithms from prior work can achieve.
>
> Thank you very much for your suggestions. We have made substantial revisions in the latest version of Appendix C as per your requests (see Appendix C in the latest version).
>
>
> > (minor) weakness: the upper/lower bounds seem to be loose. In particular, it would be good if the authors commented on the fact that the lower bound has a min{S,A} term - where does this come from? can it be improved? In general, the regime that we care about is when S≥A, so the fact that the rate is only sharp when S≤A is not very meaningful.
>
> About $\textrm{min}\\{S,A\\}$ factor: Regarding $\mathrm{min}\\{S,A\\}$ term, the introduction of $\textrm{min}\\{S,A\\}$ is introduced as follows. In constructing hard instances, there is a step where we need to ensure $\mathbb{P}_h ( s_i| s_i,a_i)=1$ for $i\leq \textrm{min}\\{S,A\\}$. This particular step is crucial for our proof. Exploring ways to improve the lower bound on $\textrm{min}\\{S, A\\}$ could be a promising avenue for future research. We believe our results are still meaningful, and the condition $S < O(A)$ also holds significance. Regarding the lower bound in the online setting, we have made improvements (Theorem G.2), improving the original $\mathcal{O}(H^2SA\textrm{min}\\{S,A\\})$ to $\mathcal{O}(H^3SA\textrm{min}\\{S,A\\})$. Although it does not exactly match the upper bound, it has been significantly tightened. We appreciate the reviewer for bringing up this question.
>
>
> About $H$ factor: Regarding the lower bound in the online setting, we have made improvements (Theorem G.2), improving the original $\mathcal{O}(H^2SA\textrm{min}\\{S,A\\})$ to $\mathcal{O}(H^3SA\textrm{min}\\{S,A\\})$. Although it does not exactly match the upper bound, it has been significantly tightened. We appreciate the reviewer for bringing up this question. As for the lower bound in the offline setting, even though we employed the same MDP construction as in proving the lower bound in the online setting, we did not fully exploit the property of our MDP construction, where transitions are different for each h, because of the construction of the behavior policy. This results in our lower bound having a dependence on $H$ of $\mathcal{O}(H^2)$. How to prove a tighter lower bound for the $H$-factor would be a highly intriguing avenue for future research.
>
> > I'm a bit confused about the discussion of the Hausdorff metric used by Metelli et al. The quantity that they propose seems to be some notion of diameter. I'm not sure why it should go to zero as you collect more samples, as even when $\mathcal{R}=\widehat{\mathcal{R}}$ the quantity doesn't seem like it should be zero since the set of possible rewards could be large. But your results imply that an upper bound on this quantity goes to zero, so what am I missing here?
>
> If we understood correctly, the reviewer is asking why the Hausdorff metric appears to be a kind of diameter for sets and why it tends to zero? The Hausdorff metric $D^{\sf H}$ (see Section C.2) is indeed a measure of distance between sets, not a diameter. For instance, when two sets are equal, the Hausdorff metric becomes zero. And so the upper bound goes to zero.

---

> ### Author Response · Authors · 2023-11-20
> **Response Part 2**
>
> > While the definition of the metric seems mathematically well defined, it seems a bit weird: to define the reward mapping, you need a value and advantage function (V,A) (e.g., Eq. 3.2). However, inside the definition of the metric, you define a new value function of a particular policy and reward function (e.g., in Eq. 3.1). How do these two value functions relate?
>
> $(V, A) \in \mathcal{V} \times \mathcal{A}$ represents a set of parameters, not a value function. Each pair $(V, A)$ determines a solution to an IRL problem (Lemma 2.2). Let $r^\theta = \mathscr{R}(V, A)$, where $\mathscr{R}$ is a reward mapping induced by an IRL problem $(\mathcal{M}, \pi^E)$. The connection between the parameter $\theta = (V, A)$ and the value function defined in Eq. 3.1 on Page 3 is as follows: $V^{\pi^E}_h(s; r^\theta) = V_h(s)$ and $A^{\pi^E}_h(s, a; r^\theta)=A_h(s,a)$.
>
>
> > Many typos throughout, especially in the appendix.
>
> We thank the reviewer for this feedback. We have addressed and corrected the majority of the typos as suggested.

---

> > ### Comment · Reviewer_cGXj · 2023-11-22
> > **Reviewer Acknowledgement**
> >
> > Thank you for your response. I have no further questions at this point.

---

### Official Review · Reviewer_uRxz · 2023-10-31

**Soundness:** 3 good
**Presentation:** 3 good
**Contribution:** 3 good
**Rating:** 8
**Confidence:** 2

**Summary:**

This paper proposes RIP and RLE method which opterates in offline and online IRL settings respectively. These methods do have some theoretical guarantees.

**Strengths:**

1. This paper builds metrics for both online and offline IRL settings.
2. Informed by the pessimism principle, RLP is proposed for offline IRL setting with theoretical guarantees.
3. RLE achieves great sample complexity compared to other online IRL methods.

**Weaknesses:**

Lack of some experiment results.

I think the theoretical analysis for IRL is important. However, the title of this paper is a little confusing. The sample complexity analysis seems not relevant to this title.

**Questions:**

None

---

> ### Author Response · Authors · 2023-11-20
> **Response**
>
> We thank the reviewer for the valuable feedback on our paper. We respond to the specific questions as follows.
>
> > ... the title of this paper is a little confusing.
>
> Our paper's title “Is Inverse Reinforcement Learning Harder than Standard Reinforcement Learning?” aims to convey that we can apply standard RL techniques to address IRL problems, including techniques such as pessimism and some exploring strategies. I apologize for any confusion. In the latest version, we provide the framework for our IRL algorithms in Section E.4. Here, we give a condition (Condition E.4) that is sufficient to do IRL. Then, we illustrate that pessimism can fulfill Condition E.4. We appreciate the reviewer for bringing up this question.

---

> > ### Comment · Reviewer_uRxz · 2023-11-23
> > **Thanks for your response**
> >
> > Thanks for your response. I would like to keep my score.

---

### Official Review · Reviewer_RnbG · 2023-11-01

**Soundness:** 3 good
**Presentation:** 1 poor
**Contribution:** 3 good
**Rating:** 5
**Confidence:** 3

**Summary:**

This paper gives theoretical guarantees for both online and offline setting in the inverse reinforcement problem.

**Strengths:**

I think this work really pushes the inverse RL community research efforts further by answering:

> can we give theoretical guarantees for the inverse RL algorithm enforced by pessimism for offline and exploration for online settings?

This is a really nice idea worthy for publication. My score reflect the weaknesses.

**Weaknesses:**

Firstly note that this is an emergency review. I will rely on discussion with authors and reviewers, and other reviews, to decide on the score.

I have only a few weaknesses for this work as follows:

- The main paper writing needs to be improved. Yes, the soundness of this paper maybe good, considering similar pessimism and exploration ideas from past works. But this submission looks like a hurried submission with many typos like `Theoretical low bound in offline setting.`, ` $\pi_b = \pi_E ().,$`, `RLP utilizes empirical MDP and Pessimism frameworks`, `for all offline IRL problems with probability at least $1 − \delta$, has to take at least ... samples in expectation.` (both h.p. and in expectation?!), and so on. In addition to typos, the out-of-margin equation formatting makes a strenuous reading experience. To be honest, I am not sure if the authors can fix this writing issue of 56 pages during the rebuttal period, but I will welcome some attempts since proceedings require good quality.

- The closest work I can think of is [1] that provides theoretical guarantees for imitation learning ($\approx$ reward-free IRL+RL) in both online and offline data setting. The current work stops at reward learning, that is, the IRL problem. But without the extra RL step using the learned reward, is an incomplete story. The paper talks about similarities with RLHF; yes, there is a connection but one needs to eventually learn the optimal policy. Yes, one can just do planning with the learned reward and learned transition model, but equipping it with traditional model-based guarantees is important for making connections with other relevant works. _Model based guarantee_ will be unsatisfactory since [1] gives results for general function approximation.

- Moreover, this manuscript subsumes many results from (Li et al., 2023) which is a non-peer reviewed work. This makes it hard to check soundness since one needs to evaluate both (at least the relevant parts required for this submission). I am mentioning this due to my emergency review. My score reflects the fact that generative model setting (samples from every state-action pairs $\approx$ uniform concentrability) in Lindner et al. (2023), is equipped with pessimism and optimism terms to account for partial concentrability using the usual techniques in offline RL (Rashidinejad, et al. 2021) and thereafter.

I am open to discussions with the authors and reviewers to make sure the work quality matches the score, which I believe so at this point, but a potential reach to 6 or 8 definitely exists. All the best for future decisions!

[1] Jonathan D. Chang, Masatoshi Uehara, Dhruv Sreenivas, Rahul Kidambi, Wen Sun. Mitigating Covariate Shift in Imitation Learning via Offline Data Without Great Coverage, NeurIPS 2021.

**Questions:**

-na-

---

> ### Author Response · Authors · 2023-11-20
> **Response**
>
> We thank the reviewer for the valuable feedback on our paper. We respond to the specific questions as follows.
>
> > The main paper writing needs to be improved...
>
> We sincerely apologize for the confusion caused by writing issues. Thank you very much for pointing out the concrete writing issues, and we have corrected most of them in the latest version.
>
> > The current work stops at reward learning, that is, the IRL problem. But without the extra RL step using the learned reward, is an incomplete story… Yes, one can just do planning with the learned reward and learned transition model, but equipping it with traditional model-based guarantees is important for making connections with other relevant works. Model-based guarantee will be unsatisfactory since [1] gives results for general function approximation.
>
> Regarding the use of learned rewards, in fact, we do have guarantees for the performance of performing RL algorithms with the learned rewards. I apologize for any confusion; our previous versions did not emphasize this part. In the latest version, we specifically discuss guarantees for performing RL algorithms with learned rewards (see Section C.4 and Corollary I.6, I.7 in the latest version). We appreciate the reviewer for bringing up this question.
>
> [1] introduces an algorithm for imitation learning under function approximation, analyzing numerous concrete examples such as discrete MDPs, linear models, and GP models. However, our paper focuses on Inverse Reinforcement Learning (IRL) within the tabular setting. Regarding “Yes, one can just do planning with the learned reward and learned transition model, but equipping it with traditional model-based guarantees is important for making connections with other relevant works. Model-based guarantee will be unsatisfactory since [1] gives results for general function approximation.”, we may not have fully understood your meaning. Could you please provide further clarification or explanation?
>
> > My score reflects the fact that the generative model setting (samples from every state-action pairs ≈ uniform concentrability) in Lindner et al. (2023), is equipped with pessimism and optimism terms to account for partial concentrability using the usual techniques in offline RL (Rashidinejad, et al. 2021) and thereafter.
>
> While the algorithm in  Lindner et al. (2023) also incorporates pessimism, we emphasize a fundamental difference in the purpose of pessimism between  Lindner et al. (2023) and our work. In  Lindner et al. (2023), the $E^h_k(s,a)$ defined in Eq (FBI) serves as an uncertainty measure, and the algorithm minimizes this $E^h_k(s,a)$ at each step to determine the sampling strategy (see Section 6.1 in Lindner et al. (2023)). On the other hand, our introduction of pessimism aims to impose a high penalty on state-action pairs that are rarely visited. This is done to ensure that the recovered rewards satisfy the monotonicity condition: $\widehat{\mathscr{R}}_h(s,a)\leq \mathscr{R}^\star_h(s,a)$. This condition is crucial for obtaining guarantees in performing RL algorithms with learned rewards, as detailed in our latest version (see Proposition C.6 and Section E.4).
>
>  [1] Jonathan D. Chang, Masatoshi Uehara, Dhruv Sreenivas, Rahul Kidambi, Wen Sun. Mitigating Covariate Shift in Imitation Learning via Offline Data Without Great Coverage, NeurIPS 2021.

---

> > ### Comment · Reviewer_RnbG · 2023-11-21
> > **Clarification**
> >
> > Thank you for the detailed response.
> >
> > Regarding:
> > > Yes, one can just do planning with the learned reward and learned transition model, but equipping it with traditional model-based guarantees is important for making connections with other relevant works.
> >
> > Here I tried to bring up the "incomplete" story of IRL. IRL stops at learning the reward functions. But how do you then do policy optimization with this reward function for making comparisons with RL? Of course, one can do model-based (empirical estimates of both reward and transition function) RL. But equipping it with sample complexity results will become crucial.
> >
> > I do think the authors have improved the manuscript compared to pre-rebuttal stage. I will update my score post authors-reviewers discussions.

---

> > > ### Author Response · Authors · 2023-11-21
> > > **Thank you for your response**
> > >
> > > Thank you for your clarifications!
> > >
> > > In the latest version, we have included the sample complexity for extra RL steps with learned rewards (see Corollary I.6 & I.7). We appreciate the reviewer for bringing up this question.

---

> > > > ### Author Response · Authors · 2023-11-22
> > > > **Confirmation for score**
> > > >
> > > > Dear reviewer,
> > > >
> > > > Just wanted to confirm that you would be updating the score after the author-reviewer discussion period ends today? We would love to have a confirmation (would also appreciate if you can update it now), as the deadline for us to post messages is ending soon.
> > > >
> > > > Best,
> > > > Authors

---

### Official Review · Reviewer_9kcm · 2023-11-01

**Soundness:** 4 excellent
**Presentation:** 4 excellent
**Contribution:** 2 fair
**Rating:** 5
**Confidence:** 5

**Summary:**

The paper builds on recent works on Inverse Reinforcement Learning (IRL). It introduces a new metric different from the literature to analyze the distance between the feasible set of reward functions compatible with an expert's demonstrations in the limit of infinite samples and the estimated set. Their metric actually considers the distance between two mappings whose image set coincides with the feasible set. Next, the paper analyzes the sample complexity of estimating this mapping in the offline and online (forward model) settings. They provide algorithms for both the settings (and therefore upper bounds to the sample complexity) as well as lower bounds. Finally, the paper provides a sample complexity analysis on the error obtained when doing transfer learning with the estimated mapping, i.e., when transferring the estimated mapping instead of the true one.

**Strengths:**

- the main strengths are the sample complexity results for the offline and online (forward model) settings for IRL, which are the first results of this kind for IRL;
- the idea of connecting transferability with concentrability when analyze the transfer learning setting is interesting

**Weaknesses:**

- no technical novelty in the proofs which are based on (i) previous works on offline RL and (ii) previous works on IRL
- the lower bounds (both for offline and online settings) are definitely not tight, since they do not depend on the confidence $\delta$
- the metric introduced to evaluate the complexity in the offline setting is actually not a metric in the mathematical sense

**Questions:**

I am willing to adjust my score if the authors successfully answer my questions.

1) Why do you use $d^\pi$ (Definition 3.1) as a (pre)metric between reward functions? This is not a metric, and therefore the problem that you highlight in Lemma C.4 for the Hausdorff distance between sets sussists also for your (pre)metric $D_\Theta^\pi$ (Definition 3.2). It might be that the proposed (pre)metric $D_\Theta^\pi$ is zero even if the reward sets do not coincide.

2) What is the rationale behind choosing the (pre)metric in Definition 3.1? In the usual pipeline, the reward recovered by IRL is then used for training RL agents. It does not seem to me that the (pre)metric in Definition 3.1 guarantees anything on how close the performance of the trained RL agent with the learned reward. In "Towards Theoretical Understanding of Inverse Reinforcement Learning" the authors focus on the actual distance between reward functions, not induced value functions. Can the author elaborate?

3) Why do your lower bounds not depend on $\delta$? This is quite significant especially when $\delta$ is small. Inspecting the proofs in comparison with  "Towards Theoretical Understanding of Inverse Reinforcement Learning", it seems that the authors have adapted one construction only (the one that provides the part of the lower bound that does not depend on $\delta$). Why this choice?

4) How did you manage in the proof of the upper bound the fact that rewards defined as in the ground truth reward mapping are not bounded in $[-1,+1]$ even though the parameters $(V,A)\in\mathcal{V}\times\mathcal{A}$? In "Towards Theoretical Understanding of Inverse Reinforcement Learning", Lemma B.1 (appendix), they normalize the reward functions, but you don't. What allows you to avoid this step?

5) The lower bounds do not match the upper bounds, especially for what concerns the dependence on the horizon H. What is the reason for this gap?

COMMENTS:
- a section with the limitations of the results is missing and should be added;
- the title has nothing to do with the paper; the paper concerns a sample complexity analysis in the IRL setting, stop. Nothing in the paper gives novel results on whether IRL is harder than RL, so the title must be changed;
- the proof of the second part of Lemma C.1, although easy, is missing;
- the proof of Proposition C.2 is missing;
- in Section 1, Introduction, when listing the contributions, authors state that this work contributes at "providing an answer to the
longstanding non-uniqueness issue in IRL". This statement is factually false. Indeed, the authors investigate IRL as the reconstruction of the feasible reward set (and this formulation is not introduced in the paper), providing novel analysis for the offline and online (with forward model);
- the use of O-tilde is incorrect for what concerns the dependence on $\delta$. Conventionally, O-tilde does not hide dependences on $\log(1/\delta)$. This is not just a cosmetic comment, but seems to hide an additional term present in the upper bound and not present in the lower bound, spotting an additional term that is not matched;
- you use the same symbol $d$ for both the visit distribution and the metric, and maybe you could change one of the two symbols to improve the presentation;
- the paper contains many typos both in the main paper and in the appendix.

---

> ### Author Response · Authors · 2023-11-20
> **Response Part 1**
>
> We thank the reviewer for the valuable feedback on our paper. We respond to the specific questions as follows.
>
> > Why do you use $d^\pi$ (Definition 3.1) as a (pre)metric between reward functions?
>
> * Unlike generative models, in the offline setting, there are unreachable state-action pairs. We cannot expect the recovered rewards to be very close to the ground truth reward for every state-action pair. As a result, in the offline setting, many classical metrics such as $L_{\infty}$ distance between rewards cannot be employed. Therefore, considering the use of policy and value function to define is a very natural choice.
>
> * $d^\pi$ (Definition 3.1) can provide guarantees for performing RL algorithms or doing transfer learning with learned rewards (see Section C.2 and I.3).
>
>
>
> > ...This is not a metric, and therefore the problem that you highlight in Lemma C.4 for the Hausdorff distance between sets sussists also for your (pre)metric $D^\pi_{\Theta}$ (Definition 3.2)...It might be that the proposed (pre)metric $D^\pi_{\Theta}$is zero even if the reward sets do not coincide.
>
> We first clarify that Lemma C.4 (Lemma C.3  in the latest version) is intended to compare the Hausdorff metric and mapping-based metric, rather than metrics between rewards. The reason for “the proposed (pre)metric $D^\pi_{\Theta}$is zero even if the reward sets do not coincide” is due to the selection of metric between rewards: $d^{\pi}$ (it is easy to see that $d^\pi(r,r')=0$ does not imply $r=r'$). Lemma C.4 (Lemma C.3  in the latest version) aims to demonstrate that $D^{\mathsf{M}}$ is strictly stronger than $D^{\mathsf{H}}$ for some $d$, but lacks a discussion of the mapping-based metric in the first version. Therefore, we have modified Lemma C.4 (Lemma C.3  in the latest version) to include a discussion of both the Hausdorff metric and the mapping-based metric (See Lemma C.3 in in the latest version).
>
> > What is the rationale behind choosing the (pre)metric in Definition 3.1?... It does not seem to me that the (pre)metric in Definition 3.1 guarantees anything on how close the performance of the trained RL agent with the learned reward.In "Towards Theoretical Understanding of Inverse Reinforcement Learning" the authors focus on the actual distance between reward functions, not induced value functions. Can the author elaborate?
>
> Regarding the rationale behind choosing the (pre)metric in Definition 3.1 and why we do not use the actual distance between reward functions, please refer to the answer to “ Why do you use $d^\pi$ (Definition 3.1) as a (pre)metric between reward functions?”.
>
> Regarding the lack of guarantees for the performance of the trained RL agent with the learned reward, in fact, we do have guarantees for the performance of the trained RL agent with the learned rewards. We apologize for any confusion; our previous versions did not emphasize this part. In the latest version, we specifically discuss guarantees for performing RL algorithms with learned rewards (see Section C.4 and Corollary I.6, I.7). We appreciate the reviewer for bringing up this question.
>
> > Why do your lower bounds not depend on $1-\delta$?
>
> We apologize for any confusion caused. We failed to clarify that our lower bounds also hold for small $\delta$, such as $\delta \leq 1/3$ in the first version.  This error has been corrected in our latest version. We appreciate the reviewer for bringing up this question. We believe that our results hold true when $\delta$ is very small, and thus, they are highly meaningful.
>
> > How did you manage in the proof of the upper bound the fact that rewards defined as in the ground truth reward mapping are not bounded in $ [−1,+1]$  even though the parameters $(V,A)\in\mathcal{V}\times \mathcal{A}$?
>
> Although the reward is not bounded within $[-1, 1]$, we emphasize that $\mathcal{V}\times \mathcal{A}$ is bounded (see the Definition of $\mathcal{V}\times \mathcal{A}$ on page 4). Specifically when $(V,A)\in \mathcal{V}\times \mathcal{A})$, we have $|V_h|_{\infty},|A_h|_{\infty}\leq H-h+1$, which is sufficient for our proof (see the proof of Theorem 4.2).

---

> ### Author Response · Authors · 2023-11-20
> **Response Part 2**
>
> > The lower bounds do not match the upper bounds, especially for what concerns the dependence on the horizon H. What is the reason for this gap?
>
> Regarding the lower bound in the online setting, we have made improvements (Theorem G.2), improving the original $\mathcal{O}(H^2SA\textrm{min}\\{S,A\\})$ to $\mathcal{O}(H^3SA\textrm{min}\\{S,A\\})$. Although it does not exactly match the upper bound, it has been significantly tightened. We appreciate the reviewer for bringing up this question.
>
> As for the lower bound in the offline setting, even though we employed the same MDP construction as in proving the lower bound in the online setting, we did not fully exploit the property of our MDP construction, where transitions are different for each h, because of the construction of the behavior policy. This results in our lower bound having a dependence on $H$ of $\mathcal{O}(H^2)$. How to prove a tighter lower bound for the $H$-factor would be a highly intriguing avenue for future research.
>
> Regarding the $\mathrm{min}\\{S,A\\}$ term, the $\textrm{min}\\{S,A\\}$ factor is introduced as follows. In constructing hard instances, there is a step where we need to ensure $\mathbb{P}_h ( s_i| s_i,a_i)=1$ for $i\leq \textrm{min}\\{S,A\\}$. This particular step is crucial for our proof. Exploring ways to improve the lower bound on $\textrm{min}\\{S, A\\}$ could be a promising avenue for future research. We believe our results are still meaningful, and the condition $S < O(A)$ also holds significance.
>
> > ‘The proof of the second part of Lemma C.1, although easy, is missing;’ and ‘the proof of Proposition C.2 is missing;’
>
> Thank you for pointing these out. We have added the proofs in Section C.5.
>
> > In Section 1, Introduction, when listing the contributions, the authors state that this work contributes at "providing an answer to the longstanding non-uniqueness issue in IRL". This statement is factually false. Indeed, the authors investigate IRL as the reconstruction of the feasible reward set (and this formulation is not introduced in the paper), providing novel analysis for the offline and online (with forward model);
>
> We appreciate the reviewer's feedback. We want to emphasize that, to the best of our knowledge, we are the first to consider IRL using reward mapping instead of just feasible sets. Additionally, we introduce a mapping-based performance metric to address the no-uniqueness problem. We believe our approach differs from previous work (see Section C).
>
> > the title has nothing to do with the paper; the paper concerns a sample complexity analysis in the IRL setting, stop. Nothing in the paper gives novel results on whether IRL is harder than RL, so the title must be changed;
>
> Our paper's title “Is Inverse Reinforcement Learning Harder than Standard Reinforcement Learning?” aims to convey that we can apply standard RL techniques to address IRL problems, including techniques such as pessimism and some exploring strategies. I apologize for any confusion. In the latest version, we provide the framework for our IRL algorithms in Section E.4. Here, we give a condition (Condition E.4) that is sufficient to do IRL. Then, we illustrate that pessimism can fulfill Condition E.4. We appreciate the reviewer for bringing up this question.
>
>
> > About symbols and typos.
>
> We appreciate this suggestion. We have addressed the modifications related to symbols and typos as suggested in the latest version.

---

> > ### Comment · Reviewer_9kcm · 2023-11-22
> > **Re: Response Part 2**
> >
> > I thank the authors for the detailed answers and for having updated the paper. I have carefully re-read the paper and still I have some concerns that have not been fully solved by the authors.
> >
> > - **[On the use of $d^{pi}$]** While I fully agree on the impossibility of using the $L_\infty$-norm of the difference between rewards, I don't really agree on the fact that the expectation under $d^{\pi}$ is a proper index. Indeed, the authors claim that " $d^{\pi}$ (Definition 3.1) can provide guarantees for performing RL algorithms or doing transfer learning with learned rewards (see Section C.2 and I.3).", but looking at Section C.2, only with $d^{all}$ it is possible to have guarantees when performing RL. It doesn't seem that for the offline setting the authors are able to provide guaranteed w.r.t. $d^{all}$. This makes the argument in favor of $d^{\pi}$ quite weak in my opinion.
> >
> > - **[On the dependence on $\delta$]** I am not satisfied by the authors' answer about the role of $\delta$ in the bounds. The authors claim that the bounds hold for $\delta \le 1/3$. However, there is no presence of $\delta$ in the sample complexity lower and upper bounds. It is hard to believe that such dependence is really not present since when $\delta \rightarrow 0$ clearly the sample complexity must diverge to infinity. One possibility is that the authors have hidden it in the $\widetilde{O}$. I have already pointed out that the **proper use of $\widetilde{O}$ should not hide the polynomial dependence on $\log(1/\delta)$**. This is indeed the case for the upper bounds reported in the paper. However, looking carefully at the lower bounds proofs, there is no dependence on $\log(1/\delta)$ (but only on $1-\delta$). This spots an important lack of tightness since the lower bound does not diverge to infinity when $\delta\rightarrow 0$. **I want to stress that this in combination with an inappropriate use of $\widetilde{O}$ is severely hiding a suboptimality of the lower bound in comparison with the upper bounds**.
> >
> > - **[Title]** I remain convinced that the chosen title is inappropriate.
> >
> > - **[Pessimism]** By re-reading the paper, I realized that the use of pessimism in the reward function has a quite strange effect compared to pessimism in standard off-line RL. Here, pessimism is directly applied to the reward function (eq. 4.4) but it does not seem to have a significant role in the ability to achieve the desired results. From a technical perspective, looking at eq. (E.8), if we do not use pessimism we just obtain the bonus $b^\theta_h(s,a)$ instead of twice the bonus $2 b^\theta_h(s,a)$; but the pessimism has no impact on the computation of $C^*$. This is radically different compared to the case of off-line RL where the pessimism is essential to obtain the desired covering wrt to the optimal policy (instead of the undesirable uniform covering over all the policies). I am aware that I am raising this point by the end of the discussion period, but I think that **if the role of pessimism is so marginal in the paper (as I suspect at this point) and if without using it, it is possible to derive the same results, this is an important concern to report. I would really appreciate it if the authors could say whether without pessimism they can obtain (apart from constants) the same results.**
> >
> > I am reserving the possibility to adjust my score after the discussion with the other reviewers.

---

> ### Author Response · Authors · 2023-11-22
> **Response to additional questions**
>
> We thank the reviewer for the additional questions. We respond to them as follows.
>
> > **[On the use of $d^\pi$]** … looking at Section C.2, only with $d^{\rm all}$ it is possible to have guarantees when performing RL. It doesn't seem that for the offline setting the authors are able to provide guaranteed w.r.t. . This makes the argument in favor of $d^\pi$  quite weak in my opinion.
>
> We apologize we gave the **wrong pointer** there, it should be Proposition C.6 in Section C.4.
>
> There we show that, a small $d^\pi$ where $\pi$ is any $\bar\epsilon$ near-optimal policy, combined with monotonicity of the recovered reward ($\hat{r}\le r$ pointwise), ensures an RL guarantee with $\hat{r}$. So $d^\pi$ plus this additional monotonicity condition indeed gives an RL guarantee. Further, our RLP algorithm also guarantees monotonicity (see the blue text in Theorem 4.2, bottom of Page 7) along with the $D^{\pi}$ guarantee. We believe this justifies the $d^\pi$ metric as well as our offline results.
>
>
> > **[On the dependence on $\delta$]** …proper use of $\tilde{O}(\cdot)$ should not hide the polynomial dependence on $\log(1/\delta)$...  This spots an important lack of tightness since the lower bound does not diverge to infinity when $\delta\to 0$.
>
> For our upper bounds, we were indeed using $\tilde{O}(\cdot)$ to suppress $\log(1/\delta)$ and the other polylog factors, as we explained in each occurrence of it in the upper bounds.
>
> For our lower bounds, indeed, the statements are only for $\delta=1/3$ (which then implies the same statement for all $\delta\le 1/3$) for which $\log(1/\delta)$ is a constant. We have slightly tweaked the statements to clarify this, but we agree with the reviewer that this is looser than what we want.
>
> Upon initial inspection, we believe it is **possible to improve our lower bounds to have an additional $\log(1/\delta)$ factor**, by using the inequality $1-{\rm TV}\ge \frac{1}{2}e^{-\rm KL}\ge \delta$ whenever ${\rm KL}\le \log(1/(2\delta))$ instead of Pinsker’s inequality, and arguing that $n\le O(\log(1/\delta))$ (times the other factors) will result in ${\rm KL}\le \log(1/(2\delta))$ (as KL scales linearly in the sample size $n$ due to its additivity). However, we need to go through the full argument more carefully. We will include this in our final version if it works out.
>
> > **[Pessimism]** By re-reading the paper, I realized that the use of pessimism in the reward function has a quite strange effect compared to pessimism in standard off-line RL. Here, pessimism is directly applied to the reward function (eq. 4.4) but it does not seem to have a significant role in the ability to achieve the desired results… if the role of pessimism is so marginal in the paper (as I suspect at this point) and if without using it, it is possible to derive the same results, this is an important concern to report. I would really appreciate it if the authors could say whether without pessimism they can obtain (apart from constants) the same results.
>
> We agree that for reward estimation ($d^\pi$ or $d^{\rm all}$ guarantee) alone, a simple empirical mean estimator is enough, as the reviewer pointed out.
>
> However, **pessimism additionally ensures (entrywise) monotonicity of the learned reward** (cf. blue text in Theorem 4.2). The role of this monotonicity is in **downstream RL** (Proposition C.6,  Thm/Corollaries I.4->I.7), where monotonicity is required for the learned policy (from the recovered reward) to have guarantees.
>
> This is fully aligned with the use of pessimism in standard offline RL (as per the reviewer’s question), where the monotonicity of the rewards ($\hat{r}\le r$ pointwise) is crucial for achieving learning guarantees under single-policy concentrability only instead of all-policy concentrability. For example, in offline bandits, we need to rule out seldom-chosen bad arms (not covered by the optimal policy) that happen to have a large empirical reward by a big negative bonus.
>
> If the reviewer agrees with this, we then also believe this kind of parallellization to standard offline RL gives **some justification to our title**. As an example, we gave a more **modular argument in Section E.4 on how pessimism implies IRL** ($d^\pi$ bound + monotonicity) in a black-box fashion. (Looking back, we probably should have structured the main results using this, where we first show pessimism implies IRL black-box by this argument, then reduce to standard pessimistic offline RL algorithms to achieve the pessimism. We will think about whether we can restructure our paper along this direction in its next version).
>
> ---
> We appreciate again the reviewer's prompt response to our rebuttal, and would be happy to answer additional questions before the rebuttal period ends.

---

### Comment · Area_Chair_YcLf · 2023-11-17
**From AC.**

Dear Authors,

Thank you for submitting your paper to ICLR!

If possible, can you:

1. Answer the questions asked by reviewers.
2. It would be great if you posted a comment explaining how your approach to IRL reward construction compares to that explained in [1]. The approach in [1] has some similarity to the setting of option 2 in your equation 3.3. For the avoidance of doubt, I am not asking you to cite the paper [1], just to post a comment explaining how your work is related to theirs.

[1] K. Ciosek "Imitation Learning by Reinforcement Learning"

---

> ### Author Response · Authors · 2023-11-20
> **Response to AC**
>
> Dear AC,
>
> Thank you for your question and pointing out the work [1]. The algorithm in [1] utilizes an MDP solver to solve an MDP where the reward is the indicator of the expert policy, followed by imitation learning. It is important to note that the algorithm in [1] is limited to imitation learning and cannot learn rewards. In contrast, our algorithm not only enables imitation learning (using an MDP solver to solve the  MDP with the rewards learned by our algorithms RLP and RLE) (see Section C.4) but also provides performance guarantees for using the learned rewards to perform RL in different environments via transfer learning (see Corollary I.6, I.7), a feature not achievable by the imitation Algorithm in [1].

---

### Author Response · Authors · 2023-11-20
**Revision uploaded**

We thank all reviewers again for their valuable feedback on our paper. We have incorporated the reviewers’ suggestions and uploaded a revised version of our paper. For clarity, all changes (other than typo fixes) are marked in blue.


Best,
Authors

---

### Author Response · Authors · 2023-11-22
**Additional comment on "modular reduction" and our contributions**

Dear Reviewers and AC,

We'd like to make an additional short remark on our contributions, which we realized during the interactions with Reviewer 9kcm and RnbG.

Overall, our RLP algorithm and its proof technique can be viewed as a **modular reduction from offline IRL to standard offline RL** consisting of two steps:
1. Pessimism implies accuracy (in $d^\pi$ and $d^{\rm all}$) and monotonicity ($\hat{r}\le r$) of learned rewards in a black-box fashion; we provided such a modular argument in Section E.4 in our revision. Monotonicity is further important for using the learned rewards to do RL in both the original MDP and transfer learning settings (Proposition C.6, Thm/Corollaries I.4->I.7);
2. Standard offline RL algorithm can achieve the desired pessimism in step 1.

Please refer to our latest response to Reviewer 9kcm for more details about this.

We believe this overall message is new and of interest to the IRL literature. Further, it gives an justification of our title (partly, that was why we chose it in the first place), as it shows how offline IRL can be reduced to standard offline RL, and thus "IRL is not much harder".

Best,
Authors

---

### Meta-Review · Area_Chair_YcLf · 2023-12-05

**Metareview:**

This paper addresses the question of characterising the sample complexity of inverse reinforcement learning (finding the set of reward feasible for a given expert dataset). The problem of finding the rewards is considered on its own, without the associated question of what happens when we want to use that reward to learn an imitator policy (which is OK).

Strengths:
- relatively novel setting
- extensive theoretical analysis of the problem

Weaknesses:
- there needs to be more discussion about why pessimism is necessary for the proofs to go through. Pessimism is a crucial component of the proposed algorithm, so this is a major issue.
- reviewer RnbG found issues with the math in the revised version. Specifically:
  * In Prop C.6, eq.C(27) does not follow from the one-sided inequality in eq.C(26).
  * $d^\pi$ is used to denote both visitation distribution and the distance.
- the dependence on the terms $\log(1/\delta)$ should be made explicit (as per reviewer 9kcm).
- ideally the paper should include table-lookup experiments to ensure the method is practical. Having said that, this is not strictly required and was not the main reason for the reject recommendation
- the title of the paper is not a great match for the content. I encourage the authors to resubmit with a different title at a later date.

**Justification For Why Not Higher Score:**

There seems to be consensus among the reviewers RnbG, 9kcm and cGXj that the paper isn't ready at this stage. This does not change the fact that the paper has many good ideas and is likely to be more successful the next time round. Also see the list of weaknesses in the meta-review.

**Justification For Why Not Lower Score:**

N/A

---

### Decision · Program_Chairs · 2024-01-16

Reject